# Sample complexity of data-driven tuning of model hyperparameters in neural networks with structured parameter-dependent dual function

**Maria-Florina Balcan**
Carnegie Mellon University
`ninamf@cs.cmu.edu`

**Anh Tuan Nguyen**
Carnegie Mellon University
`atnguyen@cs.cmu.edu`

**Dravyansh Sharma**
Toyota Technological Institute at Chicago
`dravy@ttic.edu`

## Abstract

Modern machine learning algorithms, especially deep learning-based techniques, typically involve careful hyperparameter tuning to achieve the best performance. Despite the surge of intense interest in practical techniques like Bayesian optimization and random search-based approaches to automating this laborious and compute-intensive task, the fundamental learning-theoretic complexity of tuning hyperparameters for deep neural networks is poorly understood. Inspired by this glaring gap, we initiate the formal study of hyperparameter tuning complexity in deep learning through a recently introduced data-driven setting. We assume that we have a series of learning tasks, and we have to tune hyperparameters to do well on average over the distribution of tasks. A major difficulty is that the utility function as a function of the hyperparameter is very volatile and furthermore, it is given implicitly by an optimization problem over the model parameters. To tackle this challenge, we introduce a new technique to characterize the discontinuities and oscillations of the utility function on any fixed problem instance as we vary the hyperparameter; our analysis relies on subtle concepts including tools from algebraic geometry, differential geometry and constrained optimization. We use this to show that the learning-theoretic complexity of the corresponding family of utility functions is bounded. We instantiate our results and provide sample complexity bounds for concrete applications—tuning a hyperparameter that interpolates neural activation functions and setting the kernel parameter in graph neural networks.

## 1  Introduction

Developing deep neural networks that work best for a given application typically corresponds to a tedious selection of hyperparameters and architectures over extremely large search spaces. This process of adapting a deep learning algorithm or model to a new application domain takes up significant engineering and research resources, and often involves unprincipled techniques with limited or no theoretical guarantees on their effectiveness. While the success of pre-trained (foundation) models have shown the usefulness of transferring effective parameters (weights) of learned deep models across tasks [1, 2], it is less clear how to leverage prior experience of "good" hyperparameters to new tasks. In this work, we develop a principled framework for tuning continuous hyperparameters in deep networks by leveraging similar problem instances and obtain sample complexity guarantees for learning provably good hyperparameter values.

39th Conference on Neural Information Processing Systems (NeurIPS 2025).

The vast majority of practitioners still use a naive "grid search" based approach which involves selecting a finite grid of (often continuous-valued) hyperparameters and selecting the one that performs the best. A lot of recent literature has been devoted to automating and improving this hyperparameter tuning process, prominent techniques include Bayesian optimization [3, 4, 5, 6] and random search based methods [7, 8]. While these approaches work well in practice, they either lack a formal basis or enjoy limited theoretical guarantees only under strong assumptions. For example, Bayesian optimization analysis assumes that the performance of the deep network as a function of the hyperparameter can be approximated as a noisy evaluation of an expensive function, typically making assumptions on the form of this noise, and requires setting several hyperparameters and other design choices including the amount of noise, the acquisition function which determines the hyperparameter search space, the type of kernel and its bandwidth parameter. Other techniques, including random search methods and spectral approaches [9] make fewer assumptions but only work for a discrete and finite grid of hyperparameters. This necessitates the need of analyzing the landscape of hyperparameter loss (utility) landscape.

In this work, we consider the problem of tuning neural network hyperparameters in a multi-task setting. We assume there is a fixed but unknown distribution over the learning tasks, and we have access to tasks drawn from that distribution at the training time. We want to learn the hyperparameters to use on future tasks from the same distribution, i.e. on any future task we use the learned hyperparameter, and then we find the best network weights for that task. To address this question, we observe a parallel to prior work on data-driven algorithm design [10, 11]. Despite a similar formulation, our goal is to learn the network hyperparameters, and not configure the training algorithm that learns the network weights. The best weights learned during training for a fixed task ultimately depend on the hyperparameter selected in a complex way, making our setting more challenging than those previously studied under this paradigm.

Mathematically, we aim to analyze the learning-theoretic complexity of a specific function class $\mathcal{U} = \{u_\alpha : \mathcal{X} \to [0, H] \mid \alpha \in \mathcal{A}\}$, where $\mathcal{A} = [\alpha_1, \alpha_2] \subset \mathbb{R}$ is the hyperparameter space and $\mathcal{X}$ is the set of problem instances. We note that each problem instance $\boldsymbol{x} \in \mathcal{X}$ corresponds to a training dataset for a single task (e.g. a dataset of images, as opposed to a single image). We assume there is an application-specific problem problem distribution $\mathcal{D}$ over the datasets (tasks) in $\mathcal{X}$. Each function $u_\alpha(\boldsymbol{x})$ in $\mathcal{U}$ is defined as: $u_\alpha(\boldsymbol{x}) = \sup_{\boldsymbol{w}:(\alpha,\boldsymbol{w})\in\mathcal{A}\times\mathcal{W}} f(\boldsymbol{x}, \alpha, \boldsymbol{w})$, where $u_\alpha(\boldsymbol{x})$ measures the performance of the deep network on a problem instance $\boldsymbol{x}$ and hyperparameter $\alpha$ using the best parameter $\boldsymbol{w}$. Here $\mathcal{W} = [w_{\min}, w_{\max}]^d \subset \mathbb{R}^d$ represents a $d$-dimensional parameter space. Our goal is to establish an upper bound on the learning-theoretic complexity of $\mathcal{U}$, leveraging the structural properties of $f(\boldsymbol{x}, \alpha, \boldsymbol{w})$. We introduce two key notations in our analysis: the parameter-dependent dual function $f_{\boldsymbol{x}}(\alpha, \boldsymbol{w}) := f(\boldsymbol{x}, \alpha, \boldsymbol{w})$, describing the performance corresponding to hyperparameter $\alpha$ and parameter $\boldsymbol{w}$ for a fixed problem instance $\boldsymbol{x}$, and the dual utility function $u_{\boldsymbol{x}}^*(\alpha) := u_\alpha(\boldsymbol{x}) = \max_{\boldsymbol{w}\in\mathcal{W}} f_{\boldsymbol{x}}(\alpha, \boldsymbol{w})$. Inspired by model hyperparameter tuning applications (see Section 5), we assume that the parameter-dependent dual $f_{\boldsymbol{x}}(\alpha, \boldsymbol{w})$ is a the piecewise polynomial function in $\alpha$ and $\boldsymbol{w}$ (see Section 2 for details). Based on this structure of $f_{\boldsymbol{x}}(\alpha, \boldsymbol{w})$, we aim to understand the structure of $u_{\boldsymbol{x}}^*(\alpha)$ which by the earlier work of Balcan et al. [12][1] , would imply learnability of $\mathcal{U} = \{u_\alpha : \mathcal{X} \to [0, H] \mid \alpha \in \mathcal{A}\}$.

The major difficulty we have to overcome is that even if $f_{\boldsymbol{x}}(\alpha, \boldsymbol{w})$ is a nicely structured piecewise polynomial function, the function $u_{\boldsymbol{x}}^*(\alpha) = \sup_{\boldsymbol{w}:(\alpha,\boldsymbol{w})\in\mathcal{A}\times\mathcal{W}} f_{\boldsymbol{x}}(\alpha, \boldsymbol{w})$ appears to be less structured; in particular, the function $u_{\boldsymbol{x}}^*(\alpha)$ is not necessarily piecewise polynomial[2], and might not even have a closed form[3]. Our key technical innovation is to show that for the case of one hyperparameter $\alpha$ we can still find enough structure in the dual functions $u_{\boldsymbol{x}}^*$; specifically by leveraging the structure of parameter-dependent dual functions $f_{\boldsymbol{x}}(\alpha, \boldsymbol{w})$ we show how to bound the number of discontinuities and local maxima of $u_{\boldsymbol{x}}^*$, which in turn implies that they have bounded oscillations (defined in Section 2.1), which then imply a bound on the pseudo-dimension of $\mathcal{U}$ (using results from [12]).

---

[1]This work introduces the relation between the structure of the dual $u_{\boldsymbol{x}}^*$ and the learnability of the primal class $\mathcal{U}$ in a different context of algorithm configuration, but the approach is useful for our setting.

[2]Consider the case where $f_{\boldsymbol{x}}(\alpha, w) = w\alpha - \frac{w^3}{3}$ for $\alpha, w \geq 0$, and define $u_{\boldsymbol{x}}^*(\alpha) = \sup_{w\geq 0} f_{\boldsymbol{x}}(\alpha, w)$. One can easily show that in this case, $u_{\boldsymbol{x}}^*(\alpha) = \frac{2}{3}\alpha^{\frac{3}{2}}$, which is not a polynomial in $\alpha$.

[3]As another example, consider $f_{\boldsymbol{x}}(\alpha, w) = w\alpha^3 + (1/4)w^4 - (3/2)\alpha w^2$ for $\alpha, w \in [0, 1]^2$. In this case, $u_{\boldsymbol{x}}^*(\alpha) = \sup_{w\in[0,1]} f_{\boldsymbol{x}}(\alpha, w)$ has pieces corresponding to $w = 1$ and $\partial f_{\boldsymbol{x}}/\partial w = \alpha^3 + w^3 - 3\alpha w = 0$. The latter is the Folium of Descartes, and does not admit a closed-form expression in terms of elementary functions.

Our proposed framework implies generalization guarantees for data-driven model hyperparameter tuning across various applications. We demonstrate this through two concrete examples including the following. Our first application is to tuning an interpolation hyperparameter for the activation function used at each node of the neural network. Different activation functions perform well on different datasets [13, 14]. We analyze the sample complexity of tuning the best combination from a pair of activation functions by learning a real-valued hyperparameter that interpolates between them. We tune the hyperparameter across multiple problem instances, an important setting for multi-task learning. Our contribution is related to neural architecture search (NAS, [15, 16, 17]). NAS automates the discovery and optimization of neural network architectures, replacing human-led design with computational methods. Several techniques have been proposed [18, 19, 20], but they lack principled theoretical guarantees (see additional related work in Appendix A), and multi-task learning is a known open research direction [21]. We also instantiate our framework for tuning the graph kernel parameter in Graph Neural Networks (GNNs) [22] designed for more effectively deep learning with structured data. Hyperparameter tuning for graph kernels has been studied in the context of classical non-neural models [23, 24], in this work we provide the first provable guarantees for tuning the graph hyperparameter for the more effective modern approach of graph neural networks. While we focus on these specific applications, our proposed framework's generality makes it applicable to many other scenarios.

**Our contributions.** In this work, we provide an analysis for the learnability of parameterized algorithms involving both parameters and hyperparameters, when the learner has access to problem instances drawn from a fixed, unknown distribution. Our analysis captures model hyperparameter tuning in deep networks with a piecewise polynomial parameter-dependent dual function. Concretely,

1. We introduce general tools which connect the number of discontinuities and local maxima of a piecewise continuous function with its learning-theoretic complexity (Section 2.1).
2. We show that when the parameter-dependent dual function $f_{\boldsymbol{x}}(\alpha, \boldsymbol{w})$ computed by a deep network $f$ on a fixed dataset $\boldsymbol{x}$ is *piecewise constant*, the function $u_{\boldsymbol{x}}^*$ is also piecewise constant. This structure occurs in classification tasks with a 0-1 loss objective. We then establish an upper-bound for the pseudo-dimension of $\mathcal{U}$, which translates to learning guarantees for $\mathcal{U}$ (Theorem 3.2).
3. As our main contribution, we prove that when the parameter-dependent dual function $f_{\boldsymbol{x}}(\alpha, \boldsymbol{w})$ exhibits a *piecewise polynomial* structure, under mild regularity assumptions, we can establish an upper bound for the number of discontinuities and local extrema of the dual utility function $u_{\boldsymbol{x}}^*$. We use tools from differential geometry to identify the smooth 1-manifolds in the space $\mathcal{A} \times \mathcal{W}$ corresponding to the derivative curves which determine $u_{\boldsymbol{x}}^*$ and results from algebraic geometry and constrained optimization to bound the number of local extrema of $u_{\boldsymbol{x}}^*$ along such manifolds. We then translate the structure of $u_{\boldsymbol{x}}^*$ to learning guarantees for $\mathcal{U}$ (Theorem 4.1).
4. Finally, we investigate data-driven algorithm configuration problems for deep networks, including tuning interpolation parameters for neural activation functions (Theorem 5.1) and hyperparameter tuning for semi-supervised learning with graph convolutional networks (Theorem H.2). We first uncover the underlying piecewise structure relevant for our framework, and then use this structure to obtain learnability guarantees for tuning the hyperparameters for both classification and regression problems.

**Related work.** Data-driven algorithm design has been successfully applied to tune fundamental algorithms in machine learning and beyond (Appendix A). Our work is the first work to focus on tuning hyperparameters for deep networks using a data-driven lens. A key technical challenge that we overcome is that varying the hyperparameter even slightly can lead to a significantly different learned deep network (even for the same training set) with completely different parameters (weights) which is hard to characterize directly. This is very different from a typical data-driven problem where one is able to show closed forms or precise structural properties for the variation of the learning algorithm's behavior as a function of the hyperparameter [12]. We elaborate further on our technical novelties below. Our theoretical advances are potentially useful beyond deep networks, to algorithms with a tunable hyperparameter and several learned parameters. We discuss the challenges in our setting and explain why developing novel techniques is required. We also provide a brief technical overview of our approach. See Appendix B for further discussion.

**Technical challenges and novelty.** In typical prior work on data-driven hyperparameter tuning [25, 26, 27, 28], the hyperparameter tuning process does not involve the parameter $\boldsymbol{w}$. In such cases, the approach is to show that the utility function $u_{\boldsymbol{x}}^*(\alpha)$ admits a specific piecewise structure in $\alpha$, typically piecewise polynomial or rational. Some works involve the parameter $\boldsymbol{w}$ [29, 30], but the optimal parameter $\boldsymbol{w}_{\boldsymbol{x}}^*(\alpha) \in \arg\max_{\boldsymbol{w} \in \mathcal{W}} f_{\boldsymbol{x}}(\alpha, \boldsymbol{w})$ either has a closed analytical form

or can be approximated by a function $\hat{w}_x(\alpha)$ with bounded error. In contrast, in our setting, $u_x^*(\alpha) = \max_{w \in \mathcal{W}} f_x(\alpha, w)$ is defined via an optimization problem involving the parameter $w$ and no closed form solution for $w_x^*(\alpha)$ is known. This difference highlights the challenges of our setting, overcoming which requires the development of novel techniques.

**Technical overview.** We show in Section 2.1 that if $u_x^*(\alpha)$ is piecewise continuous, we can bound $\mathrm{Pdim}(\mathcal{U})$ via bounding the number of discontinuities and local extrema of $u_x^*(\alpha)$. We then show in Section 3 that if $f_x^*(\alpha, w)$ is piecewise constant, $u_x^*(\alpha)$ is also piecewise constant with finitely many discontinuities. We then present our main technical contribution in Section 4: if $f_x^*(\alpha, w)$ is piecewise polynomial, then we can bound the pseudo-dimension of $\mathcal{U}$. Concretely, we propose the notion of *monotonic curves* and study their relevant properties (Appendix F.3). We show that the domain $\mathcal{A}$ of $u_x^*(\alpha)$ can be partitioned into multiple intervals, over each of which $u_x^*(\alpha)$ is given by the pointwise maximum of $f_x^*(\alpha, w)$ along some fixed set of monotonic curves. Leveraging tools from algebraic and differential geometry, we show that by bounding the number of local maxima of $f_x(\alpha, w)$ along monotonic curves, we can bound the number of discontinuities and local extrema of $u_x^*(\alpha, w)$ in $\mathcal{A}$, thereby establishing learning-theoretic guarantees for $\mathcal{U}$.

## 2 Problem setting and preliminaries

**Models with hyperparameters.** We introduce the data-driven model hyperparameter tuning framework for models with hyperparameters $\alpha$ and trainable parameters $w$. Concretely, let $f : \mathcal{X} \times \mathcal{A} \times \mathcal{W} \to [0, H]$ be the *parameter-dependent utility function* that represents the model's performance, where $f(x, \alpha, w)$ measures the performance corresponding to problem instance $x \in \mathcal{X}$, parameters $w \in \mathcal{W} = [w_{\min}, w_{\max}]^d$, and hyperparameter $\alpha \in [\alpha_{\min}, \alpha_{\max}]$. Here each *problem instance* $x \in \mathcal{X}$ in our hyperparameter tuning framework represents the complete training dataset for a single task, unlike single-task machine learning setting where $x \in \mathcal{X}$ typically represents a single example. For example, in the problem of tuning the interpolation parameter of neural activation functions (Section 5), each problem instance $x = (X, Y)$ consists of a set $X$ of $T$ examples, and the corresponding labels $Y$. We define the *utility function* $u_\alpha(x)$ quantifying the model's performance with hyperparameter $\alpha$ on problem instance $x$ as $u_\alpha(x) = \sup_{w:(\alpha, w) \in \mathcal{A} \times \mathcal{W}} f(x, \alpha, w)$. This formulation can be interpreted as follows: for a given hyperparameter $\alpha$ and problem instance $x$, we determine the optimal model parameters $w$ that maximize performance.

**The dual utility function.** For a fixed problem instance $x \in \mathcal{X}$, we can define the *dual utility function* $u_x^* : \mathcal{A} \to [0, H]$, representing the performance of the network corresponding to the hyperparameter $\alpha$ and the best parameter $w$, as $u_x^*(\alpha) = \sup_{w:(\alpha, w) \in \mathcal{A} \times \mathcal{W}} f_x(\alpha, w)$, where $f_x(\alpha, w) := f(x, \alpha, w)$ is called the *parameter-dependent dual function*.

**Data-driven model hyperparameter tuning problem.** Let $\mathcal{U} = \{u_\alpha : \mathcal{X} \to [0, H] \mid \alpha \in \mathcal{A}\}$ be the *utility function class*. In the data-driven setting, we assume an application-specific problem distribution $\mathcal{D}$ over the set of problem instances $\mathcal{X}$. Here, $\mathcal{D}$ represents a fixed but unknown distribution of over datasets or tasks. Our goal here is find a good hyperparameter $\hat{\alpha}$ given a small sample of tasks drawn from $\mathcal{D}$, that achieves nearly the same utility on average as the optimal hyperparameter $\alpha^* \in \arg\max_{\alpha \in \mathcal{A}} \mathbb{E}_{x \sim \mathcal{D}}[u_\alpha(x)]$. Our approach is to bound the pseudo-dimension and the Rademacher complexity of the function class $\mathcal{U}$, given the structure of the parameter-dependent dual function $f_x(\alpha, w)$.

We consider two situations where $f_x(\alpha, w)$ is either *piecewise constant* (Section 3) or *piecewise polynomial* (Section 4). For the piecewise constant case, we assume that for any $x$, there is a finite partition $\mathcal{P}$ of the domain $\mathcal{A} \times \mathcal{W}$ consisting of connected components such that $f_x(\alpha, w)$ remains constant over each connected component in $\mathcal{P}$. For the second case, we assume that there are algebraic sets $Z_{h_{x,i}} = \{(\alpha, w) \in \mathcal{A} \times \mathcal{W} \mid h_{x,i}(\alpha, w) = 0\}$ partitioning the domain $\mathcal{A} \times \mathcal{W}$, where $h_{x,i}(\alpha, w)$ is a polynomial in $\alpha$ and $w$. In each connected component of the partition, the piece function $f_{x,i}$ which equals $f_x(\alpha, w)$ in the component is also a polynomial of $\alpha$ and $w$.

### 2.1 Discontinuities, local extrema and the pseudo-dimension

In this section, we establish a connection between the structure of the dual utility function $u_x^*$ and the pseudo-dimension of $\mathcal{U} = \{u_\alpha : \mathcal{X} \to \mathbb{R} \mid \alpha \in \mathbb{R}\}$. Concretely, we show that if $u_x^*$ has bounded number of discontinuities and local extrema, then $\mathcal{U}$ has bounded pseudo-dimension.

**Lemma 2.1.** *Consider a real-valued function class $\mathcal{U} = \{u_\alpha : \mathcal{X} \to \mathbb{R} \mid \alpha \in \mathbb{R}\}$, of which each dual function $u_{\boldsymbol{x}}^*(\alpha)$ is piecewise continuous, with at most $B_1$ discontinuities and $B_2$ local maxima. Then $\mathrm{Pdim}(\mathcal{U}) = \mathcal{O}(\log(B_1 + B_2))$.*

The detailed proof of Lemma 2.1 can be found in Appendix D.2. The idea is to show that $u_{\boldsymbol{x}}^*(\alpha)$ has bounded oscillations (Definition 4, Lemma D.2), which is defined as the maximum number of discontinuities that the function $\mathbb{I}_{\{h(\alpha) \geq z\}}$ can have, for any real threshold $z$. We then use a result by Balcan et al. [12] (Theorem D.1) connecting the oscillations and $\mathrm{Pdim}(\mathcal{U})$. Note that Lemma 2.1 does not apply to piecewise constant functions, which have an infinite number of local extrema. However, we can still give an analogous result for this case as follows (see Appendix D.2 for proof).

**Lemma 2.2.** *Consider a real-valued function class $\mathcal{U} = \{u_\alpha : \mathcal{X} \to \mathbb{R} \mid \alpha \in \mathbb{R}\}$, of which each dual function $u_{\boldsymbol{x}}^*(\alpha)$ is piecewise constant with at most $B$ discontinuities. Then $\mathrm{Pdim}(\mathcal{U}) = \mathcal{O}(\log B)$.*

# 3 Piecewise constant parameter-dependent dual function

We first examine the case where $f_{\boldsymbol{x}}(\alpha, \boldsymbol{w})$ is piecewise constant. Concretely, we assume there exists a partition $\mathcal{P}_{\boldsymbol{x}} = \{R_{\boldsymbol{x},1}, \ldots, R_{\boldsymbol{x},N}\}$ of the domain $\mathcal{A} \times \mathcal{W}$ of $f_{\boldsymbol{x}}$, where each $R_{\boldsymbol{x},i}$ is a connected component (Definition 6), and the function $f_{\boldsymbol{x}}(\alpha, \boldsymbol{w})$ restricted on $R_{\boldsymbol{x},i}$ is constant, i.e., $f_{\boldsymbol{x},i}(\alpha, \boldsymbol{w}) = c_{\boldsymbol{x},i}$ for any $(\alpha, \boldsymbol{w}) \in R_{\boldsymbol{x},i}$. Consequently, we can rewrite $u_{\boldsymbol{x}}^*(\alpha)$ as follows:

$$u_{\boldsymbol{x}}^*(\alpha) = \sup_{\boldsymbol{w} \in \mathcal{W}} f_{\boldsymbol{x}}(\alpha, \boldsymbol{w}) = \max_{i \in \{1, \ldots, N\}} \sup_{\boldsymbol{w} : (\alpha, \boldsymbol{w}) \in R_{\boldsymbol{x},i}} f_{\boldsymbol{x},i}(\alpha, \boldsymbol{w}) = \max_{i \in \{1, \ldots, N\} : \exists \boldsymbol{w} \text{ s.t. } (\alpha, \boldsymbol{w}) \in R_{\boldsymbol{x},i}} c_{\boldsymbol{x},i}.$$

We first show Lemma 3.1, which asserts that $u_{\boldsymbol{x}}^*(\alpha)$ is a piecewise constant function and provides an upper bound for the number of discontinuities in $u_{\boldsymbol{x}}^*(\alpha)$.

**Lemma 3.1.** *Assume that the piece functions $f_{\boldsymbol{x},i}(\alpha, \boldsymbol{w}) = c_{\boldsymbol{x},i}$ are constant for all $i \in [N]$ and all problem instances $\boldsymbol{x}$. Then $u_{\boldsymbol{x}}^*(\alpha)$ has $\mathcal{O}(N)$ discontinuity points, partitioning $\mathcal{A}$ into $\mathcal{O}(N)$ intervals. In each interval, $u_{\boldsymbol{x}}^*(\alpha)$ is a constant function.*

*Proof.* The proof idea of Lemma 3.1 is demonstrated in Figure 3. For each connected set $R_{\boldsymbol{x},i}$ corresponding to a piece function $f_{\boldsymbol{x},i}(\alpha, \boldsymbol{w}) = c_{\boldsymbol{x}}, i$, let

$$\alpha_{R_{\boldsymbol{x},i}, \inf} = \inf_\alpha \{\alpha : \exists \boldsymbol{w}, (\alpha, \boldsymbol{w}) \in R_{\boldsymbol{x},i}\}, \quad \alpha_{R_{\boldsymbol{x},i}, \sup} = \sup_\alpha \{\alpha : \exists \boldsymbol{w}, (\alpha, \boldsymbol{w}) \in R_{\boldsymbol{x},i}\}.$$

There are $N$ connected components, and therefore $2N$ such points. Reordering those points and removing duplicate points as $\alpha_{\min} = \alpha_0 < \alpha_1 < \alpha_2 < \cdots < \alpha_t = \alpha_{\max}$, where $t = \mathcal{O}(N)$ we claim that for any interval $I_i = (\alpha_i, \alpha_{i+1})$ where $i = 0, \ldots, t - 1$, the function $u_{\boldsymbol{x}}^*$ remains constant.

Consider any interval $I_i$. By the above construction of $\alpha_i$, for any $\alpha \in I_i$, there exists a *fixed* set of regions $\mathbf{R}_{\boldsymbol{x}, I_i} = \{R_{\boldsymbol{x}, I_i, 1}, \ldots, R_{\boldsymbol{x}, I_i, n}\} \subseteq \mathcal{P}_{\boldsymbol{x}} = \{R_{\boldsymbol{x},1}, \ldots, R_{\boldsymbol{x},N}\}$, such that for any connected set $R \in \mathbf{R}_{\boldsymbol{x}, I_i}$, there exists $\boldsymbol{w}$ such that $(\alpha, \boldsymbol{w}) \in R$. Besides, for any $R \notin \mathbf{R}_{\boldsymbol{x}, I_i}$, there does not exist $\boldsymbol{w}$ such that $(\alpha, \boldsymbol{w}) \in R$. This implies that for any $\alpha \in I_i$, we can write $u_{\boldsymbol{x}}^*(\alpha)$ as

$$u_{\boldsymbol{x}}^*(\alpha) = \sup_{\boldsymbol{w} \in \mathcal{W}} f_{\boldsymbol{x}}(\alpha, \boldsymbol{w}) = \sup_{R \in \mathbf{R}_{\boldsymbol{x}, I_i}} \sup_{\boldsymbol{w} : (\alpha, \boldsymbol{w}) \in R} f_{\boldsymbol{x}}(\alpha, \boldsymbol{w}) = \max_{c \in \boldsymbol{C}_{\boldsymbol{x}, I_i}} c,$$

where $\boldsymbol{C}_{\boldsymbol{x}, I_i} = \{c_{R_{\boldsymbol{x},j}} \mid R_{\boldsymbol{x},j} \in \mathbf{R}_{\boldsymbol{x}, I_i}\}$ contains the constant value that $f_{\boldsymbol{x}}(\alpha, \boldsymbol{w})$ takes over $R$. Since the set $\boldsymbol{C}_{\boldsymbol{x}, I_i}$ is fixed, $u_{\boldsymbol{x}}^*(\alpha)$ remains constant over $I_i$.

Hence, we conclude that over any interval $I_i = (\alpha_i, \alpha_{i+1})$, for $i = 1, \ldots, t - 1$, the function $u_{\boldsymbol{x}}^*(\alpha)$ remains constant. Therefore, there are only the points $\alpha_i$, for $i = 0, \ldots, t - 1$, at which the function $u_{\boldsymbol{x}}^*$ may not be continuous. Since $t = \mathcal{O}(N)$, we have the conclusion. $\square$

By combining Lemma 3.1 and Corollary 2.2, we have the following result, which establishes learning guarantees for the utility function class $\mathcal{U}$ when $f_{\boldsymbol{x}}(\alpha, \boldsymbol{w})$ admits piecewise constant structure.

**Theorem 3.2.** *Consider the utility function class $\mathcal{U} = \{u_\alpha : \mathcal{X} \to [0, H] \mid \alpha \in \mathcal{A}\}$. Assume that for any $\boldsymbol{x} \in \mathcal{X}$ there is a partition $\mathcal{P}_{\boldsymbol{x}} = \{R_{\boldsymbol{x},1}, \ldots, R_{\boldsymbol{x},N}\}$ of $\mathcal{A} \times \mathcal{W}$, over each of which $f_{\boldsymbol{x},i}$ remains constant. Then for any distribution $\mathcal{D}$ over $\mathcal{X}$, and any $\delta \in (0, 1)$, with probability at least $1 - \delta$ over the draw of $S = \{\boldsymbol{x}_1, \ldots, \boldsymbol{x}_m\} \sim \mathcal{D}^m$, we have that*

$$|\mathbb{E}_{\boldsymbol{x} \sim \mathcal{D}}[u_{\hat{\alpha}_S}(\boldsymbol{x})] - \mathbb{E}_{\boldsymbol{x} \sim \mathcal{D}}[u_{\alpha^*}(\boldsymbol{x})]| = \mathcal{O}\left(\sqrt{\frac{1}{m} \log(N/\delta)}\right).$$

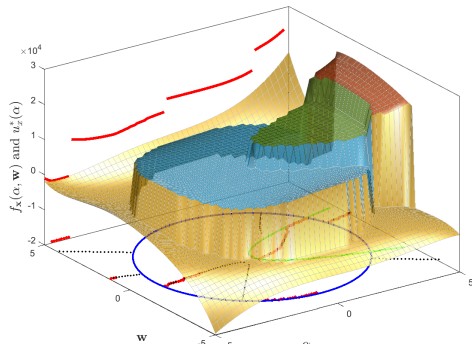
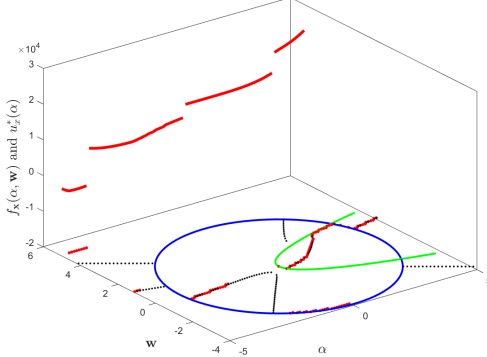

(a) The piecewise structure of $u_{\boldsymbol{x}}^*(\alpha)$ and piecewise polynomial surface of $f_{\boldsymbol{x}}(\alpha, \boldsymbol{w})$ in sheer view.

(b) Removing the surface $f_{\boldsymbol{x}}(\alpha, \boldsymbol{w})$ for better view of $u_{\boldsymbol{x}}^*(\alpha)$, the boundaries, and the derivative curves.

Figure 1: A demonstration of the proof idea for Theorem 4.1 in 2D ($\boldsymbol{w} \in \mathbb{R}$). Here, the domain of $f_{\boldsymbol{x}}^*(\alpha, \boldsymbol{w})$ is partitioned into four regions by two boundaries: a circle (blue line) and a parabola (green line). In each region $i$, the function $f_{\boldsymbol{x}}(\alpha, \boldsymbol{w})$ is a polynomial $f_{\boldsymbol{x},i}(\alpha, \boldsymbol{w})$, of which the derivative curve $\frac{\partial f_{\boldsymbol{x},i}}{\partial \boldsymbol{w}} = 0$ is demonstrated by the black dot in the plane of $(\alpha, \boldsymbol{w})$. The value of $u_{\boldsymbol{x}}^*(\alpha)$ is demonstrated in the red line, and the red dots in the plane $(\alpha, \boldsymbol{w})$ corresponds to the position where $f_{\boldsymbol{x}}(\alpha, \boldsymbol{w}) = u_{\boldsymbol{x}}^*(\alpha)$. We can see that it occurs in either the derivative curves or in the boundary. Our goal is to leverage this property to control the number of discontinuities and local maxima of $u_{\boldsymbol{x}}^*(\alpha)$, which can be converted to the generalization guarantee of the utility function class $\mathcal{U}$.

*Here, $\hat{\alpha}_S \in \arg\min_{\alpha \in \mathcal{A}} \frac{1}{m} \sum_{i=1}^{m} u_\alpha(\boldsymbol{x}_i)$, and $\alpha^* \in \max_{\alpha \in \mathcal{A}} \mathbb{E}_{\boldsymbol{x} \sim \mathcal{D}}[u_\alpha(\boldsymbol{x})]$.*

*Proof.* From Lemma 3.1, we know that any dual utility function $u_{\boldsymbol{x}}^*$ is piecewise constant and has at most $\mathcal{O}(N)$ discontinuities. Combining with Corollary 2.2, we conclude that $\text{Pdim}(\mathcal{U}) = \mathcal{O}(\log(N))$. Finally, a standard result from learning theory (Theorem C.1) gives us the final guarantee. $\qquad\square$

In many applications, the partition of $f_{\boldsymbol{x}}(\alpha, \boldsymbol{w})$ into connected components is typically defined by $M$ boundary functions $h_{\boldsymbol{x},1}(\alpha, \boldsymbol{w}), \ldots, h_{\boldsymbol{x},M}(\alpha, \boldsymbol{w})$. When the boundary functions are polynomials with bounded degree, we have the following concrete result for such a case, which is a direct consequence of Theorem 3.2.

**Corollary 3.3.** *Consider the utility function class $\mathcal{U} = \{u_\alpha : \mathcal{X} \to [0, H] \mid \alpha \in \mathcal{A}\}$. Assuming that for any $\boldsymbol{x}$, there are $M$ boundary functions $h_{\boldsymbol{x},1}, \ldots, h_{\boldsymbol{x},M}$ partitioning the domain $\mathcal{A} \times \mathcal{W}$ into $\mathcal{P}_{\boldsymbol{x}} = \{R_{\boldsymbol{x},1}, \ldots, R_{\boldsymbol{x},N}\}$, over each of which $f_{\boldsymbol{x},i}$ remains constant. Assuming that the boundary functions $h_{\boldsymbol{x},i}$, where $i = 1, \ldots, M$, are polynomials in $\alpha, \boldsymbol{w}$ of degree at most $\Delta$. Then for any distribution $\mathcal{D}$ over $\mathcal{X}$, and any $\delta \in (0, 1)$, with probability at least $1 - \delta$ over the draw of $S = \{\boldsymbol{x}_1, \ldots, \boldsymbol{x}_m\} \sim \mathcal{D}^m$, we have that*

$$\left| \mathbb{E}_{\boldsymbol{x} \sim \mathcal{D}}[u_{\hat{\alpha}_S}(\boldsymbol{x})] - \mathbb{E}_{\boldsymbol{x} \sim \mathcal{D}}[u_{\alpha^*}(\boldsymbol{x})] \right| = \mathcal{O}\left( \sqrt{\frac{d \log(M\Delta) + \log(1/\delta)}{m}} \right)$$

*Proof.* From Theorem F.9, there are at most $N \leq \mathcal{O}\left( \frac{M\Delta}{d+1} \right)^{d+1}$ connected components created by $M$ polynomial boundary functions in $\boldsymbol{x}$ and $\alpha$ of degree at most $\Delta$. Combining with Theorem 3.2, we have the final conclusion. $\qquad\square$

**Remark 1.** *Theorem 3.2 can be applied beyond polynomial boundaries, for example, settings where the boundary functions consist of Pfaffian functions (which is a generalization of polynomial functions that includes exponential and logarithmic functions). See Appendix J for a detailed discussion.*

# 4 Piecewise polynomial parameter-dependent dual function

We examine the case where $f_{\boldsymbol{x}}(\alpha, \boldsymbol{w})$ exhibits a piecewise polynomial structure. Concretely, the domain $\mathcal{A} \times \mathcal{W} = [\alpha_{\min}, \alpha_{\max}] \times [w_{\min}, w_{\max}]^d$ is partitioned into $N$ connected components by $M$ algebraic sets $Z_{h_{\boldsymbol{x},j}} = \{(\alpha, \boldsymbol{w}) \in \mathcal{A} \times \mathcal{W} \mid h_{\boldsymbol{x},j}(\alpha, \boldsymbol{w}) = 0\}$, for $j = 1, \ldots, M$. Here, each *boundary function* $h_{\boldsymbol{x},j}(\alpha, \boldsymbol{w})$ is a polynomial in $\alpha$ and $\boldsymbol{w}$ of degree at most $\Delta_b$. The resulting partition $\mathcal{P}_{\boldsymbol{x}} = \{R_{\boldsymbol{x},1}, \ldots, R_{\boldsymbol{x},N}\}$ consists of connected components $R_{\boldsymbol{x},i}$, each formed by a connected component of $\mathcal{A} \times \mathcal{W} - \cup_{j=1}^{M} Z_{h_{\boldsymbol{x},j}}$ and its adjacent boundaries (Definition 8). Within each connected component $R_{\boldsymbol{x},i}$, the function $f_{\boldsymbol{x}}(\alpha, \boldsymbol{w})$ is a polynomial $f_{\boldsymbol{x},i}(\alpha, \boldsymbol{w})$ of degree at most $\Delta_p$. We call $f_{\boldsymbol{x},i}$ the *piece function*. The dual utility function $u_{\boldsymbol{x}}^*(\alpha)$ can then be written as:

$$u_{\boldsymbol{x}}^*(\alpha) = \sup_{\boldsymbol{w}:(\alpha,\boldsymbol{w}) \in \mathcal{A} \times \mathcal{W}} f_{\boldsymbol{x}}(\alpha, \boldsymbol{w}) = \max_{i \in \{1,\ldots,N\}} \sup_{\boldsymbol{w}:(\alpha,\boldsymbol{w}) \in R_{\boldsymbol{x},i}} f_{\boldsymbol{x},i}(\alpha, \boldsymbol{w}) = \max_{i \in \{1,\ldots,N\}} u_{\boldsymbol{x},i}^*(\alpha),$$

where $u_{\boldsymbol{x},i}^*(\alpha) = \sup_{\boldsymbol{w}:(\alpha,\boldsymbol{w}) \in R_{\boldsymbol{x},i}} f_{\boldsymbol{x},i}(\alpha, \boldsymbol{w})$.

For a better understanding of the proof techniques used for the main results (Theorem 4.1), we encourage the reader to consider the simpler case where $d = 1$ (see Appendix F.4). Figure 1 demonstrates the piecewise structure of the dual function in this case. We can see that for any $\alpha$ the value of the dual $u_{\boldsymbol{x}}^*(\alpha)$ is given by the value of $f_{\boldsymbol{x}}(\alpha, \boldsymbol{w})$ on some point $(\tilde{\alpha}, \tilde{\boldsymbol{w}})$ that lies either on some derivative curve or on one of the boundaries. Our proof obtains a bound on the number of all such points $(\tilde{\alpha}, \tilde{\boldsymbol{w}})$ that are "critical", i.e. where a switch may occur between boundaries and derivative curves (or between pairs of boundaries, or pairs of derivative curves). Between such critical points, we use the Lagrange's multiplier theorem (Theorem F.3) to express the locations of the local extrema of $u_{\boldsymbol{x}}^*$ as intersections of algebraic varieties. In summary, we use the above arguments to bound the number of discontinuities and local maxima of $u_{\boldsymbol{x}}^*(\alpha)$, which can be converted to the generalization guarantee of the utility function class $\mathcal{U}$.

Our main result in this section applies to the general multidimensional case, $\boldsymbol{w} \in \mathbb{R}^d$. While the overall proof strategy is similar, it needs significant extensions and refinements (Remark 4). We begin with the following regularity assumption on the piece and boundary functions $f_{\boldsymbol{x},i}$ and $h_{\boldsymbol{x},j}$.

**Assumption 1.** *Given any dual utility functions $u_{\boldsymbol{x}}^*(\alpha)$ with corresponding boundary functions $h_{\boldsymbol{x},j}$, for $j = 1, \ldots, M$, and piece functions $f_{\boldsymbol{x},i}$, for $i = 1, \ldots, N$, satisfying the following property: for any set of $S$ boundary functions $h_1, \ldots, h_S$ chosen from the set of boundaries $\{h_{\boldsymbol{x},1}, \ldots, h_{\boldsymbol{x},M}\}$, and any piece function $f$ chosen from the set of piece functions $\{f_{\boldsymbol{x},1}, \ldots, f_{\boldsymbol{x},N}\}$, we assume*

1. *Extended linearly independent constraints qualification (ELICQ): The Jacobian $J_{\boldsymbol{h}}(\alpha, \boldsymbol{w})$ has full row rank when evaluated at $(\alpha, \boldsymbol{w}) \in \boldsymbol{h}^{-1}(\boldsymbol{0})$. Moreover, if $S \leq d$, we assume that $J_{\boldsymbol{h}}(\boldsymbol{w})$ has full row rank hen evaluated at $(\alpha, \boldsymbol{w}) \in \boldsymbol{h}^{-1}(\boldsymbol{0})$.*

2. *Non-degeneracy (ND): the polynomial function $P(\alpha, \boldsymbol{w}, \boldsymbol{\lambda}) = \det(J_{(\boldsymbol{h}, \nabla_{\boldsymbol{w}} L)}(\boldsymbol{w}, \boldsymbol{\lambda}))$ is not identically zero on any $(d+1)$-dimensional irreducible component of the real algebraic set $Z_{\boldsymbol{h}} = \{(\alpha, \boldsymbol{w}, \boldsymbol{\lambda}) \in \mathbb{R}^{S+d+1} \mid \boldsymbol{h}(\alpha, \boldsymbol{w}, \boldsymbol{\lambda}) = \boldsymbol{0}\}$. Here, $L(\alpha, \boldsymbol{w}, \boldsymbol{\lambda}) = f(\alpha, \boldsymbol{w}) + \boldsymbol{\lambda}^{\top} \boldsymbol{h}(\alpha, \boldsymbol{w})$.*

**Remark 2.** *1. The ELICQ assumption consists of two parts. The LICQ part is a standard assumption in parametric optimization literature (or in constraint optimization in general, see e.g., [31, 32]) and corresponds to assuming that $J_{\boldsymbol{h}}(\boldsymbol{w})$ has full row rank when evaluated at $(\alpha, \boldsymbol{w}) \in \boldsymbol{h}^{-1}(\boldsymbol{0})$, for $S \leq d$. The second part, assuming that the Jacobian $J_{\boldsymbol{h}}(\alpha, \boldsymbol{w})$ has full row rank when evaluated at $(\alpha, \boldsymbol{w}) \in h^{-1}(\boldsymbol{0})$, essentially says that $\boldsymbol{0}$ is a regular value of $\boldsymbol{h}$. This assumption is directly implied by LICQ when $S \leq d$. For $S \geq d+1$ (where LICQ is inapplicable as $\mathsf{rank}(J_{\boldsymbol{h}}(\boldsymbol{w})) \leq d$) this assumption implies that the solutions that satisfy $\boldsymbol{h}(\alpha, \boldsymbol{w}) = \boldsymbol{0}$ are isolated points for $S = d+1$, or the solution set is empty for $S > d+1$. We note that this assumption is relatively mild since Sard's theorem (Theorem F.15) asserts that the set of a regular value of a smooth map ($\boldsymbol{h}$ in this case) has Lebesgue measure $\boldsymbol{0}$ in the co-domain.*

2. *The ND assumption is also relatively mild. Roughly speaking, it requires that there is at least one point $(\alpha, \boldsymbol{w}, \boldsymbol{\lambda})$ satisfying $\boldsymbol{h}(\alpha, \boldsymbol{w}) = 0$, where $P(\alpha, \boldsymbol{w}, \boldsymbol{\lambda})$ is non-zero. Intuitively, we use this assumption to show that the intersection of derivative curves with boundaries can be decomposed into a bounded number of monotonic curves (Definition 19), which we show to have favorable structure (Lemma F.19), playing an important role in our analysis.*

We now state our main structural result, and provide a brief proof sketch (full proof in Appendix F.5).

**Theorem 4.1.** *Consider the utility function class $\mathcal{U} = \{u_\alpha : \mathcal{X} \to [0, H] \mid \alpha \in [\alpha_{\min}, \alpha_{\max}]\}$. Assume that $f_{\boldsymbol{x}}(\alpha, \boldsymbol{w})$ admits piecewise polynomial structure with*

$N$ piece functions $f_{\boldsymbol{x},i}$ (for $i = 1,\ldots,N$) and $M$ boundaries functions $h_{\boldsymbol{x},j}$ (for $j = 1,\ldots,M$) satisfying Assumption 1. Then for any distribution $\mathcal{D}$ over $\mathcal{X}$, for any $\delta \in (0,1)$, w.p. at least $1 - \delta$ over the draw of $S = \{\boldsymbol{x}_1,\ldots,\boldsymbol{x}_m\} \sim \mathcal{D}^m$, we have $|\mathbb{E}_{\boldsymbol{x}\sim\mathcal{D}}[u_{\hat{\alpha}_{ERM}}(\boldsymbol{x})] - \mathbb{E}_{\boldsymbol{x}\sim\mathcal{D}}[u_{\alpha^*}(\boldsymbol{x})]| = \mathcal{O}\left(\sqrt{\frac{\log N + d\log(\Delta M) + \log(1/\delta)}{m}}\right)$. Here, $\hat{\alpha}_S \in \arg\min_{\alpha\in\mathcal{A}} \frac{1}{m}\sum_{i=1}^{m} u_\alpha(\boldsymbol{x}_i)$, $\alpha^* \in \max_{\alpha\in\mathcal{A}} \mathbb{E}_{\boldsymbol{x}\sim\mathcal{D}}[u_\alpha(\boldsymbol{x})]$, and $\Delta = \max\{\Delta_p, \Delta_d\}$ is the maximum degree of piece functions $f_{\boldsymbol{x},i}$ and boundaries $h_{\boldsymbol{x},j}$.

*Proof Sketch.* The proof consists of three main steps. **Step 1**: bound the number of possible discontinuities and local maxima of $v_{\boldsymbol{x}}^*(\alpha)$ under Assumption 3, a stronger assumption compared to Assumption 1. **Step 2**: we show that, for any $u_{\boldsymbol{x}}^*(\alpha)$ that satisfies Assumption 1, we can construct $v_{\boldsymbol{x}}^*(\alpha)$ satisfies Assumption 3 and is arbitrarily close to $u_{\boldsymbol{x}}^*(\alpha)$. **Step 3**: we can use this property to recover the learning guarantee for $\mathcal{U}$. The key ideas of each step are sketched below.

1. We show that if the piece functions $f_{\boldsymbol{x},i}$ and boundaries $h_{\boldsymbol{x},i}$ satisfy a stronger assumption (Assumption 3), we can bound the pseudo-dimension of $\mathcal{U}$ (Theorem F.25). The steps are:

(a) From Lemma 2.1, we know that it suffices to bound the number of discontinuities and local maxima of $u_{\boldsymbol{x}}^*$. From Lemma F.1 and Proposition F.2, we can bound the number of discontinuities and local maxima of $u_{\boldsymbol{x}}^*$ by bounding those of $u_{\boldsymbol{x},i}^*$ for any piece $i$. Hence, in the next few steps, our main object of study is $u_{\boldsymbol{x},i}^*$.

(b) We first demonstrate that the hyperparameter domain $\mathcal{A}$ can be partitioned into intervals using the set of points $\mathcal{A}_1 \subset \mathcal{A}$ that has at most $\mathcal{O}\left((2\Delta)^{d+1}\left(\frac{eM}{d+1}\right)^{d+1}\right)$ elements. For each interval $I_t$, there exists a set of subsets of boundaries $\mathbf{S}_{\boldsymbol{x},t}^1 \subset 2^{\mathbf{H}_{\boldsymbol{x},i}}$ such that for any set of boundaries $\mathcal{S} = \{h_{\mathcal{S},1},\ldots,h_{\mathcal{S},S}\} \in \mathbf{S}_{\boldsymbol{x},t}^1$, the intersection of boundaries $\{(\alpha,\boldsymbol{w}) \mid h(\alpha,\boldsymbol{w}) = 0, h \in \mathcal{S}\}$ in $\mathcal{S}$ contains a feasible point $(\alpha,\boldsymbol{w})$ for any $\alpha$ in that interval. The key idea of this step is to upper bound the number of $\alpha$-extreme points (Definition 7) of connected components of such intersections, using Warren's theorem (Lemma F.8). In words, the goal of this step is to partition the hyperparameter space $\mathcal{A}$ into intervals, that over any interval $I_t$ in the partition, there is some set of set of boundaries $\mathbf{S}_{\boldsymbol{x},t}^1$ such that for any $\alpha' \in I_t$, there is at least one point that lies in the intersection of set of boundaries $\mathcal{S} \in \mathbf{S}_{\boldsymbol{x},t}^1$ that has $\alpha$-coordinate equal to $\alpha'$.

(c) Now note that, for any $\boldsymbol{w}_\alpha$ that maximizes $f_{\boldsymbol{x},i}(\alpha,\boldsymbol{w})$ for any fixed $\alpha$, due to the Lagrangian multipliers theorem (Theorem F.3), there is a set of boundaries $\mathcal{S} = \{h_{\mathcal{S},1},\ldots,h_{\mathcal{S},S}\}$ such that $\boldsymbol{h}_{\mathcal{S}}(\alpha,\boldsymbol{w}_\alpha) = \boldsymbol{0}$, $\nabla_{\boldsymbol{w}} L_{\mathcal{S}}(\alpha,\boldsymbol{w}_\alpha,\boldsymbol{\lambda}_\alpha) = \boldsymbol{0}$, for some $\boldsymbol{\lambda}_\alpha$. Here, $\boldsymbol{h}_{\mathcal{S}} = (h_{\mathcal{S},1},\ldots,h_{\mathcal{S},S})$ is the polynomial mapping constructed by the boundary functions corresponding to some set $\mathcal{S}$ of boundaries, $L_{\mathcal{S}}(\alpha,\boldsymbol{w},\boldsymbol{\lambda}) = f_{\boldsymbol{x},i}(\alpha,\boldsymbol{w}) + \boldsymbol{\lambda}^\top \boldsymbol{h}_{\mathcal{S}}(\alpha,\boldsymbol{w})$ is the corresponding Lagrangian function. Next, our goal is to decompose the solution set of the above Lagrangian into a union of monotonic curves (Definition 19), which we show to have the following property: for any $\alpha \in I_t$, there is at most one point $(\alpha,\boldsymbol{w},\boldsymbol{\lambda})$ in the monotonic curve $C$ (Lemma F.19).

Towards this goal, we refine the partition of $\mathcal{A}$ into intervals, using the set of points $\mathcal{A}_2$ that contains points in $\mathcal{A}_1$ and some additional points in set $\mathcal{B}$, for a total of $\mathcal{O}\left((2\Delta)^{2d+1}\left(\frac{eM}{d}\right)^d + \Delta^{4d}\left(\frac{eM}{d}\right)^d\right)$ elements. Here, $\mathcal{B}$ is the set of $\alpha$-coordinates of the set of points in $\mathcal{B}'$, which contains the points $(\alpha,\boldsymbol{w},\boldsymbol{\lambda})$ that satisfy

$$\boldsymbol{h}_{\mathcal{S}}(\alpha,\boldsymbol{w}) = \boldsymbol{0}, \quad \nabla_{\boldsymbol{w}} L_{\mathcal{S}}(\alpha,\boldsymbol{w},\boldsymbol{\lambda}) = \boldsymbol{0}, \quad \det(J_{\overline{\boldsymbol{k}}_{\mathcal{S}}}(\boldsymbol{w},\boldsymbol{\lambda})) = 0.$$

where $J_{\overline{\boldsymbol{k}}_{\mathcal{S}}}(\boldsymbol{w},\boldsymbol{\lambda})$ is the Jacobian of $\overline{\boldsymbol{k}}_{\mathcal{S}} = (\boldsymbol{h}_{\mathcal{S}}, \nabla_{\boldsymbol{w}} L_{\mathcal{S}})$ with respect to $(\boldsymbol{w},\boldsymbol{\lambda})$. Under Assumption 3.2, the number of such points is bounded, leading to the number of elements in $\mathcal{B}$ being bounded. In any interval $I_t$ in the partition of $\mathcal{A}$ created by the set of points $\mathcal{A}_2$, we claim that the set $Z_{\overline{\boldsymbol{k}}_{\mathcal{S}}} = \{(\alpha,\boldsymbol{w},\boldsymbol{\lambda}) \mid \overline{\boldsymbol{k}}_{\mathcal{S}}(\alpha,\boldsymbol{w},\boldsymbol{\lambda}) = \boldsymbol{0}\}$ defines a smooth one-dimensional manifold in the space of $(\alpha,\boldsymbol{w},\boldsymbol{\lambda})$.

(d) Next, we want to partition the hyperparameter space $\alpha$ into intervals, such that in any interval, the function $u_{\boldsymbol{x},i}^*$ can be written as the point-wise maximum of the piece function $f_{\boldsymbol{x},i}(\alpha,\boldsymbol{w})$ along some fixed set of monotonic curves. Concretely, we further refine the partition of $\mathcal{A}$ into intervals, using the set of points $\mathcal{A}_3 \subset \mathcal{A}$ that contains the points in $\mathcal{A}_2$ and some extra points in the set $\mathcal{D}$. Here, $\mathcal{D}$ contains the $\alpha$-extreme points of connected components of the algebraic set $Z_{\overline{\boldsymbol{k}}_{\mathcal{S}}}$ (where $\overline{\boldsymbol{k}}_{\mathcal{S}} = (\boldsymbol{h}_{\mathcal{S}}, \nabla_{\boldsymbol{w}} L_{\mathcal{S}})$ as defined in (c)) and the $\alpha$-coordinate of the intersections between $Z_{\overline{\boldsymbol{k}}_{\mathcal{S}}}$ with

another boundary $h' \notin \mathcal{S}$, for some fixed set of the boundaries $\mathcal{S}$. In any interval $I_t$, there is a fixed set $\mathcal{C}_t$ of monotonic curves such that for any $\alpha \in \mathcal{I}$, the function $u^*_{\boldsymbol{x},i}$ can be written as $u^*_{\boldsymbol{x},i}(\alpha') = \max_{C \in \mathcal{C}_t} g_C(\alpha')$, where $g_C(\alpha') = f_{\boldsymbol{x},i}(\alpha', \boldsymbol{w}_{\alpha'})$, and $(\alpha', \boldsymbol{w}_{\alpha'}, \boldsymbol{\lambda}_{\alpha'})$ is the unique point in monotonic curve $C$ that has $\alpha$-coordinate equal to $\alpha'$ (a property of monotonic curve, see Lemma F.19). Therefore, over this interval, $u^*_{\boldsymbol{x},i}(\alpha)$ is continuous, since each $g_C(\alpha')$ is continuous (Lemma F.1).

(e) Moreover, we show that in each interval $I_t$, any local maximum of $u^*_{\boldsymbol{x},i}(\alpha)$ is a local maximum of $f_{\boldsymbol{x},i}(\alpha, \boldsymbol{w})$ along a monotonic curve (Lemma F.21). Again, we can control the number of such points using Bezout's theorem (Corollary F.10) and Assumption 3.3. Finally, we put together all the potential discontinuities and local extrema of $u^*_{\boldsymbol{x},i}$. Combining with Lemma 2.1 we have the upper-bound for $\text{Pdim}(\mathcal{U})$ (Theorem F.26).

2. We show that for any function class $\mathcal{U}$ whose dual functions $u^*_{\boldsymbol{x}}$ have piece functions and boundaries satisfying Assumption 1, we can construct a new function class $\mathcal{V}$. The dual functions $v^*_{\boldsymbol{x}}$ of $\mathcal{V}$ have piece functions and boundaries that satisfy Assumption 3. Moreover, we show that $\|u^*_{\boldsymbol{x}} - v^*_{\boldsymbol{x}}\|_\infty$ can be made arbitrarily small. See Lemma F.30 for a proof overview as well as the detailed proof.

3. Finally, using the results from Step (1), we establish an upper bound on the pseudo-dimension for the function class $\mathcal{V}$ described in Step (2). The result can then be used to determine the learning-theoretic complexity of $\mathcal{U}$ by applying Lemma C.3 and Lemma C.4. Standard learning theory literature then allows us to translate the learning-theoretic complexity of $\mathcal{U}$ into its learning guarantee. This final step is detailed in Appendix F.5. $\qquad\square$

### 4.1 Relaxation to approximate-ERM oracle

The preceding analysis implicitly assumes access to an *exact-ERM (e-ERM) oracle* in defining the dual utility function. In practice, however, such an exact oracle is unavailable; instead, we typically have access to an *approximate-ERM (a-ERM) oracle* (for example, the weights learned using gradient descent based optimization) which we formalize below.

**Assumption 2** ($\xi$-a-ERM oracle and its induced function class). *Consider a utility function class $\mathcal{U} = \{u_\alpha : \mathcal{X} \to [0, H] \mid \alpha \in \mathcal{A}\}$. Assume that we access to a $\xi$-a-ERM oracle such that given a problem instance $\boldsymbol{x}$ and a hyperparameter $\alpha$, return $\tilde{\boldsymbol{w}}_\alpha$ such that $|u^*_{\boldsymbol{x}}(\alpha) - v^*_{\boldsymbol{x}}(\alpha)| < \xi$, where $u^*_{\boldsymbol{x}}(\alpha) = \sup_{\boldsymbol{w}:(\alpha,\boldsymbol{w}) \in \mathcal{A} \times \mathcal{W}} f_{\boldsymbol{x}}(\alpha, \boldsymbol{w})$, and $v^*_{\boldsymbol{x}}(\alpha) = f_{\boldsymbol{x}}(\alpha, \tilde{\boldsymbol{w}}_\alpha)$. We further define $\mathcal{V} = \{v_\alpha : \mathcal{X} \to [0, H] \mid \alpha \in \mathcal{A}\}$ the induced function class by the $\xi$-a-ERM oracle, where $v_\alpha(\boldsymbol{x}) := v^*_{\boldsymbol{x}}(\alpha)$.*

The below gives learning guarantees for the induced function class $\mathcal{V}$ (proof in Appendix G).

**Theorem 4.2.** *Assuming $\mathcal{U} = \{u_\alpha : \mathcal{X} \to [0, H] \mid \alpha \in [\alpha_{\min}, \alpha_{\max}]\}$ satisfying Assumption 1. Let $\mathcal{V} = \{v_\alpha : \mathcal{X} \to [0, H] \mid \alpha \in \mathcal{A}\}$ be the induced function class by $\xi$-a-ERM oracle. Then for any distribution $\mathcal{D}$ over $\mathcal{X}$, for any $\delta \in (0, 1)$, w.p. at least $1 - \delta$ over the draw of $S = \{\boldsymbol{x}_1, \dots, \boldsymbol{x}_m\} \sim \mathcal{D}^m$, we have $|\mathbb{E}_{\boldsymbol{x} \sim \mathcal{D}}[v_{\hat{\alpha}_{ERM}}(\boldsymbol{x})] - \mathbb{E}_{\boldsymbol{x} \sim \mathcal{D}}[v_{\alpha^*}(\boldsymbol{x})]| = \mathcal{O}\left( \sqrt{\frac{\log N + d \log(\Delta M) + \log(1/\delta)}{m}} + \xi \right).$*

## 5 Applications

We demonstrate the application of our results to two specific hyperparameter tuning problems in deep learning. We note that the problem might be presented as analyzing a loss function class $\mathcal{L} = \{\ell_\alpha : \mathcal{X} \to [0, H] \mid \alpha \in \mathcal{A}\}$ instead of utility function class $\mathcal{U} = \{u_\alpha : \mathcal{X} \to [0, H] \mid \alpha \in \mathcal{A}\}$, but our results still hold, just by defining $u_\alpha(\boldsymbol{x}) = H - \ell_\alpha(\boldsymbol{x})$. We establish bounds on the complexity of tuning the linear interpolation hyperparameter for activation functions, which is motivated by DARTS [14], and exploring the tuning of graph kernel parameters in Graph Neural Networks (GNNs), in both regression and classification problems. We present below the guarantee for tuning interpolation hyperparameter of neural activation functions here (details and other applications in Appendix H).

We consider a feed-forward neural network $f$ with $L$ layers, $W_i$ parameters in the $i^{th}$ layer, and $W = \sum_{i=1}^L W_i$ is the total number of parameters. We denote by $k_i$ the number of computational nodes in layer $i$, and let $k = \sum_{i=1}^L k_i$. At each node, we choose between two piecewise polynomial activation functions, $o_1 : \mathbb{R} \to \mathbb{R}$ and $o_2 : \mathbb{R} \to \mathbb{R}$. The domain of the activation functions is partitioned by finitely many *breakpoints*, such that the activation function is polynomial in each partition. For example, 0 is a breakpoint of the ReLU activation function. Liu et al. [14] propose a simple method for selecting activation functions: during training, they define a general activation function $\sigma$ as a weighted

combination of $o_1$ and $o_2$. While their framework is more general, allowing for multiple activation functions and layer-specific activation, we analyze a simplified version. The combined activation function is given by $\sigma(x) = \alpha o_1(x) + (1 - \alpha) o_2(x)$, where $\alpha \in [0, 1]$ is the interpolation hyperparameter. This framework can express functions like the parametric ReLU, $\sigma(z) = \max\{0, z\} + \alpha \min\{0, z\}$, which empirically outperforms the regular ReLU which corresponds to $\alpha = 0$ [33].

In parametric regression, the final layer output is $g(\alpha, \boldsymbol{w}, \boldsymbol{x}) = \hat{y} \in \mathbb{R}^D$, where $\boldsymbol{w} \in \mathcal{W} \subset \mathbb{R}^W$ is the parameter vector and $\alpha$ is the architecture hyperparameter. The loss for a single example $(x, y)$ is $\|g(\alpha, \boldsymbol{w}, x) - y\|^2$, and for $T$ examples, we define

$$\ell_\alpha((X, Y)) = \min_{\boldsymbol{w} \in \mathcal{W}} \frac{1}{T} \sum_{(x,y) \in (X,Y)} \|g(\alpha, \boldsymbol{w}, x) - y\|^2 = \min_{\boldsymbol{w} \in \mathcal{W}} f((X, Y), \alpha, \boldsymbol{w}).$$

With $\mathcal{X}$ as the space of $T$-example sets, we define the loss function class $\mathcal{L}^{\mathrm{AF}} = \{\ell_\alpha : \mathcal{X} \to \mathbb{R} \mid \alpha \in [\alpha_{\min}, \alpha_{\max}]\}$. We aim to provide a learning-theoretic guarantee for $\mathcal{L}^{\mathrm{AF}}$. Given a problem instance $(X, Y)$, the key idea is to establish a piecewise polynomial structure for the parameter-dependent dual function $f_{(X,Y)}(\alpha, \boldsymbol{w})$, and then apply our main result Theorem 4.1. We establish this structure by extending the inductive argument due to Bartlett et al. [34] who establish the piecewise polynomial structure of the neural network output as a function of just the parameters $\boldsymbol{w}$ (i.e. when there are no hyperparameters) on any fixed collection of input examples.

**Theorem 5.1.** *Let $\mathcal{L}^{AF}$ denote loss function class defined above, with activation functions $o_1, o_2$ having maximum degree $\Delta$ and maximum breakpoints $p$. Given a problem instance $\boldsymbol{x} = (X, Y)$, the dual loss function is defined as $\ell_{\boldsymbol{x}}^*(\alpha) := \min_{\boldsymbol{w} \in \mathcal{W}} f(\boldsymbol{x}, \alpha, \boldsymbol{w}) = \min_{w \in \mathcal{W}} f_{\boldsymbol{x}}(\alpha, \boldsymbol{w})$. Then, $f_{\boldsymbol{x}}(\alpha, \boldsymbol{w})$ admits piecewise polynomial structure with bounded pieces and boundaries. Further, if the piecewise structure of $f_{\boldsymbol{x}}(\alpha, \boldsymbol{w})$ satisfies Assumption 1, then for any $\delta \in (0, 1)$, w.p. at least $1 - \delta$ over the draw of problem instances $S = \{\boldsymbol{x}_1, \ldots, \boldsymbol{x}_m\} \sim \mathcal{D}^m$, where $\mathcal{D}$ is some distribution over $\mathcal{X}$, we have*

$$|\mathbb{E}_{\boldsymbol{x} \sim \mathcal{D}}[\ell_{\hat{\alpha}_S}(\boldsymbol{x})] - \mathbb{E}_{\boldsymbol{x} \sim \mathcal{D}}[\ell_{\alpha^*}(\boldsymbol{x})]| = \mathcal{O}\left(\sqrt{\frac{L^2 W \log \Delta + LW \log(Tpk) + \log(1/\delta)}{m}}\right).$$

# 6 Conclusion and future work

In this work, we combined tools from multiple fields, including algebraic geometry, differential geometry, constrained optimization, and statistical learning theory to provide bounds on the sample complexity of hyperparameter tuning when the parameter-dependent dual function admits polynomial structures. We further show that this piecewise polynomial structure is observed in data-driven model hyperparameter tuning of neural networks, specifically for tuning graph kernels for graph convolutional networks and interpolation parameters for neural activation functions. We note that our proposed framework is applicable beyond neural networks for data-driven algorithm design.

A major open question is to extend our work to multiple hyperparameters (i.e., $\boldsymbol{\alpha} \in \mathbb{R}^n$). For the case of one-dimensional hyperparameter, we showed that when the parameter-dependent dual function $f_{\boldsymbol{x}}(\boldsymbol{\alpha}, \boldsymbol{w})$ is piecewise polynomial, then the dual utility function $u_{\boldsymbol{x}}^*(\alpha)$ has bounded oscillations, even if it is not as structured as the parameter-dependent dual; this bounded oscillations structure is sufficient to imply learnability. An interesting open technical question is determining the analogous structure to bounded oscillations in the case of multiple hyperparameters. In the case of one-dimensional hyperparameters, we leveraged and extended tools from differential geometry, combined with constrained optimization techniques to prove that the dual utility function has bounded oscillations. We believe that our idea can serve as a first step in generalizing to the high-dimensional regimes. Another open question is to give a simpler analysis for one-dimensional hyperparameter, potentially under weaker assumptions.

## Acknowledgments

This work was supported in part by NSF grants CCF-1910321 and IIS-1901403.

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

# Contents

# A  Additional related work

**Learning-theoretic complexity of deep nets.**   A related line of work studies the learning-theoretic complexity of deep networks, corresponding to selection of network parameters (weights) over a single problem instance. Bounds on the VC dimension of neural networks have been shown for piecewise linear and polynomial activation functions [35, 34] as well as the broader class of Pfaffian activation functions [36]. Recent work includes near-tight bounds for the piecewise linear activation functions [37] and data-dependent margin bounds for neural networks [38].

**Data-driven algorithm design.**   Data-driven algorithm design [11, 39, 40] is an emerging field that adapts algorithms' internal configuration to specific problem instances, particularly in parameterized algorithms with multiple performance-dictating hyperparameters. Unlike traditional worst-case or average-case analysis, this approach assumes problem instances come from an application-specific distribution. By leveraging available input problem instances, this approach seeks to maximize empirical utilities that measure algorithmic performance for those specific instances. This method has demonstrated effectiveness across various domains, including low-rank approximation and dimensionality reduction [41, 42, 43], accelerating linear system solvers [44, 45], mechanism design [46, 47], sketching algorithms [48], branch-and-cut algorithms for (mixed) integer linear programming [49], learning decision trees [50],among others.  Prior work has also studied online learning of algorithms [51, 52, 53, 54, 55]. In this work, we focus mainly on the statistical learning setting and extending our techniques to online learning is an interesting future direction.

**Neural architecture search.**   Neural architecture search (NAS) captures a significant part of the engineering challenge in deploying deep networks for a given application. While neural networks successfully automate the tedious task of "feature engineering" associated with classical machine learning techniques by automatically learning features from data, it requires a tedious search over a large search space to come up with the best neural architecture for any new application domain. Multiple different a pproaches with different search spaces have been proposed for effective NAS, including searching over the discrete topology of connections between the neural network nodes, and interpolation of activation functions. Due to intense recent interest in moving from hand-crafted to automatically searched architectures, several practically successful approaches have been developed including framing NAS as Bayesian optimization [18, 56, 20], reinforcement learning [15, 19], tree search [57, 58], gradient-based optimization [14], among others, with progress measured over standard benchmarks [59, 60]. [61] introduce a geometry-aware mirror descent based approach to learn the network architecture and weights simultaneously, within a single problem instance, yielding a practical algorithm but without provable guarantees. Our formulation is closely related to tuning the interpolation parameter for activation parameter in the NAS approach of [14], which can be viewed as a multi-hyperparameter generalization of our setup. We establish the first learning guarantees for the simpler case of single hyperparameter tuning.

**Graph-based learning.**   While several classical [62, 63, 64, 65] as well as neural models [22, 66, 67, 68] have been proposed for graph-based learning, the underlying graph used to represent the data typically involves heuristically set graph parameters. The latter approach is usually more effective in practice, but comes without formal learning guarantees. Our work provides the first provable guarantees for tuning the graph kernel hyperparameter in graph neural networks.

**Comparison to Hyperband [8].**   Hyperband is one of the most notable works for hyperparameter tuning of deep neural networks with principled theoretical guarantees, albeit under strong assumptions. Here, we provide a detailed comparison between the guarantees presented in Hyperband and our results.

1. **Hyperparameter configuration setting:** Theoretical results (Theorem 1, Proposition 4) in Hyperband assume finitely many distinct arms and the guarantees are with respect to the best arm in that set. Even their infinite arm setting considers a distribution over the hyperparameter space from which $n$ arms are sampled. It is assumed that $n$ is large enough to sample a good arm with high probability without actually showing that this holds for any concrete hyperparameter loss landscape. It is not clear why this assumption will hold in our cases. In sharp contrast, we seek optimality over the entire continuous hyperparameter hyperparameter range for concrete loss functions which satisfy a piecewise polynomial dual structure.

2. **Guarantees**: The notion of "sample complexity" in Hyperband is very different from ours. Intuitively, their goal is to find the best hyperparameter from learning curves over fewest training epochs, assuming the test loss converges to a fixed value for each hyperparameter after some epochs. By ruling out (successively halving) hyperparameters that are unlikely to be optimal early, they speed up the search process (by avoiding full training epochs for suboptimal hyperparameters). In contrast, we focus on model hyperparameters and assume the network can be trained to optimality for any value of the hyperparameter. We ignore the computational efficiency aspect and focus on the data (sample) efficiency aspect which is not captured in Hyperband analysis.

3. **Learning setting**: Hyperband assumes the problem instance is fixed, and aims to accelerate the random search of hyperparameter configuration for that problem instance with constrained budgets (formulated as a pure-exploration non-stochastic infinite-armed bandit). In contrast, our results assume a problem distribution $\mathcal{D}$ (data-driven setting), and bounds the sample complexity of learning a good hyperparameter for the problem distribution $\mathcal{D}$.

## B   Technical challenges and novelty

Our main contributions (Lemma 3.2, Theorem 4.1) in this paper is introducing new technique for analyzing the model hyperparameter tuning in data-driven setting, where the parameter-dependent dual function $f_{\boldsymbol{x}}(\alpha, \boldsymbol{w})$ admits a specific piecewise polynomial structure. In this section, we will make an in-depth comparison between our setting and the settings in prior work in data-driven algorithm hyperparameter tuning, and discuss why our setting is challenging and requires novel techniques to analyze.

**Challenges.**   Our setting requires significant technical novelty beyond prior work in data-driven algorithm design. Generally, most prior work on statistical data-driven algorithm design falls into two categories:

1. The hyperparameter tuning process does not involve the parameters $\boldsymbol{w}$, that is, the learning algorithm is completely determined by the hyperparameter $\alpha$ and has no parameters that need to be tuned using a training set. Some concrete examples include tuning hyperparameters of hierarchical clustering algorithms [25, 26], branch and bound (B&B) algorithms for (mixed) integer linear programming [69, 28], and graph-based semi-supervised learning [23]. The typical approach is to show that the utility function $u_{\boldsymbol{x}}^*(\alpha)$ admits specific piecewise structure of $\alpha$, typically piecewise polynomial and rational.

2. The hyperparameter tuning process involves the parameters $\boldsymbol{w}$, for example in tuning regularization hyperparameters in linear regression. However, here the optimal parameters $\boldsymbol{w}^*(\alpha)$ can either have a closed analytical form in terms of the hyperparameter $\alpha$ [29], or can be easily approximated in terms of $\alpha$ with bounded error [70].

However, in our setting, the utility function $u_{\boldsymbol{x}}^*(\alpha)$ is defined via an optimization problem $u_{\boldsymbol{x}}^*(\alpha) = \max_{\boldsymbol{w}} f_{\boldsymbol{x}}(\alpha, \boldsymbol{w})$, where the parameter-dependent dual function $f_{\boldsymbol{x}}(\alpha, \boldsymbol{w})$ admits a piecewise polynomial structure. This involves the parameter $\boldsymbol{w}$ so it does not belong to the first case, and it is not clear how to use the second approach either. This is why our problem is quite challenging and requires the development of novel techniques.

**New techniques.**   Two general approaches are known from prior work to establish a generalization guarantee for $\mathcal{U}$.

1. The first approach is to establish a pseudo-dimension bound for $\mathcal{U}$ via alternatively analyzing the learning-theoretic dimensions of the piece and boundary function classes, derived when establishing the piecewise structure of $u_{\boldsymbol{x}}^*(\alpha)$ (following Theorem 3.3 of [12]). *We build on this idea. However, in order to apply it we need significant innovation to analyze the structure of the function $u_{\boldsymbol{x}}^*$ in our case.*

2. The second approach is specialized to the case where the computation of $u_{\boldsymbol{x}}^*(\alpha)$ can be described via the GJ algorithm [48], where we can do four basic operations $(+, -, \times, \div)$ and conditional statements. However, it is not applicable to our case due to the use of a max operation in the definition.

As mentioned above, we follow the first approach though we have to develop new techniques to analyze our setting. Here, we choose to analyze $u_{\boldsymbol{x}}^*(\alpha)$ via indirectly analyzing $f_{\boldsymbol{x}}(\alpha, \boldsymbol{w})$, which is

shown in some cases to admit piecewise polynomial structure. To do that, we have to develop the following:

1. A connection between number of discontinuities and local maxima of the dual utility function $u_{\boldsymbol{x}}^*(\alpha)$, and the learning-theoretic complexity of $\mathcal{U}$.
2. An approach to upper-bound the number of discontinuities and local extrema of $u_{\boldsymbol{x}}^*(\alpha)$. This is done using ideas from differential/algebraic geometry, and constrained optimization. We note that even the tools needed from differential geometry are not readily available, but we have to identify and develop those tools (e.g. monotonic curves and its properties, see Definition 12 and Lemma 18).

That corresponds to the main contribution of our papers (Theorem 3.2, 4.1). We then demonstrate the applicability of our results to two concrete problems in hyperparameter tuning in machine learning (Section 5).

## C  Learning theory background

### C.1  Uniform convergence

**Definition 1** (Uniform convergence, [71])**.** *Let $\mathcal{F}$ be a real-valued function class which takes input from domain $\mathcal{X}$, and $\mathcal{D}$ is a distribution over $\mathcal{X}$. If for any $\epsilon > 0$, and any $\delta \in (0, 1)$, there exists a number $N(\epsilon, \delta)$ depending on $\epsilon$ and $\delta$ such that with probability at least $1 - \delta$ over the draw of $m \geq N(\epsilon, \delta)$ i.i.d. samples $\boldsymbol{x}_1, \ldots, \boldsymbol{x}_m \sim \mathcal{D}$, we have*

$$\Delta_m = \sup_{f \in \mathcal{F}} \left| \frac{1}{m} \sum_{i=1}^{m} f(\boldsymbol{x}_i) - \mathbb{E}_{\boldsymbol{x} \sim \mathcal{D}}[f(\boldsymbol{x})] \right| < \epsilon,$$

*then we say that (the empirical process of) $\mathcal{F}$ uniformly converges with sample complexity $N(\epsilon, \delta)$.*

### C.2  Shattering and pseudo-dimension

We now formally recall the definition of shattering and pseudo-dimension, the main learning-theoretic complexity used throughout this work, as well as the corresponding generalization results.

**Definition 2** (Shattering and pseudo-dimension, [72])**.** *Let $\mathcal{U}$ be a real-valued function class, of which each function takes input in $\mathcal{X}$. Given a set of inputs $S = (\boldsymbol{x}_1, \ldots, \boldsymbol{x}_m) \subset \mathcal{X}$, we say that $S$ is pseudo-shattered by $\mathcal{H}$ if there exists a set of real-valued thresholds $r_1, \ldots, r_m \in \mathbb{R}$ such that*

$$|\{(sign(u(\boldsymbol{x}_1) - r_1), \ldots, sign(u(\boldsymbol{x}_m) - r_m)) \mid u \in \mathcal{U}\}| = 2^m.$$

*The pseudo-dimension of $\mathcal{H}$, denoted as $\mathrm{Pdim}(\mathcal{U})$, is the maximum size $m$ of an input set that $\mathcal{H}$ can shatter.*

The following classical result shows that if a real-valued bounded function class has finite pseudo-dimension, then it has uniform convergence property.

**Theorem C.1** ([72])**.** *Given a real-valued function class $\mathcal{U}$ whose range is $[0, H]$, and assume that $\mathrm{Pdim}(\mathcal{U})$ is finite. Then, given any $\delta \in (0, 1)$, and any distribution $\mathcal{D}$ over the input space $\mathcal{X}$, with probability at least $1 - \delta$ over the drawn of $S = \{\boldsymbol{x}_1, \ldots, \boldsymbol{x}_m\} \sim \mathcal{D}^m$, we have*

$$\left| \frac{1}{m} \sum_{i=1}^{m} u(\boldsymbol{x}_i) - \mathbb{E}_{\boldsymbol{x} \sim \mathcal{D}}[u(\boldsymbol{x})] \right| \leq O\left( H \sqrt{\frac{1}{m} \left( \mathrm{Pdim}(\mathcal{U}) + \ln \frac{1}{\delta} \right)} \right).$$

### C.3  Rademacher complexity

Besides, we also use the notion of Rademacher complexity, as well as its connection to the pseudo-dimension. This learning-theoretic complexity notion is very useful in our analysis, especially for simplifying Assumption 3 to Assumption 1, a critical step when establishing Theorem F.25.

**Definition 3** (Rademacher complexity, [71])**.** *Let $\mathcal{F}$ be a real-valued function class mapping from $\mathcal{X}$ to $[0, 1]$. For a set of inputs $S = \{\boldsymbol{x}_1, \ldots, \boldsymbol{x}_m\}$, we define the empirical Rademacher complexity*

$\hat{\mathscr{R}}_S(\mathcal{F})$ as

$$\hat{\mathscr{R}}_S(\mathcal{F}) = \frac{1}{m}\mathbb{E}_{\epsilon_1,\ldots,\epsilon_m \sim i.i.d \text{ unif. } \pm 1}\left[\sup_{f\in\mathcal{F}}\sum_{i=1}^{m}\epsilon_i f(\boldsymbol{x}_i)\right].$$

*We then define the Rademacher complexity $\mathscr{R}_{\mathcal{D}^m}$, where $\mathcal{D}$ is a distribution over $\mathcal{X}$, as*

$$\mathscr{R}_{\mathcal{D}^m}(\mathcal{F}) = \mathbb{E}_{S\sim\mathcal{D}^m}[\hat{\mathscr{R}}_S(\mathcal{F})].$$

*Furthermore, we define*

$$\mathscr{R}_m(\mathcal{F}) = \sup_{S\in\mathcal{X}^m}\hat{\mathscr{R}}_S(\mathcal{F}).$$

**Theorem C.2** (Empirical Rademacher complexity givrs uniform convergence)**.** *Let $\mathcal{F}$ be a class of functions mapping $\mathcal{X}$ to $[0,1]$. Then for any $\delta \in (0,1)$, with probability at least $1-\delta$ over the draw of $S \sim \mathcal{D}^m$, the following holds simultaneously for all $f \in \mathcal{F}$:*

$$\mathbb{E}_{x\sim\mathcal{D}}[f(x)] \leq \frac{1}{m}\sum_{x\in S}f(x) + 2\mathscr{R}_m(\mathcal{F}) + 3\sqrt{\frac{\ln(2/\delta)}{m}}.$$

The following lemma provides a useful result that allows us to relate the empirical Rademacher complexity of two function classes when the infinity norm between their corresponding dual utility functions is upper-bounded.

**Lemma C.3** ([73])**.** *Let $\mathcal{F} = \{f_{\boldsymbol{\alpha}} : \mathcal{X} \to [0,1] \mid \boldsymbol{\alpha} \in \mathcal{A}\}$ and $\mathcal{G} = \{g_{\boldsymbol{\alpha}} : \mathcal{X} \to [0,1] \mid \boldsymbol{\alpha} \in \mathcal{A}\}$ where $\mathcal{A} \subseteq \mathbb{R}^d$. For any $S \subseteq \mathcal{X}$, we have*

$$\hat{\mathscr{R}}_S(\mathcal{F}) \leq \hat{\mathscr{R}}_S(\mathcal{G}) + \frac{1}{|S|}\sum_{\boldsymbol{x}\in S}\|f_{\boldsymbol{x}}^* - g_{\boldsymbol{x}}^*\|_\infty,$$

*where $f_{\boldsymbol{x}}^* : \mathcal{A} \to [0,1]$ and $g_{\boldsymbol{x}}^* : \mathcal{A} \to [0,1]$ are defined as*

$$f_{\boldsymbol{x}}^*(\boldsymbol{\alpha}) := f_{\boldsymbol{\alpha}}(\boldsymbol{x}), \quad g_{\boldsymbol{x}}^*(\boldsymbol{\alpha}) := g_{\boldsymbol{\alpha}}(\boldsymbol{x}).$$

The following theorem establishes a connection between pseudo-dimension and Rademacher complexity.

**Lemma C.4** ([74])**.** *Let $\mathcal{F}$ be a function class with a bounded pseudo-dimension consisting of functions $f : \mathcal{F} \to [0,1]$. Then $\mathscr{R}_m(\mathcal{F}) = \mathcal{O}\left(\sqrt{\frac{\mathrm{Pdim}(\mathcal{F})}{m}}\right).$*

# D  Additional results and omitted proofs for Section 2.1

In this Appendix, we will recall the definition of *oscillations* of a real-valued function proposed by Balcan et al. [12], and extend their results to obtain useful lemmas for bounding the pseudo-dimension of our function classes.

## D.1  Oscillations and connection to pseudo-dimension

When the function class $\mathcal{U} = \{u_\alpha : \mathcal{X} \to \mathbb{R} \mid \alpha \in \mathbb{R}\}$ is parameterized by a real-valued index $\alpha$, [12] propose a convenient way for bounding the pseudo-dimension of $\mathcal{H}$, via bounding the *oscillations* of the dual function $u_{\boldsymbol{x}}^*(\alpha) := u_\alpha(\boldsymbol{x})$ corresponding to any problem instance $\boldsymbol{x}$. In this section, we will recall the notions of oscillations and the connection of the pseudo-dimension of a function class with the oscillations of its dual functions. This tool is very helpful in our later analyses.

**Definition 4** (Oscillations, [12])**.** *A function $h : \mathbb{R} \to \mathbb{R}$ has at most $B$ oscillations if for every $z \in \mathbb{R}$, the function $\alpha \mapsto \mathbb{I}_{\{h(\alpha)\geq z\}}$ is piecewise constant with at most $B$ discontinuities.*

An illustration of the notion of oscillations can be found in Figure 2. Using the idea of oscillations, one can analyze the pseudo-dimension of parameterized function classes by alternatively analyzing the oscillations of their dual functions, formalized as follows.

**Theorem D.1** ([12])**.** *Let $\mathcal{U} = \{u_\alpha : \mathcal{X} \to \mathbb{R} \mid \alpha \in \mathbb{R}\}$, of which each dual function $u_{\boldsymbol{x}}^*(\alpha)$ has at most $B$-oscillations. Then $\mathrm{Pdim}(\mathcal{U}) = \mathcal{O}(\log B).$*

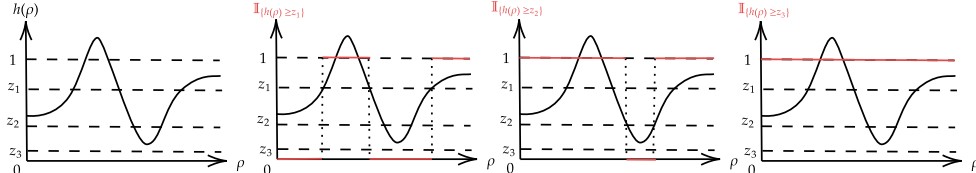

Figure 2: This figure demonstrates the oscillation property for a function $h : \mathbb{R} \to \mathbb{R}$. The oscillation of a function $h$ is defined as the maximum number of discontinuities in the function $\mathbb{I}_{\{h(\alpha)\geq z\}}$, as the threshold $z$ varies. The figure shows several graphs of $\mathbb{I}_{\{h(\alpha)\geq z\}}$ corresponding to different choices of the threshold $z$. We can observe that when $z = z_1$, the function $\mathbb{I}_{\{h(\alpha)\geq z\}}$ exhibits the highest number of discontinuities, which is 4. This number of discontinuities is also the maximum for any choice of $z$. Therefore, we conclude that $h$ has 4 oscillations.

## D.2 Omitted proofs for Section 2.1

The following result allows us to draw a connection between the number of discontinuities and local maxima of piecewise continuous function and its number of oscillations.

**Lemma D.2.** *Let $h : \mathbb{R} \to \mathbb{R}$ be a piecewise continuous function which has at most $B_1$ discontinuity points, and has at most $B_2$ local maxima. Then $h$ has at most $\mathcal{O}(B_1 + B_2)$ oscillations.*

*Proof.* We show that in each interval where $h$ is continuous, we can bound for any $z$, the number of solutions of $h(\alpha) = z$ using the number of local maxima of $h$. Aggregating the number of solutions across all continuous intervals of $h$ yields the desired result.

For any $z \in \mathbb{R}$, consider the function $g(\alpha) = \mathbb{I}_{\{h(\alpha)\geq z\}}$. By definition, on any interval over which $h$ is continuous, any discontinuity point of $g(\alpha)$ is a root of the equation $h(\alpha) = z$. Therefore, it suffices to give an upper-bound on the number of roots that the equation $h(\alpha) = z$ can have across all the intervals where $h$ is continuous.

Let $\alpha_1 < \alpha_2 < \cdots < \alpha_N$ be the discontinuity points of $h$, where $N \leq B_1$ from assumption. For convenience, let $\alpha_0 = -\infty$ and $\alpha_{N+1} = \infty$. For any $i = 1, \ldots, N$, consider an interval $I_i = (\alpha_i, \alpha_i + 1)$ over which the function $h$ is continuous. Assume that there are $E_i$ local maxima of the function $h$ in the interval $I_i$, meaning that there are at most $2E_i + 1$ local extrema. We now claim that there are at most $2E_i + 2$ roots of $h(\alpha) = z$ in $I_i$. We proceed by contradiction: assume that $\alpha_1^* < \alpha_2^* < \cdots < \alpha_{2E_i+3}^*$ are $2E_i + 3$ roots of the equation $h(\alpha) = z$, and there is no other root in between. We have the following claims:

- Claim 1: there is at least one local extremum of $h$ in $(\alpha_j^*, \alpha_{j+1}^*)$. Since $h$ has a finite number of local extrema, meaning that $h$ cannot be constant over $[\alpha_j^*, \alpha_{j+1}^*]$. Therefore, there exists some $\alpha' \in (\alpha_j^*, \alpha_{j+1}^*)$ such that $h(\alpha') \neq z$, and note that $z = h(\alpha_j^*) = h(\alpha_{j+1}^*)$. Since $h$ is continuous over $[\alpha_j^*, \alpha_{j+1}^*]$, from extreme value theorem (Theorem F.6), $h$ (when restricted to $[\alpha_j^*, \alpha_{j+1}^*]$) reaches minima and maxima over $[\alpha_j^*, \alpha_{j+1}^*]$. However, since there exists $\alpha'$ such that $h(\alpha') \neq z$, then $h$ has to achieve minima or maxima in the interior $(\alpha_j^*, \alpha_{j+1}^*)$. That is also a local extremum of $h$.
- Claim 2: there are at least $2E_i + 2$ local extrema in $(\alpha_1^*, \alpha_{E_i+2}^*)$. This claim follows directly from Claim 1.

Claim 2 leads to a contradiction. Therefore, there are at most $2E_i + 2$ roots in the interval $I_i$. which implies there are $\sum_{i=0}^{N} 2E_i + 2N$ roots in the intervals $I_i$ for $i = 1, \ldots, N$. Note that $\sum_{i=0}^{N} E_i \leq B_2$, $N \leq B_1$ by assumption, and each discontinuity point of $h$ could also be discontinuity point of $g$, we conclude that there are at most $\mathcal{O}(B_1 + B_2)$ discontinuity points for $g$, for any $z$. $\qquad\square$

We are now ready to give a formal proof for Lemma 2.1.

**Theorem 2.1 (restated).** *Consider a real-valued function class $\mathcal{U} = \{u_\alpha : \mathcal{X} \to \mathbb{R} \mid \alpha \in \mathbb{R}\}$, of which each dual function $u_{\boldsymbol{x}}^*(\alpha)$ is piecewise continuous, with at most $B_1$ discontinuities and $B_2$ local maxima. Then $\mathrm{Pdim}(\mathcal{U}) = \mathcal{O}(\log(B_1 + B_2))$.*

*Proof.* From assumption, for each problem instance $\boldsymbol{x}$, the dual utility function $u_{\boldsymbol{x}}^*(\alpha)$ is piecewise continuous with at most $B_1$ discontinuities and $B_2$ local maxima. Combining with Lemma D.2, $u_{\boldsymbol{x}}^*(\alpha)$ has $\mathcal{O}(\log(B_1 + B_2))$ oscillations. From Theorem D.1, we have the conclusion. $\square$

Compared to Lemma 2.1, the proof of Lemma 2.2 is more straightforward as it can be directly derived from the definition of oscillation (Definition 4).

**Lemma 2.2 (restated).** *Consider a real-valued function class $\mathcal{U} = \{u_\alpha : \mathcal{X} \to \mathbb{R} \mid \alpha \in \mathbb{R}\}$, of which each dual function $u_{\boldsymbol{x}}^*(\alpha)$ is piecewise constant with at most $B$ discontinuities. Then $\mathrm{Pdim}(\mathcal{U}) = \mathcal{O}(\log B)$.*

*Proof.* Consider a dual function $u_{\boldsymbol{x}}^*(\alpha)$ which is a piecewise constant function with at most $B$ discontinuities. $\mathbb{I}_{\{u_{\boldsymbol{x}}^*(\alpha) \geq z\}}$ is piecewise continuous with at most $B$ discontinuities for any threshold $z \in \mathbb{R}$. Assume, for the sake of contradiction, that there exists $z \in \mathbb{R}$ such that $\mathbb{I}_{\{u_{\boldsymbol{x}}^*(\alpha) \geq z\}}$ has $N$ discontinuities, for some $N \geq B + 1$. Since $u_{\boldsymbol{x}}^*(\alpha)$ is piecewise constant, any discontinuities of $\mathbb{I}_{\{u_{\boldsymbol{x}}^*(\alpha) \geq z\}}$ is also a discontinuity of $u_{\boldsymbol{x}}^*(\alpha)$, meaning that $u_{\boldsymbol{x}}^*(\alpha)$ has at least $N$ discontinuities, which leads to a contradiction. Therefore, we conclude that $u_{\boldsymbol{x}}^*(\alpha)$ has at most $B$ oscillations, and thus $\mathrm{Pdim}(\mathcal{H}) = \mathcal{O}(\log B)$ by Theorem D.1. $\square$

**Remark 3.** *In many applications, the partition of $f_{\boldsymbol{x}}(\alpha, \boldsymbol{w})$ into connected components is typically defined by $M$ boundary functions $h_{\boldsymbol{x},1}(\alpha, \boldsymbol{w}), \ldots, h_{\boldsymbol{x},M}(\alpha, \boldsymbol{w})$. These boundary functions are often polynomials in $d + 1$ variables with a bounded degree $\Delta$. For such cases, we can establish an upper bound on the number of connected components created by these boundary functions, represented as $\mathbb{R}^{d+1} - \cup_{i=1}^M \{(\alpha, \boldsymbol{w}) \mid h_{\boldsymbol{x},i}(\alpha, \boldsymbol{w}) = 0\}$, using only $\Delta$ and $d$. This bound (see Theorem F.9 in Appendix F.1.2) serves as a crucial step in applying Theorem 3.2.*

Notice that Lemma D.2 does not apply in the case where the function $h$ may have constant pieces, as they correspond to infinitely many local maxima. To handle this more generally, we introduce the following definition.

**Definition 5** ($B$-monotonicity). *A function $h : \mathbb{R} \to \mathbb{R}$ is said to be $B$-monotonic if its domain can be partitioned into at most $B$ intervals such that on each interval: either (a) $h$ is strictly monotonic and continuous, or (b) $h$ is a constant function.*

While a bounded number of discontinuity points and local maxima implies $B$-monotonicity, the converse may not be true. We now show how to bound the number of oscillations of $B$-monotonic functions.

**Lemma D.3.** *Any $B$-monotonic function $h : \mathbb{R} \to \mathbb{R}$ has $\mathcal{O}(B)$ oscillations.*

*Proof.* Suppose $I$ be an interval over which $h$ is piecewise constant. Then the function $g_z : \rho \mapsto \mathbb{I}_{\{h(\rho) \geq z\}}$ is also constant over $I$ for any $z \in \mathbb{R}$ (either constant zero or constant one throughout $I$ depending on $z$). On the other hand, if $h$ is strictly monotonic and continuous over $I$, then $g_z$ is piecewise constant over $I$ with at most two pieces. Thus, $g_z$ has at most $2B$ discontinuities for any $z$, or $h$ has $\mathcal{O}(B)$ oscillations by Definition 4. $\square$

We conclude with the following corollary (immediate from Lemma D.3 and Theorem D.1) for the special case of piecewise constant functions. Compared to Lemma 2.1, the proof of Lemma 2.2 is more straightforward as it can be directly derived from the definition of oscillation (Definition 4).

**Lemma 2.2 (restated).** *Consider a real-valued function class $\mathcal{U} = \{u_\alpha : \mathcal{X} \to \mathbb{R} \mid \alpha \in \mathbb{R}\}$, of which each dual function $u_{\boldsymbol{x}}^*(\alpha)$ is piecewise constant with at most $B$ discontinuities. Then $\mathrm{Pdim}(\mathcal{U}) = \mathcal{O}(\log B)$.*

*Proof.* Consider a dual function $u_{\boldsymbol{x}}^*(\alpha)$ which is a piecewise constant function with at most $B$ discontinuities. $\mathbb{I}_{\{u_{\boldsymbol{x}}^*(\alpha) \geq z\}}$ is piecewise continuous with at most $B$ discontinuities for any threshold $z \in \mathbb{R}$. Assume, for the sake of contradiction, that there exists $z \in \mathbb{R}$ such that $\mathbb{I}_{\{u_{\boldsymbol{x}}^*(\alpha) \geq z\}}$ has $N$ discontinuities, for some $N \geq B + 1$. Since $u_{\boldsymbol{x}}^*(\alpha)$ is piecewise constant, any discontinuities of $\mathbb{I}_{\{u_{\boldsymbol{x}}^*(\alpha) \geq z\}}$ is also a discontinuity of $u_{\boldsymbol{x}}^*(\alpha)$, meaning that $u_{\boldsymbol{x}}^*(\alpha)$ has at least $N$ discontinuities, which leads to a contradiction. Therefore, we conclude that $u_{\boldsymbol{x}}^*(\alpha)$ has at most $B$ oscillations, and thus $\mathrm{Pdim}(\mathcal{H}) = \mathcal{O}(\log B)$ by Theorem D.1. $\square$

# E    An illustration for the proof in Section 3

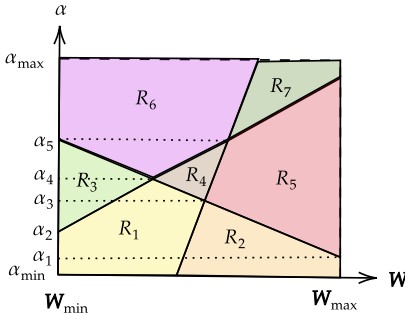

Figure 3: A demonstration of the proof idea for Theorem 3.1: We begin by partitioning the domain $\mathcal{A}$ of the dual utility function $u_{\boldsymbol{x}}^*(\alpha)$ into intervals. This partitioning is formed using two key points for each connected component $R$ in the partition $\mathcal{P}_{\boldsymbol{x}}$ of the domain $\mathcal{A} \times \mathcal{W}$ of $f_{\boldsymbol{x}}(\alpha, \boldsymbol{w})$: $\alpha_{R,\inf} = \inf_\alpha \{\alpha : \exists \boldsymbol{w}, (\alpha, \boldsymbol{w}) \in R\}$ and $\alpha_{R,\sup} = \sup_\alpha \{\alpha : \exists \boldsymbol{w}, (\alpha, \boldsymbol{w}) \in R\}$. Given that $\mathcal{P}$ contains $N$ elements, the number of such points is $\mathcal{O}(N)$. We demonstrate that the dual utility functions $u_{\boldsymbol{x}}^*$ remain constant over each interval defined by these points.

# F    Additional results and omitted proofs for Section 4

In this section, we will present the required background and supporting results for Lemma 4.

## F.1    General auxiliary results

We first present some useful auxiliary lemmas.

### F.1.1    Results for analyzing local maxima of pointwise maximum function

In this section, we recall some elementary results which are crucial in our analysis. The following lemma says that the pointwise maximum of continuous functions is also a continuous function.

**Lemma F.1.** *Let $f_i : \mathcal{X} \to \mathbb{R}$, where $i = 1, \ldots, N$, be continuous functions over a subset $\mathcal{X} \subseteq \mathbb{R}^n$, and let $f(\boldsymbol{x}) = \max_{i \in \{1,\ldots,N\}} f_i(\boldsymbol{x})$. Then we have $f(\boldsymbol{x})$ is a continuous function over $\mathcal{X}$.*

*Proof.* In the case $N = 2$, we can rewrite $f(\boldsymbol{x})$ as

$$f(\boldsymbol{x}) = \frac{f_1(\boldsymbol{x}) + f_2(\boldsymbol{x})}{2} + \frac{1}{2} |f_1(\boldsymbol{x}) - f_2(\boldsymbol{x})|,$$

which is the sum of continuous functions. Hence, $f(\boldsymbol{x})$ is continuous. Assume the claim holds for $N = k$, we then claim that it also holds for $N = k + 1$ by rewriting $f(\boldsymbol{x})$ as

$$f(\boldsymbol{x}) = \max \left\{ \max_{i \in \{1,\ldots,k\}} \{f_i(\boldsymbol{x})\}, f_{k+1}(\boldsymbol{x}) \right\}.$$

Therefore, the claim is established by induction. $\square$

The following results are helpful when we want to bound the number of local maxima of pointwise maximum of differentiable functions. In particular, we show that the local maxima of $g(\boldsymbol{x}) = \max_{i \in \{1,\ldots,n\}} g_i(\boldsymbol{x})$ is the local extrema of one of the functions $g_i(\boldsymbol{x})$. This property helps us controlling the number of local maxima of $g(\boldsymbol{x})$ by controlling the number of local maxima of each function $g_i(\boldsymbol{x})$.

**Proposition F.2.** *Let $\mathcal{X}$ be a finite-dimensional Euclidean space and $g_i : \mathcal{X} \to \mathbb{R}$, where $i = 1, \ldots, N$, be continuous functions on $\mathcal{X}$ with the local maxima on $\mathcal{X}$ is given by the set $C_i$. Then the function $g(\boldsymbol{x}) = \max_{i \in \{1,\ldots,N\}} \{g_i(\boldsymbol{x})\}$ has its local maxima contained in the union $\cup_{i \in \{1,\ldots,N\}} C_i$.*

*Proof.* Let $\boldsymbol{x}'$ be a local maxima of $g(\boldsymbol{x})$. It means there exists a neighborhood $\mathcal{N}_{\boldsymbol{x}'}$ of $\boldsymbol{x}'$ such that for any $\boldsymbol{x} \in \mathcal{N}_{\boldsymbol{x}'}$, we have $g(\boldsymbol{x}') \geq g(\boldsymbol{x})$. Let $g_i$ be a function s.t. $g_i(\boldsymbol{x}') = g(\boldsymbol{x}')$. Then

$$g_i(\boldsymbol{x}') = g(\boldsymbol{x}') \geq g(\boldsymbol{x}) = \max_{j \in \{1,\dots,N\}} g_j(\boldsymbol{x}) \geq g_i(\boldsymbol{x}),$$

for any $\boldsymbol{x} \in \mathcal{N}_{\boldsymbol{x}'}$. This means that $\boldsymbol{x}'$ is the local maxima of $g_i$, i.e, $\boldsymbol{x}' \in C_i \subset \cup_{j \in \{1,\dots,N\}} C_j$. $\quad\square$

We recall the Lagrangian multipliers theorem, which allows us to give a necessary condition for the extrema of a function over a constraint.

**Theorem F.3** (Lagrangian multipliers, [75]). *Let $h : \mathbb{R}^d \to \mathbb{R}$, $\boldsymbol{f} : \mathbb{R}^d \to \mathbb{R}^n$ be $C^1$ functions, and let $Z_{\boldsymbol{f}} = \{\boldsymbol{x} \in \mathbb{R}^d \mid \boldsymbol{f}(\boldsymbol{x}) = 0\} \subseteq \mathbb{R}^d$. Assume that for all $\boldsymbol{x}_0 \in Z_{\boldsymbol{f}}$, $\mathsf{rank}(J_{\boldsymbol{f},\boldsymbol{x}}(\boldsymbol{x}_0)) = n$. If $\boldsymbol{x}'$ is a local extrema of $h$ on $Z_{\boldsymbol{f}}$, then there exists $\boldsymbol{\lambda} = (\lambda_1, \dots, \lambda_n) \in \mathbb{R}^n$ such that:*

$$\nabla h(\boldsymbol{x}') = \sum_{i=1}^{n} \lambda_i \nabla f_i(\boldsymbol{x}'), \quad \text{and} \quad \boldsymbol{f}(\boldsymbol{x}') = \boldsymbol{0},$$

*where $\boldsymbol{\lambda}$ is called Lagrangian multipliers.*

We next recall the following well-known results that characterize the local extrema of continuously differentiable functions over a compact set.

**Lemma F.4** (Fermat's interior extremum theorem). *Let $f : D \to \mathbb{R}$ be a continuously differentiable function, where $D \subseteq \mathbb{R}^m$ is an open set, and suppose that $\boldsymbol{x}'$ is a local extrema of $f$ in $D$. Then $\nabla f(\boldsymbol{x}') = \boldsymbol{0}$.*

**Corollary F.5.** *Let $f : D \to \mathbb{R}$ be a differentiable function, where $D \subset \mathbb{R}^m$ is a compact set. If $\boldsymbol{x}' \in D$ is a local extrema of $f$ in $D$, then $x_0$ is either: (1) an interior point of $D$ and $\nabla f(\boldsymbol{x}') = \boldsymbol{0}$, or (2) a point in the boundary $\mathrm{bd}(D)$ of $D$.*

**Lemma F.6** (Extreme value theorem). *Let $f : D \to \mathbb{R}$ be a continuous function, where $D \subset \mathbb{R}^m$ is a compact set, then $f$ is bounded in $D$ and there exists $\boldsymbol{x}_1, \boldsymbol{x}_2 \in D$ such that $f(\boldsymbol{x}_1) = \sup_{\boldsymbol{x} \in D} f(\boldsymbol{x})$ and $f(\boldsymbol{x}_2) = \inf_{\boldsymbol{x} \in D} f(\boldsymbol{x})$.*

We also recall the well-known Sauer-Shelah Lemma.

**Lemma F.7** ([76]). *Let $1 \leq k \leq n$, where $k$ and $n$ are positive integers. Then*

$$\sum_{j=0}^{k} \binom{n}{j} \leq \left(\frac{en}{k}\right)^k.$$

### F.1.2 Results for bounding the number of connected components defined by algebraic sets

In this section, we formally define connected components and extreme points.

**Definition 6** (Connected components). *A connected component of a set $S \subset \mathbb{R}^d$ is a maximal nonempty subset $A \subseteq S$ such that any two points of $A$ are connected by a continuous curve lying in $A$.*

**Definition 7** (Extreme points of a connected component). *Let $S \subset \mathbb{R} \times \mathbb{R}^m$ and let $A$ be a connected component of $S$. We call $x_{A,\inf} = \inf\{x \in \mathbb{R} \mid \exists \boldsymbol{y} \in \mathbb{R}^m, (x, \boldsymbol{y}) \in A\}$, and $x_{A,\sup} = \sup\{x \in \mathbb{R} \mid \exists \boldsymbol{y} \in \mathbb{R}^m, (x, \boldsymbol{y}) \in A\}$ the $x$-extreme points of $A$.*

The following theorems allow us to upper-bound the number of connected components defined by algebraic sets and the complement of algebraic sets.

**Lemma F.8** ([77]). *Let $p$ be a polynomial in $n$ variables. If the degree of polynomial $p$ is $\Delta$, the number of connected components of $\{\boldsymbol{z} \in \mathbb{R}^n \mid p(\boldsymbol{z}) = 0\}$ is at most $2\Delta^n$.*

**Theorem F.9** ([77]). *Suppose $N \geq n$. Consider $N$ polynomials $p_1, \dots, p_N$ in $n$ variables, each of degree at most $\Delta$. Then the number of connected components of $\mathbb{R}^n - \cup_{i=1}^{N}\{\boldsymbol{z} \in \mathbb{R}^n \mid p_i(\boldsymbol{z}) = 0\}$ is $\mathcal{O}\left(\frac{N\Delta}{n}\right)^n$.*

For a connected component $C$ of $\mathbb{R}^n - \cup_{i=1}^{N} Z_{p_i}$, we can define its adjacent boundaries, which is the algebraic set $Z_{p_i}$ that is adjacent to $C$.

**Definition 8** (Adjacent boundaries). *Consider $N$ polynomials $p_1, \ldots, p_N$ in $n$ variables, and let $Z_{p_i} = \{z \in \mathbb{R}^n \mid p_i(z) = 0\}$ be the algebraic set defined by $p_i$. Let $C$ be any connected component of $\mathbb{R}^n - \cup_{i=1}^{N} Z_{p_i}$. We say that $Z_{p_i}$ is an adjacent boundary to $C$ if $\overline{C} \cap Z_{p_i} \neq \emptyset$. Here $\overline{C}$ is the closure of $C$.*

The following result is a direct consequence of Bezout's theorem [78].

**Corollary F.10.** *Let $f_1, \ldots, f_n : \mathbb{R}^n \to \mathbb{R}$ be $n$ polynomials in $n$ variables of degree $d_1, \ldots, d_n$, respectively. Assuming that $\cap_{i=1}^{n} Z_{f_i}$ has only isolated points, then the number of isolated solutions of $\cap_{i=1}^{n} Z_{f_i}$ is at most $\prod_{i=1}^{n} d_i$.*

We also have the Bezout's theorem specialized for the two-dimensional case.

**Corollary F.11** (Bezout's theorem on a plane, [79]). *Let $f_1, f_2 : \mathbb{R}^2 \to \mathbb{R}$ are two polynomials with no common factor (of degree more than 1). Let $Z_{f_1} \cap Z_{f_2} = \{(x, y) \in \mathbb{R}^2 \mid f_1(x, y) = f_2(x, y) = 0\}$. Then the number of points in $Z_{f_1} \cap Z_{f_2}$ is at most $\deg(f_1) \cdot \deg(f_2)$.*

## F.2  Background on differential geometry

In this section, we will introduce some basic terminology of differential geometry, as well as key results that we use in our proofs. First, we recall the definition of a topological manifold.

**Definition 9** (Topological manifold, [80]). *A subset $M \subseteq \mathbb{R}^n$ is a topological manifold of dimension $k \leq n$ if:*

- *For any $p \in M$, there exists an open neighborhood $U \subseteq \mathbb{R}^N$ of $p$ and a homeomorphism $\phi : U \cap M \to V$, where $V \subset \mathbb{R}^k$ is open.*

- *$M$ is equipped with the subspace topology inherited from $\mathbb{R}^n$ (i.e., a subset $S \subset M$ is open in $M$ if there is an open set $S'$ in $\mathbb{R}^n$ such that $S = S' \cap M$), is Hausdorff (any two points in $M$ have two corresponding disjoint neighborhoods), and second-countable ($M$ has a countable basis).*

Using the subspace topology, we can define the open set and neighborhood in a topological manifold as follows.

**Definition 10** (Open set). *Let $M \subseteq \mathbb{R}^n$ be a topological manifold. A subset $S \subset M$ is called open in $M$ if there exists an open set $S'$ in $\mathbb{R}^n$ such that $S = M \cap S'$.*

**Definition 11** (Neighborhood). *Let $M \subseteq \mathbb{R}^n$ be a topological manifold, and let $p$ is a point in $M$. Then a neighborhood of $p$ in $M$ is an open subset $S$ of $M$ that contains $p$.*

We will now define charts, atlases, and smooth (differentiable) manifolds. Roughly speaking, given a manifold $M$, one can think of a chart $(U, \phi)$, where $U$ is a neighborhood of some point in $M$, as a way to assign (Euclidean) coordinates to a local region in $M$. An atlas then describes the local coordinate systems that cover all $M$, and the transition map describes how we convert coordinates between overlapping charts.

**Definition 12** (Charts, atlas, and transition map, [80]). *A chart on $M \subseteq \mathbb{R}^n$ is a pair $(U, \phi)$, where $U \subset M$ is open, and $\phi : U \to \mathbb{R}^k$ is a homeomorphism. An atlas is a collection of charts $\{(U_\alpha, \phi_\alpha)\}$ covering $M$, i.e., $M \subseteq \cup_\alpha U_\alpha$. For overlapping charts $(U_\alpha, \phi_\alpha)$ and $(U_\beta, \phi_\beta)$, the map $\phi_\beta \circ \phi_\alpha^{-1} : \phi_\alpha(U_\alpha \cap U_\beta) \to \phi_\beta(U_\alpha \cap U_\beta)$ is called the transition mapping between open subsets of $R^k$.*

**Definition 13** (Smooth manifold, [80]). *A subset $M \subseteq \mathbb{R}^n$ is a smooth manifold of dimension $k \leq n$ if it has an atlas where all transitions maps are smooth ($C^\infty$).*

We now define the smooth map between smooth manifolds.

**Definition 14** (Smooth map between smooth manifolds, [80]). *Let $M \subset \mathbb{R}^n$ and $N \subset \mathbb{R}^m$ be smooth manifolds in $\mathbb{R}^n$ and $\mathbb{R}^m$, respectively. A map $\boldsymbol{f} : M \to N$ is called smooth if: for every $\boldsymbol{p} \in M$, there exists charts $(U, \phi)$ containing $p$ and $(V, \psi)$ containing $\boldsymbol{f}(\boldsymbol{p})$ such that the coordinate representation $\psi \circ \boldsymbol{f} \circ \psi^{-1} : \phi(U \cap \boldsymbol{f}^{-1}(V)) \to \psi(V)$ is smooth in the standard sense (i.e., infinitely differentiable).*

In case that $M$ and $N$ are Euclidean spaces, the definition of a smooth map just has the standard sense.

**Definition 15** (Smooth map between Euclidean spaces). $\boldsymbol{f} : \mathbb{R}^n \to \mathbb{R}^m$ *is a smooth map if it is infinitely differentiable, i.e., if every partial derivative of $\boldsymbol{f}$ exists and is continuous.*

We now define the tangent space and the regular value of a smooth map. Intuitively, the tangent space $T_p(M)$ of a smooth manifold $M$ at point $p$ represents all the directions that we can move along from $p$ and still stay in the manifold $M$.

**Definition 16** (Tangent space, [80]). *Let $\gamma : (-\epsilon, \epsilon) \to M$ be a smooth curve in $M$ with $\gamma(0) = \boldsymbol{p}$. The tangent vector $\boldsymbol{v} = \gamma'(0)$ represents a direction "along $M$" at $\boldsymbol{p}$. The tangent space $T_{\boldsymbol{p}}(M)$ is the set of all such vectors $\boldsymbol{v}$ obtained from smooth curves though $\boldsymbol{p}$, i.e.,*

$$T_{\boldsymbol{p}}(M) = \{\gamma'(0) \mid \gamma : (-\epsilon, \epsilon) \to M \text{ smooth}, \gamma(0) = \boldsymbol{p}\}.$$

**Definition 17** (Regular value, [80]). *Let $M \subseteq \mathbb{R}^n$ be a smooth $k$-dimensional submanifold, and let $\boldsymbol{f} : M \to \mathbb{R}^m$ be a smooth map. A point $\boldsymbol{\epsilon} \in \mathbb{R}^m$ is called a regular value of $\boldsymbol{f}$ if for every $\boldsymbol{p} \in \boldsymbol{f}^{-1}(\boldsymbol{\epsilon})$, the Jacobian evaluated at $\boldsymbol{p}$ of an ambient map $\tilde{\boldsymbol{f}} : R^n \to R^m$ of $\boldsymbol{f}$ extended to the ambient space $\mathbb{R}^n$, when restricted to $T_{\boldsymbol{p}}(M)$, has rank $m$:*

$$\text{rank}\left(J_{\tilde{\boldsymbol{f}}}(\boldsymbol{p})\Big|_{T_{\boldsymbol{p}}(M)}\right) = \dim\left(\{J_{\tilde{\boldsymbol{f}}}(p)\boldsymbol{z} \mid \boldsymbol{z} \in T_{\boldsymbol{p}}(M)\}\right) = m.$$

In case $M$ is an open subset in $\mathbb{R}^n$ (including $\mathbb{R}^n$ itself), we can simplify the definition of regular value as follow.

**Corollary F.12.** *Let $M$ is an open set in $\mathbb{R}^n$, and let $\boldsymbol{f} : M \to \mathbb{R}^m$ be a smooth map. Then $\epsilon$ is a regular value of $f$ if for any $p \in f^{-1}(\boldsymbol{\epsilon})$, the Jacobian matrix $J_f(p)$ has rank $m$.*

The following result shows the smooth manifold structure of the preimage of regular values.

**Lemma F.13** (Preimage theorem, [80]). *Let $\boldsymbol{f} : M \to \mathbb{R}^m$ is a smooth map, where $M \subseteq \mathbb{R}^n$ is a $k$-dimensional smooth manifold of $\mathbb{R}^n$ and $k \geq m$. Let $\boldsymbol{\epsilon} \in \mathbb{R}^m$ be a regular value of $\boldsymbol{f}$. Then $\boldsymbol{f}^{-1}(\boldsymbol{\epsilon})$ defines a $(k-m)$-dimensional smooth sub-manifold in $M$.*

In the case $M = \mathbb{R}^n$, we can further simplify the lemma above as follows.

**Corollary F.14.** *Let $\boldsymbol{f} : \mathbb{R}^n \to \mathbb{R}^m$ is a smooth map, and let $\boldsymbol{\epsilon} \in \mathbb{R}^m$ be a regular value of $\boldsymbol{f}$. Then $\boldsymbol{f}^{-1}(\boldsymbol{\epsilon})$ defines a $(k-m)$-dimensional smooth manifold in $\mathbb{R}^n$.*

The following theorem says that for any smooth map $\boldsymbol{f}$, the set of non-regular values of $\boldsymbol{f}$ has Lebesgue measure zero.

**Theorem F.15** (Sard's theorem, [80]). *Let $M \subseteq \mathbb{R}^n$ be a smooth manifold in $\mathbb{R}^n$, and let $\boldsymbol{f} : M \to \mathbb{R}^m$ be a smooth map. Then the set of non-regular value of $\boldsymbol{f}$ has Lebesgue measure zero in $\mathbb{R}^m$. Moreover, if $\dim(M) \geq m$, then $\boldsymbol{f}^{-1}(\boldsymbol{\epsilon})$ has dimension $\dim(M) - m$, and if $\dim(M) < m$, then $\boldsymbol{f}(M)$ has measure zero.*

### F.3   Monotonic curve and its properties

In this section, we propose the definition of monotonic curve and establish some important properties, which will play a key role in our main results.

**Definition 18** (Polynomial map and zero set). *Let*

$$\begin{aligned} \boldsymbol{f} :\mathbb{R} \times \mathbb{R}^n &\to \mathbb{R}^m \\ (x, \boldsymbol{y}) &\mapsto (f_1(x, \boldsymbol{y}), \ldots, f_d(x, \boldsymbol{y})) \end{aligned}$$

*be a smooth map. If for any $i = 1, \ldots, m$, $f_i(x, \boldsymbol{y})$ is a polynomial of $x$ and $\boldsymbol{y}$, then we call $\boldsymbol{f}$ the polynomial map, and $Z_{\boldsymbol{f}} = \{(x, \boldsymbol{y}) \in \mathbb{R} \times \mathbb{R}^n \mid \boldsymbol{f}(x, \boldsymbol{y}) = 0\}$ the zero set of the polynomial map $\boldsymbol{f}$.*

Specifically, if $n = m$ and $\boldsymbol{0} \in \mathbb{R}^n$ is a regular value of the polynomial map, then the zero-set $Z_{\boldsymbol{f}}$ inherits favorable structure. We note that $\boldsymbol{f}$ being a polynomial map allows us to use Warren's theorems (Theorem F.9, Lemma F.8) to derive the upper-bound for the number of connected components for $Z_{\boldsymbol{f}}$ and $\mathbb{R} \times \mathbb{R}^m - Z_{\boldsymbol{f}}$, regardless of whether $\boldsymbol{0}$ is a regular value of $\boldsymbol{f}$.

**Corollary F.16.** *Let $\boldsymbol{f} : \mathbb{R} \times \mathbb{R}^m \to \mathbb{R}^m$ be a polynomial map. Assume that $\boldsymbol{0} \in \mathbb{R}^m$ is a regular value of $\boldsymbol{f}$, then $Z_{\boldsymbol{f}}$ defines a smooth 1-dimensional manifold in $\mathbb{R} \times \mathbb{R}^m$.*

*Proof.* This result follows directly from Corollary F.14. □

We are now ready to define monotonic curves.

**Definition 19** ($x$-Monotonic curve). *Let $\boldsymbol{f} : \mathbb{R} \times \mathbb{R}^m \to \mathbb{R}^m$ be a polynomial map, and assume that $\mathbf{0} \in \mathbb{R}^m$ is a regular value of $\boldsymbol{f}$. Let $C \subset Z_{\boldsymbol{f}}$ be an connected open set in $Z_{\boldsymbol{f}}$. The curve $C$ is said to be $x$-monotonic if for any point $(a, \boldsymbol{b}) \in C$, we have $\det(J_{\boldsymbol{f},\boldsymbol{y}}(a, \boldsymbol{b})) \neq 0$, where $J_{\boldsymbol{f},\boldsymbol{y}}(a, \boldsymbol{b})$ is a Jacobian of $f$ with respect to $\boldsymbol{y}$ evaluated at $(a, \boldsymbol{b})$, defined as*

$$J_{f,\boldsymbol{y}}(a, \boldsymbol{b}) = \left[ \frac{\partial f_i}{\partial y_j}(a, \boldsymbol{b}) \right]_{m \times m}.$$

A key property of an $x$-monotonic curve $C$ is that for any $x_0$, there exists at most one $\boldsymbol{y}$ such that $(x_0, \boldsymbol{y}) \in C$. We will formalize this claim in Lemma F.19, but first, we will review some fundamental results necessary for the proof.

**Theorem F.17** (Implicit function theorem, [81]). *Let $\boldsymbol{f}$*

$$\boldsymbol{f} : \mathbb{R}^n \times \mathbb{R}^m \to \mathbb{R}^m$$
$$(\boldsymbol{x}, \boldsymbol{y}) \mapsto (f_1(\boldsymbol{x}, \boldsymbol{y}), \dots, f_m(\boldsymbol{x}, \boldsymbol{y})),$$

*be a continuously differentiable function. Consider a point $(\boldsymbol{a}, \boldsymbol{b}) \in \mathbb{R}^n \times \mathbb{R}^m$ such that $\boldsymbol{f}(\boldsymbol{a}, \boldsymbol{b}) = \mathbf{0}$ and the Jacobian*

$$J_{\boldsymbol{f},\boldsymbol{y}} = \left[ \frac{\partial f_i}{\partial y_j}(\boldsymbol{a}, \boldsymbol{b}) \right]_{m \times m}$$

*is invertible, then there exists a neighborhood $U$ of $\boldsymbol{a}$ in $\mathbb{R}^n$ and a neighborhood $V$ of $\boldsymbol{b}$ in $\mathbb{R}^m$, such that there exists a unique function $\boldsymbol{g} : U \to V$ such that $\boldsymbol{g}(\boldsymbol{a}) = \boldsymbol{b}$ and $f(\boldsymbol{x}, \boldsymbol{g}(\boldsymbol{x})) = 0$ for all $\boldsymbol{x} \in U$. We can also say that for $(\boldsymbol{x}, \boldsymbol{y}) \in U \times V$, we have $\boldsymbol{y} = g(\boldsymbol{x})$. Moreover, g is continuously differentiable and, if we denote*

$$J_{f,\boldsymbol{x}}(\boldsymbol{a}, \boldsymbol{b}) = \left[ \frac{\partial f_i}{\partial x_j}(\boldsymbol{a}, \boldsymbol{b}) \right]_{m \times n}$$

*then*

$$\left[ \frac{\partial g_i}{\partial x_j}(\boldsymbol{x}) \right]_{m \times n} = - \left[ J_{f,\boldsymbol{y}}(\boldsymbol{x}, g(\boldsymbol{x})) \right]_{m \times m}^{-1} \cdot \left[ J_{f,\boldsymbol{x}}(\boldsymbol{x}, g(\boldsymbol{x})) \right]_{m \times n}.$$

**Theorem F.18** (Vector-valued mean value theorem). *Let $S \subseteq \mathbb{R}^n$ be an open subset on $\mathbb{R}^n$ and let $\boldsymbol{f} : S \to \mathbb{R}^m$ be a continuously differentiable function. Consider $\boldsymbol{x}, \boldsymbol{y} \in S$ such that the line segment connecting these two points is contained in $S$, i.e. $L(\boldsymbol{x}, \boldsymbol{y}) \subset S$, where $L(\boldsymbol{x}, \boldsymbol{y}) = \{ t\boldsymbol{x} + (1-t)\boldsymbol{y} \mid t \in [0, 1] \}$. Then for every $\boldsymbol{a} \in \mathbb{R}^m$, there exists a point $\boldsymbol{z} \in L(\boldsymbol{x}, \boldsymbol{y})$ such that $\langle \boldsymbol{a}, \boldsymbol{f}(\boldsymbol{y}) - \boldsymbol{f}(\boldsymbol{x}) \rangle = \langle \boldsymbol{a}, J_{\boldsymbol{f},\boldsymbol{x}}(\boldsymbol{z})(\boldsymbol{y} - \boldsymbol{x}) \rangle$.*

We are now ready to present a formal statement and proof for the key property of $x$-monotonic curves, which essentially says that for a monotonic curve $C$ any $x_0$, there is at most one $\boldsymbol{y}$ such that $(x_0, \boldsymbol{y}) \in C$.

**Lemma F.19.** *Let $\boldsymbol{f} : \mathbb{R} \times \mathbb{R}^m \to \mathbb{R}^m$ be a polynomial map, and assume that $\mathbf{0} \in \mathbb{R}^m$ is a regular value of $\boldsymbol{f}$. Let $\boldsymbol{f}$ be a monotonic curve in $Z_{\boldsymbol{f}}$. Then for any $x_0 \in \mathbb{R}$, there is at most one point $\boldsymbol{y} \in \mathbb{R}^m$ such that $(x_0, \boldsymbol{y}) \in C$.*

*Proof.* (of Lemma F.19) Since $\mathbf{0} \in \mathbb{R}^m$ is a regular value of $\boldsymbol{f}$, then $Z_{\boldsymbol{f}}$ defines a smooth 1-dimensional manifold in $\mathbb{R} \times \mathbb{R}^m$. Therefore, $C$ is a connected open subset of an 1-dimensional smooth manifold $V_{\boldsymbol{f}}$ means that $C$ is diffeomorphic to $(0, 1)$. This means there exists a continuously differentiable function $\boldsymbol{h}$, where

$$\boldsymbol{h} : (0, 1) \to C$$
$$t \mapsto (x, \boldsymbol{y}) = (h_0(t), h_1(t), \dots, h_m(t)) \in C,$$

with corresponding inverse function $\boldsymbol{h}^{-1} : C \to (0, 1)$ which is also continuously differentiable.

We will prove the statement by contradiction. Assume that there exists $(x_0, \boldsymbol{y_1}), (x_0, \boldsymbol{y_2}) \in C$ where $\boldsymbol{y_1} \neq \boldsymbol{y_2}$. Then we have two corresponding values $t_1 = \boldsymbol{h}^{-1}(x_0, \boldsymbol{y_1}) \neq t_2 = \boldsymbol{h}^{-1}(x_0, \boldsymbol{y_2})$. Using

Mean-value Theorem (Theorem F.18) for the function $\boldsymbol{h}$, for any $\boldsymbol{a} \in \mathbb{R}^{m+1}$, there exists $z_{\boldsymbol{a}} \in (0, 1)$ such that
$$\langle \boldsymbol{a}, (0, \Delta \boldsymbol{y}) \rangle = \langle \boldsymbol{a}, \Delta t J_{h,t}(z_a) \rangle,$$
where $\Delta \boldsymbol{y} = \boldsymbol{y_2} - \boldsymbol{y_1} \neq \boldsymbol{0}$, $\Delta t = t_2 - t_1 \neq 0$, and $J_{h,t}(z_a) = (\frac{\partial h_0}{\partial t}(z_{\boldsymbol{a}}), \frac{\partial h_1}{\partial t}(z_{\boldsymbol{a}}), \ldots, \frac{\partial h_m}{\partial t}(z_{\boldsymbol{a}}))$.

Choose $\boldsymbol{a} = \boldsymbol{a_1} = (1, 0, \ldots, 0) \in \mathbb{R} \times \mathbb{R}^m$, then from above, there exists $z_{\boldsymbol{a}_1} \in (0, 1)$ such that $\frac{\partial h_0}{\partial t}\big|_{t=z_{\boldsymbol{a}_1}} = 0$. Now, consider the point $(x_{\boldsymbol{a}_1}, \boldsymbol{y}_{\boldsymbol{a}_1}) = h(z_{\boldsymbol{a}_1})$. From the assumption, $\det(J_{f,\boldsymbol{y}}(x_{\boldsymbol{a}_1}, \boldsymbol{y}_{\boldsymbol{a}_1})) \neq 0$. Therefore, from Implicit Function Theorem (Theorem F.17), there exists neighborhoods $U$ of $x_{\boldsymbol{a}_1}$ in $\mathbb{R}$, $V$ of $\boldsymbol{y}_{\boldsymbol{a}_1}$ in $\mathbb{R}^m$, such that there exists a continuously differentiable function $\boldsymbol{g} : U \to \mathbb{R}^m$, such that for any $(x, \boldsymbol{y}) \in U \times V$, we have $\boldsymbol{y} = g(x)$. Again, at the point $(x_{\boldsymbol{a}_1}, \boldsymbol{y}_{\boldsymbol{a}_1})$ corresponding to $t = z_{\boldsymbol{a}_1}$, we have
$$\frac{\partial y_i}{\partial t}\bigg|_{t=z_{\boldsymbol{a}_1}} = \frac{\partial g_i}{\partial x} \cdot \frac{\partial x}{\partial t}\bigg|_{t=z_{\boldsymbol{a}_1}} = 0.$$

This means that at the point $t = z_{\boldsymbol{a}_1}$, we have $\frac{\partial x}{\partial t}\big|_{t=z_{\boldsymbol{a}_1}} = \frac{\partial y_i}{\partial t}\big|_{t=z_{\boldsymbol{a}_1}} = 0$.

Note that since $\boldsymbol{h}$ is a diffeomorphism, we have $t = (\boldsymbol{h}^{-1} \circ \boldsymbol{h})(t)$. From chain rule, we have $1 = J_{\boldsymbol{h}^{-1},\boldsymbol{h}} \cdot J_{h,t}$. However, if we let $t = z_{\boldsymbol{a}_1}$, then $J_{h,t}(\boldsymbol{a}_1) = 0$, meaning that $J_{\boldsymbol{h}^{-1},\boldsymbol{h}} \cdot J_{h,t}(z_{\boldsymbol{a}_1}) = 0$, leading to a contradiction. $\qquad\square$

From Definition 20 and Proposition F.19, for each $x$-monotonic curve $C$, we can define their $x$-end points, which are the maximum and minimum of $x$-coordinate that a point in $C$ can have.

**Definition 20** ($x$-end points of a monotonic curve)**.** *Let $C$ be a monotonic curve as defined in Definition 19. Then we call $\sup\{x \mid \exists \boldsymbol{y}, (x, \boldsymbol{y}) \in V\}$ and $\inf\{x \mid \exists \boldsymbol{y}, (x, \boldsymbol{y}) \in V\}$ the $x$-end points of $C$.*

We now show that the pointwise maximum of continuous functions along monotonic curves is also continuous. We then relate the local maxima of the pointwise maximum with the local maxima of continuous functions along the monotonic curves in Proposition F.21.

**Proposition F.20.** *Let $M \subset \mathbb{R}^n$ be a topological manifold, and let $S$ be an open subset in $M$. Let $p$ be a point in $S$, and assume that $V$ is a neighborhood of $x$ in $S$. Then $V$ is also a neighborhood of $p$ in $M$.*

*Proof.* First, note that since $V$ is a neighborhood of $p$ in $S$, $V$ is an open set in the subspace topology $S$, meaning that there exists an open set $T$ in $M$ such that $V = S \cap T$. However, note that both $S$ and $T$ are open sets in $M$, which implies $V$ is also an open set in $M$. And since $V$ contains $p$, we have that $V$ is a neighborhood of $p$ in $M$. $\qquad\square$

**Proposition F.21.** *Let $\mathcal{C} = \{C_1, \ldots, C_k\}$ be a set of $k$ $x$-monotonic curves as defined in Definition 19 that have $x_1, x_2$ as $x$-end points. Consider a smooth function $g : \mathbb{R} \times \mathbb{R}^m \to \mathbb{R}$, and let $h : (x_1, x_2) \to \mathbb{R}$ defined as*
$$h(x) = \max_{i=1,\ldots,k} g(I_{C_i}(x)),$$
*where $I_{C_i} : (x_1, x_2) \to \mathbb{R} \times \mathbb{R}^m$ maps $x$ to the point $I_{C_i}(x) = (x, \boldsymbol{y}_x) \in C_i$ . Then $h(x)$ is continuous over $(x_1, x_2)$, and for any local maximum $x'$ of $h(x)$, there exist a point $(x', \boldsymbol{y}_{x'})$ that is a local maximum of the function $g(x, \boldsymbol{y})$ restricted to some monotonic curve $C \in \mathcal{C}$. Moreover, if $h$ is strictly monotonically decreasing (resp. strictly monotonically increasing, constant) at $x' \in (x_1, x_2)$, then $h(x') = g(I_{C_i}(x'))$ for some $i$ such that $g \circ I_{C_i}$ is strictly monotonically decreasing (resp. strictly monotonically increasing, constant).*

*Proof.* From the property of monotonic curves, it is easy to show that $I_{C_i}$ is a diffeomorphism between $(x_1, x_2)$ and $C_i$. Therefore, $g \circ I_{C_i} : (x_1, x_2) \to \mathbb{R}$ is a continuous function. This implies that $h$ is the pointwise maximum of continuous functions, and hence is also continuous.

Now consider any monotonic curve $C \in \mathcal{C}$. Assume $x'$ is a local maximum of $g \circ I_C$ in $(x_1, x_2)$. By definition, there exists an open neighborhood $V$ of $x'$ in $(x_1, x_2)$ such that for any $x \in V$,

$g(I_C(x')) \geq g(I_C(x))$. Since $I_C$ is a diffeomorphism between $(x_1, x_2)$ and $C$, this implies $I_C(V)$ is also an open set in $V$ that contains $I_C(x')$. Now for any $(x, \boldsymbol{y}_x) \in I_C(V)$ where $\boldsymbol{y}_x$ is the unique value corresponding to $x$ in $C$, we have $g(x, y_x) = g(I_C(x)) \leq g(I_C(x')) = g(x', \boldsymbol{y}_{x'})$. This means $(x', \boldsymbol{y}_{x'})$ is a local maximum of $g$ in $C$.

Finally, it suffices to give a proof for the case of $k = 2$. Let $h(x) = \max\{g(I_{C_1}(x)), g(I_{C_2}(x))\}$. We claim that any local maximum of $h$ would be a local maximum of either $g \circ I_{C_1}$ or $g \circ I_{C_2}$ in $(x_1, x_2)$. Assume that $x'$ is a local maximum of $h$ in $(x_1, x_2)$, then there exists an open neighbor $V$ of $x'$ in $(\alpha_1, \alpha_2)$ such that for any $x \in V$, $h(x) \leq h(x')$. WLOG, assume that $h(x') = g(I_{C_1}(x'))$. Then we have:

$$g(I_{C_1}(x')) = h(x') \geq h(x) = \max\{g(I_{C_1}(x)), g(I_{C_2}(x))\} \geq g(I_{C_1}(x)),$$

for any $x \in V$. This means that $x'$ is a local maximum of $g \circ I_{C_1}$ in $(\alpha_1, \alpha_2)$. Combining with the above, we also have $(x', \boldsymbol{y}_{x'})$ is the local maximum of $g$ in $C_1$.

Similarly, suppose that $h$ is strictly monotonically decreasing at $x'$. Then for a sufficiently small left-neighborhood $L = (x' - \varepsilon, x')$, we must have that $h$ is strictly monotonically decreasing over $L$ and $h(x) = g(I_{C_i}(x))$ for all $x \in L$ for some fixed $i$. Moreover, $g \circ I_{C_i}$ must be strictly monotonically decreasing over sufficiently small $R = (x', x' + \varepsilon')$, as $g(I_{C_i}(x)) \leq h(x)$ over $R$ and $h$ is strictly monotonically decreasing for sufficiently small $R$. $\qquad\square$

**Proposition F.22.** *Let $\mathcal{C} = \{C_1, \ldots, C_k\}$ be a set of $k$ x-monotonic curves (Definition 19) that have $x_1, x_2$ as x-end points. Consider a smooth function $g : \mathbb{R} \times \mathbb{R}^m \to \mathbb{R}$. Define $g_i : \mathbb{R} \to \mathbb{R}$ as $g_i(x) = g(I_{C_i}(x))$, where $I_{C_i} : (x_1, x_2) \to \mathbb{R} \times \mathbb{R}^m$ maps $x$ to the point $I_{C_i}(x) = (x, \boldsymbol{y}_x) \in C_i$. Let $h : (x_1, x_2) \to \mathbb{R}$ be given as*
$$h(x) = \max_{i=1,\ldots,k} g_i(x).$$

*Then if each $g_i$ is $B_i$-monotonic (Definition 5), then $h$ is $O(\sum_{i=1}^{k} B_i)$-monotonic.*

*Proof.* Let $A_i$ denote the finite set (of smallest size) of critical points for $g_i$, such that between any two consecutive points in $A_i$, $g_i$ is either strictly monotonic or a constant. By Definition 5 and the assumption that $g_i$ is $B_i$-monotonic, we have that $|A_i| = O(B_i)$. Let $A = \cup_i A_i$. Clearly, $|A| \leq \sum_i |A_i| = O(\sum_{i=1}^{k} B_i)$. Consider any two consecutive points $a_1, a_2$ in $A$. We claim that $h$ is 3-monotonic over $(a_1, a_2)$.

To establish this claim, we consider two cases for any point $a \in (a_1, a_2)$.

(i) Case: $h(a) = g_i(a)$ and $g_i$ is strictly monotonically decreasing at $a$. We claim that for any $a' \in (a_1, a)$, $h(a')$ is strictly monotonically decreasing. If not, then $h(a') = g_j(a')$ for some function $g_j$ that is either constant or monotonically increasing at $a'$ (since there are no critical points for $g_j$ in $(a_1, a_2)$). But this contradicts $h(a) = g_i(a)$ as $g_i(a) < g_i(a') \leq g_j(a') \leq g_j(a) \leq h(a)$.

(ii) Case: $h(a) = g_i(a)$ and $g_i$ is strictly monotonically increasing at $a$. By inverting the argument for Case (i) above, for any $a' \in (a, a_2)$, $h(a')$ is strictly monotonically increasing.

Let $a_1' = \sup\{a \in (a_1, a_2) \mid h$ is strictly monotonically decreasing at $a\}$ and $a_2' = \inf\{a \in (a_1, a_2) \mid h$ is strictly monotonically increasing at $a\}$. By cases (i) and (ii) above, and using Proposition F.21, we have $a_1 \leq a_1' \leq a_2' \leq a_2$. Thus, $h$ is 3-monotonic over $(a_1, a_2)$ as it must be strictly decreasing over $(a_1, a_1')$, constant over $(a_1', a_2')$ and strictly increasing over $(a_2', a_2)$. Therefore, $h$ is $3|A|$-monotonic or equivalently $O(\sum_{i=1}^{k} B_i)$-monotonic over the entire domain. $\qquad\square$

## F.4 Simpler illustrative case: hyperparameter tuning with a single parameter

We provide some intuition for our novel proof techniques by first considering a simpler setting. We first consider the case where there is a single parameter and only one piece function. That is, we assume that $d = 1$, $N = 1$, and $M = 0$. Since there is only one piece in this case, we abuse the notations and use interchangeably $u_{\boldsymbol{x}}^*$ and $f_{\boldsymbol{x}}$ for $u_{\boldsymbol{x},1}^*$ and $f_{\boldsymbol{x},1}$, respectively. This means $f_{\boldsymbol{x}}$ is now a polynomial function of $\alpha, \boldsymbol{w}$ of degree at most $\Delta_p$, instead of admitting a piecewise polynomial structure.

We first present a structural result for the dual function class $\mathcal{U}^*$, which establishes that any function $u_{\boldsymbol{x}}^*$ in the dual utility function class $\mathcal{U}^*$ is piecewise continuous with $O(\Delta_p^2)$ pieces. Furthermore,

we show that there are $O(\Delta_p^3)$ oscillations in $u_{\boldsymbol{x}}^*$ which implies a bound on the pseudo-dimension of $\mathcal{U}^*$ using results in Section 2.1.

Our proof approach is summarized as follows. We note that the supremum over $\boldsymbol{w} \in \mathcal{W}$ in the definition of $u_{\boldsymbol{x}}^*$ can only be achieved at a domain boundary or at a point $(\alpha, \boldsymbol{w})$ that satisfies $k_{\boldsymbol{x}}(\alpha, \boldsymbol{w}) = \frac{\partial f_{\boldsymbol{x}}(\alpha, \boldsymbol{w})}{\partial \boldsymbol{w}} = 0$, which defines an algebraic curve. We partition this algebraic curve into *monotonic arcs* [82, 83], which intersect $\alpha = \alpha_0$ at most once for any $\alpha_0$. Intuitively, a point of discontinuity of $u_{\boldsymbol{x}}^*$ can only occur when the set of monotonic arcs corresponding to a fixed value of $\alpha$ changes as $\alpha$ is varied, which corresponds to $\alpha$-extreme points of the monotonic arcs. We use a direct consequence of Bezout's theorem (Corollary F.10) to upper bound these extreme points of $k_{\boldsymbol{x}}(\alpha, \boldsymbol{w}) = 0$ to obtain an upper bound on the number of pieces of $u_{\boldsymbol{x}}^*$. Next, we seek to upper bound the number of local extrema of $u_{\boldsymbol{x}}^*$ to bound its oscillating behavior within the continuous pieces. To this end, we need to examine the behavior of $u_{\boldsymbol{x}}^*$ along the algebraic curve $h_{\boldsymbol{x}}(\alpha, \boldsymbol{w}) = 0$ and use the Lagrange's multiplier theorem (Theorem F.3) to express the locations of the extrema as intersections of algebraic varieties (in $\alpha, \boldsymbol{w}$ and the Lagrange multiplier $\lambda$). Another application of Bezout's theorem (Corollary F.10) gives us the desired upper bound on the number of local extrema of $u_{\boldsymbol{x}}^*$.

**Lemma F.23.** *Let $d = 1$, $N = 1$, $M = 0$. That is, there is a single parameter, a single piece function, and no boundary functions. Assume that $\mathcal{A} = [\alpha_{\min}, \alpha_{\max}]$ and $\mathcal{W} = [w_{\min}, w_{\max}]$. Then*

    *(a) The hyperparameter domain $\mathcal{A}$ can be partitioned into $\mathcal{O}(\Delta_p^2)$ intervals such that $u_{\boldsymbol{x}}^*$ is a continuous function over any interval in the partition.*

    *(b) $u_{\boldsymbol{x}}^*$ is $\mathcal{O}(\Delta_p^2)$-monotonic (Definition 5).*

*Proof.* (a) Denote $k_{\boldsymbol{x}}(\alpha, w) = \frac{\partial f_{\boldsymbol{x}}(\alpha, w)}{\partial w}$. From assumption, $f_{\boldsymbol{x}}(\alpha, w)$ is a polynomial of $\alpha$ and $w$, therefore it is differentiable everywhere in the compact domain $[\alpha_{\min}, \alpha_{\max}] \times [w_{\min}, w_{\max}]$. Consider any $\alpha_0 \in [\alpha_{\min}, \alpha_{\max}]$, we have $\{(\alpha, w) \mid \alpha = \alpha_0\} \cap ([\alpha_{\min}, \alpha_{\max}] \times [w_{\min}, w_{\max}])$ is an intersection of a hyperplane and a compact set, hence it is also compact. Thus Fermat's interior extremum theorem (Lemma F.4) applies and, for any $\alpha_0$, $f_{\boldsymbol{x}}(\alpha_0, w)$ attains the local maxima $w$ either in $w_{\min}, w_{\max}$, or for $w \in (w_{\min}, w_{\max})$ such that $k_{\boldsymbol{x}}(\alpha_0, w) = 0$. Note that from assumption, $f_{\boldsymbol{x}}(\alpha, w)$ is a polynomial of degree at most $\Delta_p$ in $\alpha$ and $w$. This implies $k_{\boldsymbol{x}}(\alpha, w)$ is a polynomial of degree at most $\Delta_p - 1$.

Denote $C_{\boldsymbol{x}} = V(k_{\boldsymbol{x}})$ the zero set of $k_{\boldsymbol{x}}$ in $R := \mathcal{A} \times \mathcal{W}$. If $C_{\boldsymbol{x}}$ has any components consisting of axis-parallel straight lines $\alpha = \alpha_1$, we include the corresponding discontinuities (later) via the intersections of $C_{\boldsymbol{x}}$ with the boundary lines $\boldsymbol{w} = w_{\min}, w_{\max}$. Therefore, assume that $C_{\boldsymbol{x}}$ does not have such components. For any $\alpha_0$, $C_{\boldsymbol{x}}$ intersects the line $\alpha = \alpha_0$ in at most $\Delta_p - 1$ points by Bezout's theorem (Corollary F.11). This implies that, for any $\alpha$, there are at most $\Delta_p + 1$ candidate values of $w$ which can possibly maximize $f_{\boldsymbol{x}}(\alpha, w)$, which can be either $w_{\min}, w_{\max}$, or on some point in $C_{\boldsymbol{x}}$. We can decompose $C_{\boldsymbol{x}}$ into *monotonic arcs* using well-known techniques from algebraic geometry [82, 83]. We then define the *candidate arc set* $\mathcal{C} : \mathcal{A} \to \mathcal{M}_\alpha(C_{\boldsymbol{x}})$ as the function that maps $\alpha_0 \in \mathcal{A}$ to the set of all maximal $\alpha$-monotonic arcs of $C_{\boldsymbol{x}}$ (Definition 19, informally arcs that intersect any line $\alpha = \alpha_0$ at most once) that intersect with $\alpha = \alpha_0$. By the argument above, we have $|\mathcal{C}(\alpha)| \leq \Delta_p + 1$ for any $\alpha$.

We now have the following claims: (1) $\mathcal{C}$ is a piecewise constant function, and (2) any point of discontinuity of $u_{\boldsymbol{x}}^*$ must be a point of discontinuity of $\mathcal{C}$. For (1), we will show that $\mathcal{C}$ is piecewise constant, with the piece boundaries contained in the set of $\alpha$-extreme points[4] of $C_{\boldsymbol{x}}$ and the intersection points of $C_{\boldsymbol{x}}$ with boundary lines $\boldsymbol{w} = w_{\min}, w_{\max}$. Indeed, for any interval $I = (\alpha_1, \alpha_2) \subseteq \mathcal{A}$, if there is no $\alpha$-extreme point of $C_{\boldsymbol{x}}$ in the interval, then the set of arcs $\mathcal{C}(\alpha)$ is fixed over $I$ by Definition 19. Next, we will prove (2) via an equivalent statement: assume that $\mathcal{C}$ is continuous over an interval $I \subseteq \mathcal{A}$, we want to prove that $u_{\boldsymbol{x}}^*$ is also continuous over $I$. Note that if $\mathcal{C}$ is continuous over $I$, then $u_{\boldsymbol{x}}^*(\alpha)$ involves a maximum over a fixed set of $\alpha$-monotonic arcs of $C_{\boldsymbol{x}}$, and the straight lines $w = w_{\min}, w = w_{\max}$. Since $f_{\boldsymbol{x}}$ is continuous along these arcs, so is the maximum $u_{\boldsymbol{x}}^*$ (Lemma F.1).

The above claim implies that the number of discontinuity points of $\mathcal{C}$ upper-bounds the number of discontinuity points of $u_{\boldsymbol{x}}^*(\alpha)$. Using a well-known result for algebraic curves (e.g. [84]), the number

---

[4]An $\alpha$-extreme point of an algebraic curve $C$ is a point $p = (\alpha, W)$ such that there is an open neighborhood $N$ around $p$ for which $p$ has the smallest or largest $\alpha$-coordinate among all points $p' \in N$ on the curve.

of $\alpha$-extreme points is $\mathcal{O}(\Delta_p^2)$. Moreover, there are $\mathcal{O}(\Delta_p)$ intersection points between $C_{\boldsymbol{x}}$ and the boundary lines $\boldsymbol{w} = w_{\min}$ and $\boldsymbol{w} = w_{\max}$. Thus, the total discontinuities of $\mathcal{C}$ are $\mathcal{O}(\Delta_p^2)$, giving an upper bound on the number of discontinuity points of $u_{\boldsymbol{x}}^*$.

(b) Consider any interval $I \subset \mathcal{A}$ over which the function $\mathcal{C}$ is continuous. In particular, this means that there is no $\alpha$-extreme point corresponding to any $\alpha \in I$. By Proposition F.2 and Proposition F.22, it suffices to bound the $B$-monotonicity of $f_{\boldsymbol{x}}$ along the algebraic curve $C_{\boldsymbol{x}}$ (i.e. along a fixed set of monotonic arcs of $C_{\boldsymbol{x}}$) and the straight lines $\boldsymbol{w} = w_{\min}$ and $\boldsymbol{w} = w_{\max}$.

To bound the number of elements of the set of local maxima of $f_{\boldsymbol{x}}$ along the algebraic curve $C_{\boldsymbol{x}}$, consider the Lagrangian
$$\mathcal{L}(\alpha, w, \lambda) = f_{\boldsymbol{x}}(\alpha, w) + \lambda k_{\boldsymbol{x}}(\alpha, w).$$
From the Lagrange's multiplier theorem, any local maxima of $f_{\boldsymbol{x}}$ along the algebraic curve $C_{\boldsymbol{x}}$ is also a critical point of $\mathcal{L}$, which satisfies the following equations

$$\frac{\partial \mathcal{L}}{\partial \alpha} = \frac{\partial f_{\boldsymbol{x}}}{\partial \alpha} + \lambda \frac{\partial k_{\boldsymbol{x}}}{\partial \alpha} = \frac{\partial f_{\boldsymbol{x}}}{\partial \alpha} + \lambda \frac{\partial^2 f_{\boldsymbol{x}}}{\partial \alpha \partial w} = 0,$$

$$\frac{\partial \mathcal{L}}{\partial w} = \frac{\partial f_{\boldsymbol{x}}}{\partial w} + \lambda \frac{\partial k_{\boldsymbol{x}}}{\partial w} = \frac{\partial f_{\boldsymbol{x}}}{\partial w} + \lambda \frac{\partial^2 f_{\boldsymbol{x}}}{\partial w^2} = 0,$$

$$\frac{\partial \mathcal{L}}{\partial \lambda} = k_{\boldsymbol{x}} = \frac{\partial f_{\boldsymbol{x}}}{\partial w} = 0.$$

Plugging $\frac{\partial f_{\boldsymbol{x}}}{\partial w} = 0$ into the second equation above, we get that either $\lambda = 0$ or $\frac{\partial^2 f_{\boldsymbol{x}}}{\partial w^2} = 0$. In the former case, the first equation implies $\frac{\partial f_{\boldsymbol{x}}}{\partial \alpha} = 0$. Thus, we consider two cases for critical points of $\mathcal{L}$. Let's suppose $(\tilde{\alpha}, \tilde{w})$ is a local maximum of $f_{\boldsymbol{x}}$ along $C_{\boldsymbol{x}}$. There must exist a $\lambda$ such that $(\tilde{\alpha}, \tilde{w}, \lambda)$ satisfies the Lagrangian equations above.

**Case $\lambda = 0$.** $\frac{\partial f_{\boldsymbol{x}}(\tilde{\alpha},\tilde{w})}{\partial \boldsymbol{w}} = 0$, $\frac{\partial f_{\boldsymbol{x}}(\tilde{\alpha},\tilde{w})}{\partial \alpha} = 0$. For this to be true, $\tilde{\alpha}, \tilde{w}$ must be at the intersection of the algebraic curves $\frac{\partial f_{\boldsymbol{x}}(\alpha,w)}{\partial \boldsymbol{w}} = 0$, $\frac{\partial f_{\boldsymbol{x}}(\alpha,w)}{\partial \alpha} = 0$. By Bezout's theorem on the plane (Corollary F.11), these algebraic curves intersect in at most $\Delta_p^2$ points, unless the polynomials $\frac{\partial f_{\boldsymbol{x}}(\alpha,w)}{\partial w}$, $\frac{\partial f_{\boldsymbol{x}}(\alpha,w)}{\partial \alpha}$ have a common factor. In this case, we can write $\frac{\partial f_{\boldsymbol{x}}(\alpha,w)}{\partial w} = g(\alpha, w)g_1(\alpha, w)$ and $\frac{\partial f_{\boldsymbol{x}}(\alpha,w)}{\partial \alpha} = g(\alpha, w)g_2(\alpha, w)$ where $g = \gcd\left(\frac{\partial f_{\boldsymbol{x}}}{\partial w}, \frac{\partial f_{\boldsymbol{x}}}{\partial \alpha}\right)$ and $g_1, g_2$ have no common factors. Now for any point $(\tilde{\alpha}, \tilde{w})$ on the curve $g(\alpha, w) = 0$, we have both $\frac{\partial f_{\boldsymbol{x}}(\tilde{\alpha},\tilde{w})}{\partial w} = 0, \frac{\partial f_{\boldsymbol{x}}(\tilde{\alpha},\tilde{w})}{\partial \alpha} = 0$. Therefore, for any two points $(\alpha_1, w_1)$ and $(\alpha_2, w_2)$ on the curve $g(\alpha, w) = 0$, $f_{\boldsymbol{x}}(\alpha_1, w_1) = f_{\boldsymbol{x}}(\alpha_2, w_2)$. Thus, $f_{\boldsymbol{x}}$ is constant along $g(\alpha, w) = 0$. Recall that we are computing the local maxima on an interval $I$, corresponding to a fixed set of monotonic arcs of $k_{\boldsymbol{x}}$ (and therefore also $g$, which is a factor of $k_{\boldsymbol{x}}$). By Bezout's theorem (Corollary F.11), $g_1(\alpha, w) = 0, g_2(\alpha, w) = 0$ intersect in at most $\deg(g_1)\deg(g_2) \leq \Delta_p^2$ points (since they do not have any common factors), which are the only other candidate points for which $(\tilde{\alpha}, \tilde{w}, 0)$ satisfies the Lagrangian. Thus, the number of local maxima of $u_{\boldsymbol{x}}^*$ that correspond to this case (i.e. across all monotonic arcs not corresponding to $g$) is $\mathcal{O}(\Delta_p^2)$.

**Case $\lambda \neq 0$.** $\frac{\partial f_{\boldsymbol{x}}(\tilde{\alpha},\tilde{w})}{\partial \boldsymbol{w}} = 0$, $\frac{\partial^2 f_{\boldsymbol{x}}(\tilde{\alpha},\tilde{w})}{\partial \boldsymbol{w}^2} = 0$. This essentially corresponds to the $\alpha$-extreme points computed above (see e.g. [85]), and do not occur over any interval $I$ considered here.

Similarly, the equations $f_{\boldsymbol{x}}(\alpha, w_{\min}) = 0$ and $f_{\boldsymbol{x}}(\alpha, w_{\max}) = 0$ also have at most $\Delta_p$ solutions each. Putting together, by Theorem F.22, we conclude that $u_{\boldsymbol{x}}^*$ is $\mathcal{O}(\Delta_p^2)$-monotonic. $\qquad\square$

**Theorem F.24.** $\mathrm{Pdim}(\mathcal{U}^*) = \mathcal{O}(\log \Delta_p)$.

*Proof.* From Theorem F.23 and Theorem D.3, we conclude that $u_{\boldsymbol{x}}^*$ has at most $\mathcal{O}(\Delta_p^2)$ oscillations for any $u_{\boldsymbol{x}}^* \in \mathcal{U}^*$. Therefore, using Theorem D.1, we conclude that $\mathrm{Pdim}(\mathcal{U}^*) = \mathcal{O}(\log \Delta_p)$. $\qquad\square$

**Remark 4.** ***Challenges of generalizing the one-dimensional parameter and single region setting above to high-dimensional parameters and multiple regions.*** *Recall that in the simple setting above, we assume that $f_{\boldsymbol{x}}(\alpha, \boldsymbol{w})$ is a polynomial in the whole domain $R := [\alpha_{\min}, \alpha_{\max}] \times [w_{\min}, w_{\max}]$. In this case, our approach is to characterize the manifold on which the optimal solution of $\max_{\boldsymbol{w}:(\alpha,\boldsymbol{w})\in R} f_{\boldsymbol{x}}(\alpha, \boldsymbol{w})$ lies, as $\alpha$ varies. We then use algebraic geometry tools to upper bound the number of discontinuity points and local extrema of $u_{\boldsymbol{x}}^*(\alpha) = \max_{\boldsymbol{w}:(\alpha,\boldsymbol{w})\in R} f_{\boldsymbol{x}}(\alpha, \boldsymbol{w})$, leading*

*to a bound on the pseudo-dimension of the utility function class $\mathcal{U}$ by using our proposed tools in Section 2.1. However, to generalize this idea to high-dimensional parameters and multiple regions is much more challenging due to the following issues: (1) handling the analysis of multiple pieces while accounting for polynomial boundary functions is tricky as the $\boldsymbol{w}^*$ maximizing $f_{\boldsymbol{x}}(\alpha, \boldsymbol{w})$ can switch between pieces as $\alpha$ is varied, (2) characterizing the optimal solution $\max_{\boldsymbol{w}:(\alpha, \boldsymbol{w}) \in R} f_{\boldsymbol{x}}(\alpha, \boldsymbol{w})$ is not trivial and typically requires additional assumptions to ensure that a general position property is achieved, and care needs to be taken to make sure that the assumptions are not too strong, (3) generalizing the monotonic curve notion to high-dimensions is not trivial and requires a much more complicated analysis invoking tools from differential geometry, and (4) controlling the number of discontinuities and local maxima of $u_{\boldsymbol{x}}^*$ over the high-dimensional monotonic curves requires more sophisticated techniques.*

### F.5    Omitted proofs for Theorem 4.1

In this section, we will present a detailed proof for Theorem 4.1.

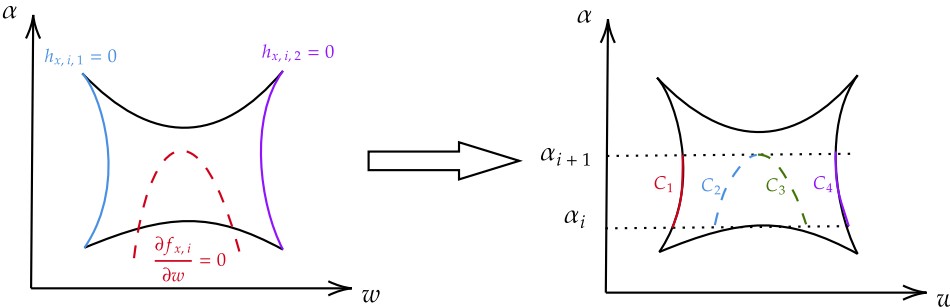

Figure 4: A simplified illustration for the proof idea of Theorem 4.1 where $\boldsymbol{w} \in \mathbb{R}$. Here, our goal is to analyze the number of discontinuities and local maxima of $u_{\boldsymbol{x},i}^*(\alpha)$. The idea is to partition the hyperparameter space $\mathcal{A}$ into intervals such that over each interval, the function $u_{\boldsymbol{x},i}^*(\alpha)$ is the pointwise maximum of $f_{\boldsymbol{x},i}(\alpha, \boldsymbol{w})$ along some fixed set of "monotonic curves" $\mathcal{C}$ (curves that intersect $\alpha = \alpha_0$ at most once for any $\alpha_0$). $u_{\boldsymbol{x},i}^*(\alpha)$ is continuous over such interval; this implies that the interval endpoints contain all discontinuities of $u_{\boldsymbol{x},i}^*(\alpha)$. In this example, over the interval $(\alpha_i, \alpha_{i+1})$, we have $u_{\boldsymbol{x},i}^*(\alpha) = \max_{C_i}\{f_{\boldsymbol{x},i}(\alpha, \boldsymbol{w}) : (\alpha, \boldsymbol{w}) \in C_i\}$. Then, we can show that over such an interval, any local maximum of $u^{\boldsymbol{x},i}(\alpha)$ is a local extremum of $f_{\boldsymbol{x},i}(\alpha, \boldsymbol{w})$ along a monotonic curve $C \in \mathcal{C}$. Finally, we bound the number of points used for partitioning and local extrema using tools from algebraic and differential geometry.

**Theorem 4.1 (restated).** *Consider the utility function class $\mathcal{U} = \{u_\alpha : \mathcal{X} \to [0, H] \mid \alpha \in \mathcal{A}\}$. Assume that the parameter-dependent dual function $f_{\boldsymbol{x}}(\alpha, \boldsymbol{w})$ admits piecewise polynomial structure with the piece functions $f_{\boldsymbol{x},i}$ and boundaries $h_{\boldsymbol{x},i}$ satisfying Assumption 1. Then for any distribution $\mathcal{D}$ over $\mathcal{X}$, for any $\delta \in (0, 1)$, with probability at least $1 - \delta$ over the draw of $S \sim \mathcal{D}^m$, we have*

$$|\mathbb{E}_{\boldsymbol{x} \sim \mathcal{D}}[u_{\hat{\alpha}_{ERM}}(\boldsymbol{x})] - \mathbb{E}_{\boldsymbol{x} \sim \mathcal{D}}[u_{\alpha^*}(\boldsymbol{x})]| = \mathcal{O}\left(\sqrt{\frac{\log N + d\log(\Delta M) + \log(1/\delta)}{m}}\right).$$

*Here, $M$ and $N$ are the number of boundaries and connected sets, $\Delta = \max\{\delta_p, \delta_d\}$ is the maximum degree of piece functions $f_{\boldsymbol{x},i}$ and boundaries $h_{\boldsymbol{x},i}$.*

*Proof.* In this section, we will first go through the general idea of the proof for Theorem 4.1, and its main steps. The proof consists of three main steps. **Step 1**: we first bound the number of possible discontinuities and local extrema of $v_{\boldsymbol{x}}^*(\alpha)$ under Assumption 3, which is a stronger assumption compared to Assumption 1. **Step 2**: we show that, for any $u_{\boldsymbol{x}}^*(\alpha)$ that satisfies Assumption 1, we can construct $v_{\boldsymbol{x}}^*(\alpha)$ satisfies Assumption 3 and is arbitrarily close to $u_{\boldsymbol{x}}^*(\alpha)$. **Step 3**: we can use this property to recover the learning guarantee for $\mathcal{U}$. The key ideas of each step are sketched below.

**First step: the proof requiring a stronger assumption**

To begin with, we first start by stating the following stronger assumption compared to Assumption 1.

**Assumption 3** (Regularity assumption)**.** *Assume that for any dual utility function $u_{\boldsymbol{x}}^*(\alpha)$, its piece functions $f_{\boldsymbol{x},i}(\alpha, \boldsymbol{w})$ (for $i = 1, \ldots, N$) and boundary functions $h_{\boldsymbol{x},j}(\alpha, \boldsymbol{w})$ (for $j = 1, \ldots, M$) satisfy the following property. For any piece function $f$ chosen from $\{f_{\boldsymbol{x},1}, \ldots, f_{\boldsymbol{x},N}\}$ and $S$ boundary functions $h_1, \ldots, h_S$ chosen from $\{h_{\boldsymbol{x},1}, \ldots, h_{\boldsymbol{x},M}\}$, the following conditions are met:*

1. *Consider the polynomial map $\boldsymbol{h}$, where*

$$
\begin{aligned}
\boldsymbol{h} : \mathbb{R}^{d+1} &\to \mathbb{R}^S \\
\boldsymbol{h}(\alpha, \boldsymbol{w}) &\mapsto (h_1(\alpha, \boldsymbol{w}), \ldots, h_S(\alpha, \boldsymbol{w})).
\end{aligned}
$$

   *Then $\boldsymbol{0} \in \mathbb{R}^S$ is a regular value of $\boldsymbol{h}$. Furthermore, if $S \le d$, we have $J_{\boldsymbol{h}}(\boldsymbol{w})$ has full row rank for $(\alpha, \boldsymbol{w}) \in \boldsymbol{h}^{-1}(\boldsymbol{0})$.*

2. *Consider the polynomial map $\overline{\boldsymbol{k}}$, where*

$$
\begin{aligned}
\overline{\boldsymbol{k}} : \mathbb{R}^{d+S+1} &\to \mathbb{R}^{d+S} \\
(\alpha, \boldsymbol{w}, \boldsymbol{\lambda}) &\mapsto (k_1(\alpha, \boldsymbol{w}, \boldsymbol{\lambda}), \ldots, k_{d+S}(\alpha, \boldsymbol{w}, \boldsymbol{\lambda})).
\end{aligned}
$$

   *Here, $S \le d$ and $\overline{\boldsymbol{k}}(\alpha, \boldsymbol{w}, \boldsymbol{\lambda}) = (k_1(\alpha, \boldsymbol{w}, \boldsymbol{\lambda}), \ldots, k_{d+S}(\alpha, \boldsymbol{w}, \boldsymbol{\lambda}))$ is defined as*

$$
\begin{aligned}
k_j(\alpha, \boldsymbol{w}, \boldsymbol{\lambda}) &= h_j(\alpha, \boldsymbol{w}), & j &= 1, \ldots, S, \\
k_{S+i}(\alpha, \boldsymbol{w}, \boldsymbol{\lambda}) &= \partial_{w_i} L(\alpha, \boldsymbol{w}, \boldsymbol{\lambda}), & i &= 1, \ldots, d,
\end{aligned}
$$

   *where $L(\alpha, \boldsymbol{w}, \boldsymbol{\lambda}) = f(\alpha, \boldsymbol{w}) + \boldsymbol{\lambda}^\top \boldsymbol{h}(\alpha, \boldsymbol{w})$. Then there exists a set of real values $\Xi$ consisting of at most $\Delta^{2S+2d}$ elements such that the Jacobian $J_{\overline{\boldsymbol{k}}}(\boldsymbol{w}, \boldsymbol{\lambda})$ has full row rank for all $(\alpha, \boldsymbol{w}, \boldsymbol{\lambda}) \in \overline{\boldsymbol{k}}^{-1}(\boldsymbol{0})$ such that $\alpha \notin \Xi$.*

3. *Consider the polynomial map $\boldsymbol{\mu}$*

$$
\begin{aligned}
\boldsymbol{\mu} : \mathbb{R}^{2d+2S+1} &\to \mathbb{R}^{2d+2S+1} \\
\boldsymbol{\mu}(\alpha, \boldsymbol{w}, \boldsymbol{\gamma}, \boldsymbol{\lambda}, \boldsymbol{\theta}) &\mapsto (\mu_1(\alpha, \boldsymbol{w}, \boldsymbol{\gamma}, \boldsymbol{\lambda}, \boldsymbol{\theta}), \ldots, \mu_{2d+2S+1}(\alpha, \boldsymbol{w}, \boldsymbol{\gamma}, \boldsymbol{\lambda}, \boldsymbol{\theta})).
\end{aligned}
$$

   *Here, $S \le d$, $\boldsymbol{\lambda} \in \mathbb{R}^S$, $\boldsymbol{\theta} \in \mathbb{R}^S$, $\boldsymbol{\gamma} \in \mathbb{R}^d$, and each $\mu_i$ is defined as follows:*

$$
\begin{cases}
\mu_j(\alpha, \boldsymbol{w}, \boldsymbol{\gamma}, \boldsymbol{\lambda}, \boldsymbol{\theta}) &= h_j(\alpha, \boldsymbol{w}), j = 1, \ldots, S \\
\mu_{S+j}(\alpha, \boldsymbol{w}, \boldsymbol{\gamma}, \boldsymbol{\lambda}, \boldsymbol{\theta}) &= \boldsymbol{\gamma}^\top \nabla_{\boldsymbol{w}} h_j(\alpha, \boldsymbol{w}), j = 1, \ldots, S \\
\mu_{2S+i}(\alpha, \boldsymbol{w}, \boldsymbol{\gamma}, \boldsymbol{\lambda}, \boldsymbol{\theta}) &= \partial_{w_i} L(\alpha, \boldsymbol{w}, \boldsymbol{\lambda}), i = 1 \ldots, d \\
\mu_{2S+d+i}(\alpha, \boldsymbol{w}, \boldsymbol{\gamma}, \boldsymbol{\lambda}, \boldsymbol{\theta}) &= \partial_{w_i}[L(\alpha, \boldsymbol{w}, \boldsymbol{\theta}) + \boldsymbol{\gamma}^\top \nabla_{\boldsymbol{w}} L(\alpha, \boldsymbol{w}, \boldsymbol{\lambda})], i = 1, \ldots, d \\
\mu_{2S+2d+1}(\alpha, \boldsymbol{w}, \boldsymbol{\gamma}, \boldsymbol{\lambda}, \boldsymbol{\theta}) &= \partial_\alpha[L(\alpha, \boldsymbol{w}, \boldsymbol{\theta}) + \boldsymbol{\gamma}^\top \nabla_{\boldsymbol{w}} L(\alpha, \boldsymbol{w}, \boldsymbol{\lambda})].
\end{cases}
$$

   *Then the Jacobian $J_{\boldsymbol{\mu}}(\alpha, \boldsymbol{w}, \boldsymbol{\gamma}, \boldsymbol{\lambda}, \boldsymbol{\theta})$ has full row rank for all $(\alpha, \boldsymbol{w}, \boldsymbol{\gamma}, \boldsymbol{\lambda}, \boldsymbol{\theta}) \in \boldsymbol{\mu}^{-1}(\boldsymbol{0})$ such that $\alpha \notin \Xi$, where $\Xi$ is defined in (2).*

Under Assumption 3, we have the following result, which gives the pseudo-dimension upper-bound for the utility function class $\mathcal{U}$.

**Theorem F.25.** *Assume that Assumption 3 holds, then for any problem instance $\boldsymbol{x} \in \mathcal{X}$, the dual utility function $u_{\boldsymbol{x}}^*(\alpha)$ satisfies the following:*

(a) *The hyperparameter domain $\mathcal{A}$ can be partitioned into at most $\mathcal{O}\left(N\Delta^{4d}\left(\frac{eM}{d}\right)^d + MN(2\Delta)^{2d+1}\left(\frac{eM}{d}\right)^d\right)$ intervals such that $u_{\boldsymbol{x}}^*(\alpha)$ is a continuous function over any interval in the partition, where $N$ and $M$ are the upper-bound for the number of pieces and boundary functions respectively, and $\Delta = \max\{\Delta_p, \Delta_b\}$ is the maximum degree of piece and boundary function polynomials.*

(b) *Over those intervals, $u_{\boldsymbol{x}}^*(\alpha)$ has $\mathcal{O}\left(N\Delta^{4d+2}\left(\frac{eM}{d}\right)^d\right)$ local maxima for any problem instance $\boldsymbol{x}$.*

*Proof.* (a) First, note that we can rewrite $u_{\boldsymbol{x},i}^*(\alpha)$ as

$$u_{\boldsymbol{x},i}^*(\alpha) = \max_{\boldsymbol{w}:(\alpha,\boldsymbol{w})\in\overline{R}_{\boldsymbol{x},i}} f_{\boldsymbol{x},i}(\alpha,\boldsymbol{w}).$$

Since $\overline{R}_{\boldsymbol{x},i}$ is a connected component in $\mathcal{A}\times\mathcal{W}$, let

$$\alpha_{\boldsymbol{x},i,\inf} = \inf\{\alpha \mid \exists\boldsymbol{w}:(\alpha,\boldsymbol{w})\in R_{\boldsymbol{x},i}\}, \alpha_{\boldsymbol{x},i,\sup} = \sup\{\alpha \mid \exists\boldsymbol{w}:(\alpha,\boldsymbol{w})\in R_{\boldsymbol{x},i}\}$$

be the $\alpha$-extreme points of $\overline{R}_{\boldsymbol{x},i}$ (Definition 7). Then, for any $\alpha\in(\alpha_{\boldsymbol{x},i,\inf},\alpha_{\boldsymbol{x},i,\sup})$, there exists $\boldsymbol{w}$ such that $(\alpha,\boldsymbol{w})\in\overline{R}_{\boldsymbol{x},i}$.

Let $\mathbf{H}_{\boldsymbol{x},i}$ be the set of adjacent boundaries of $R_{\boldsymbol{x},i}$ (Definition 8). From assumption of problem setting, there are at most $M$ boundary functions $h_{\boldsymbol{x},j}$, meaning that $|\mathbf{H}_{\boldsymbol{x},i}|\leq M$. For any subset $\mathcal{S} = \{h_{\mathcal{S},1},\ldots,h_{\mathcal{S},S}\}\subset\mathbf{H}_{\mathbf{x},i}$, where $|\mathcal{S}| = S$, consider the algebraic set $Z_{\mathcal{S}}$, where

$$Z_{\mathcal{S}} = \cap_{j=1}^S\{(\alpha,\boldsymbol{w})\in\mathbb{R}\times\mathbb{R}^d \mid h_{\mathcal{S},j}(\alpha,\boldsymbol{w}) = 0\}. \tag{1}$$

If $S > d+1$, from Assumption 3.1, $Z_{\mathcal{S}}$ is an empty set. Consider $S\leq d+1$, from Assumption 3.1, $J_{\boldsymbol{h}}(\alpha,\boldsymbol{w})$ defines a smooth $(d+1-S)$-dimensional manifold in $\mathbb{R}\times\mathbb{R}^d$. Note that, this is exactly the set of $(\alpha,\boldsymbol{w})$ defined by the equation

$$\sum_{j=1}^S h_{\mathcal{S},j}(\alpha,\boldsymbol{w})^2 = 0,$$

where the left hand side is a polynomial in $\alpha,\boldsymbol{w}$ of degree at most $2\Delta$. Therefore, from Lemma F.8, the number of connected components of $Z_{\mathcal{S}}$ is at most $2(2\Delta)^{d+1}$. Each connected component corresponds to 2 $\alpha$-extreme points, meaning that there are at most $4(2\Delta)^{d+1}$ $\alpha$-extreme points for all the connected components of $Z_{\mathcal{S}}$. Taking all possible subset $\mathcal{S}$ of $\mathbf{H}_{\boldsymbol{x},i}$ with at most $d+1$ elements, we have a total of at most $\mathcal{N}$ $\alpha$-extreme points, where

$$\mathcal{N} \leq (2\Delta)^{d+1}\sum_{S=0}^{d+1}\binom{M}{S} \leq (2\Delta)^{d+1}\left(\frac{eM}{d+1}\right)^{d+1}.$$

Here, the final inequality is from Sauer-Shelah Lemma (Lemma F.7).

Let $\mathcal{A}_1$ be the set of such $\alpha$-extreme points. Then for any interval $I_t = (\alpha_t,\alpha_{t+1})$ formed by two consecutive points in $\mathcal{A}_1$, the set $\mathbf{S}_t^1$ exists. Here, the set $\mathbf{S}_t^1\in 2^{\mathbf{H}_{\boldsymbol{x},i}}$ contains all subsets $\mathcal{S}$ of $\mathbf{H}_{\boldsymbol{x},i}$ such that for any $\mathcal{S} = \{h_{\mathcal{S},1},\ldots,h_{\mathcal{S},S}\}\in\mathbf{S}_t^1$ and any $\alpha\in(\alpha_t,\alpha_{t+1})$, there exists $\boldsymbol{w}$ such that $h_{\mathcal{S},j}(\alpha,\boldsymbol{w}) = 0$ for any $j = 1,\ldots,S$.

Now, for any *fixed* $\alpha\in I_t$, assume that $\boldsymbol{w}_\alpha$ is a local maxima of $f_{\boldsymbol{x},i}$ in $\overline{R}_{\boldsymbol{x},i}$ (which exists due to the compactness of $\overline{R}_{\boldsymbol{x},i}$), meaning that $(\alpha,\boldsymbol{w}_\alpha)$ is also a local extrema in $\overline{R}_{\boldsymbol{x},i}$. This implies there exists a set of boundaries $\mathcal{S}\in\mathbf{S}_t^1$ and $\boldsymbol{\lambda}$ such that $(\alpha,\boldsymbol{w}_\alpha)$ satisfies the following due to Lagrange multipliers theorem (Theorem F.3)

$$\begin{cases} \boldsymbol{h}(\alpha,\boldsymbol{w}_\alpha) = \boldsymbol{0} \\ \nabla_{\boldsymbol{w}}L(\alpha,\boldsymbol{w}_\alpha,\boldsymbol{\lambda}) = \boldsymbol{0}. \end{cases}$$

Here $\boldsymbol{h} = \{h_{\mathcal{S}_1},\ldots,h_{\mathcal{S},S}\}$ is the polynomial map formed by the boundary functions in $\mathcal{S}$, and

$$L(\alpha,\boldsymbol{w},\boldsymbol{\lambda}) = f_{\boldsymbol{x},i}(\alpha,\boldsymbol{w}) + \boldsymbol{\lambda}^\top\boldsymbol{h}(\alpha,\boldsymbol{w})$$

is the corresponding Lagrangian. Let $\mathcal{M}_{\mathcal{S}}$ be the set of points $(\alpha,\boldsymbol{w},\boldsymbol{\lambda})$ that satisfy the equations above, which defines an algebraic set. From Lemma F.8, the number of connected components of $\mathcal{M}_{\mathcal{S}}$ is at most $2(2\Delta)^{d+S+1}$, corresponding to at most $4(2\Delta)^{d+S+1}$ $\alpha$-extreme points. Moreover, let $\overline{\boldsymbol{k}} = (\boldsymbol{h},\nabla_{\boldsymbol{w}}L)$. Then from Assumption 3.2, there exists a set of real-valued $\Xi_{\mathcal{S}}$ of at most $\Delta^{2S+2d}$ elements such that the Jacobian $J_{\overline{\boldsymbol{k}}}(\boldsymbol{w},\boldsymbol{\lambda})$ has full row rank for all $(\alpha,\boldsymbol{w},\boldsymbol{\lambda})\in\overline{\boldsymbol{k}}^{-1}(\boldsymbol{0})$ and $\alpha\notin\Xi_{\mathcal{S}}$. Taking all possible subsets $\mathcal{S}\subset\mathbf{S}_t^1$ of at most $d$ elements and noting that $\left|\mathbf{S}_t^1\right|\leq M$, we have at most $\mathcal{O}\left((2\Delta)^{2d+1}\left(\frac{eM}{d}\right)^d + \Delta^{4d}\left(\frac{eM}{d}\right)^d\right)$ such points (that are either $\alpha$-extreme points or in the set of values $\Xi_{\mathcal{S}}$).

Let $\mathcal{A}_2$ be the (sorted) set that contains all points $\alpha$ in $\mathcal{A}_1$ and the points described above. Then for any interval $I_1 = (\alpha_t, \alpha_{t+1})$ formed by two consecutive points in $\mathcal{A}_2$, there exists a set $\mathbf{S}_t^2 \in 2^{\mathbf{H}_{\boldsymbol{x},i}}$ that contains all subsets $\mathcal{S} = \{h_{\mathcal{S},1}, \ldots, h_{\mathcal{S},S}\}$ of $\mathbf{H}_{\boldsymbol{x},i}$ such that for any $\alpha \in (\alpha_t, \alpha_{t+1})$, there exist $\boldsymbol{w}_\alpha$ and $\boldsymbol{\lambda}_\alpha$ such that $(\alpha, \boldsymbol{w}_\alpha, \boldsymbol{\lambda}_\alpha)$ satisfies

$$\begin{cases} \boldsymbol{h}(\alpha, \boldsymbol{w}_\alpha) = \mathbf{0}, \\ \nabla_{\boldsymbol{w}} L(\alpha, \boldsymbol{w}_\alpha, \boldsymbol{\lambda}_\alpha) = \mathbf{0}, \end{cases}$$

and $J_{\overline{\boldsymbol{k}}}(\boldsymbol{w}, \boldsymbol{\lambda})|_{(\alpha,\boldsymbol{w},\boldsymbol{\lambda})=(\alpha,\boldsymbol{w}_\alpha,\boldsymbol{\lambda}_\alpha)}$, where $\overline{\boldsymbol{k}} = (\boldsymbol{h}, \nabla_{\boldsymbol{w}} L)$, has full row rank. Therefore, over any interval $I_t$, the set of points $(\alpha, \boldsymbol{w}, \boldsymbol{\lambda})$ that satisfy $\overline{\boldsymbol{k}}(\alpha, \boldsymbol{w}, \boldsymbol{\lambda}) = \mathbf{0}$ defines a one-dimensional manifold in the space $\mathbb{R} \times \mathbb{R}^d \times \mathbb{R}^s$ of $\alpha, \boldsymbol{w}, \boldsymbol{\lambda}$, and furthermore, its connected components are monotonic curves (Definition 19).

Note that for the points $(\alpha, \boldsymbol{w}_\alpha, \boldsymbol{\lambda}_\alpha)$, $(\alpha, \boldsymbol{w}_\alpha)$ might not be in the feasible region $\overline{R}_{\boldsymbol{x},i}$. For each set of boundaries $\mathcal{S} \in \mathbf{S}_t^2$, for each point $(\alpha, \boldsymbol{w})$ at which $\mathcal{M}_{\mathcal{S}}$ can enter or exit the feasible region $\overline{R}_{\boldsymbol{x},i}$, there exists $\boldsymbol{\lambda}$ such that $(\alpha, \boldsymbol{w}, \boldsymbol{\lambda})$ satisfies the equations

$$\begin{cases} \boldsymbol{h}(\alpha, \boldsymbol{w}) = \mathbf{0}, \\ \nabla_{\boldsymbol{w}} L(\alpha, \boldsymbol{w}, \boldsymbol{\lambda}) = \mathbf{0}, \\ h'(\alpha, \boldsymbol{w}) = 0, \text{for some } h' \in \mathbf{H}_{\boldsymbol{x},i} - \mathcal{S}, \end{cases}$$

of which the number of solutions is finite due to Assumption 3.2. In the above constraints, the first two equations say that the point $(\alpha, \boldsymbol{w}, \boldsymbol{\lambda})$ lies in the smooth one-dimensional manifold $\mathcal{M}_{\mathcal{S}}$, and the last equation ensures that $(\alpha, \boldsymbol{w}, \boldsymbol{\lambda})$ is the intersection of $\mathcal{M}_{\mathcal{S}}$ with the boundaries $Z_{h'} = \{(\alpha, \boldsymbol{w}, \boldsymbol{\lambda}) \mid h'(\alpha, \boldsymbol{w}) = 0\}$. For each set $\mathcal{S} \in \mathbf{S}_t^2$ and possible boundary $h' \in \mathbf{H}_{\boldsymbol{x},i} - \mathcal{S}$, the number of such points is at most $2(2\Delta)^{d+S+1}$. This means that there are at most $2M(2\Delta)^{d+S+1}$ such points for each $\mathcal{S}$, since $|\mathbf{S}_{\boldsymbol{x},i} \setminus \mathcal{S}| \le M$. Taking all possible sets $\mathcal{S}$ and noting that $\mathcal{S}$ has at most $d$ elements, we have at most $\mathcal{O}\left(M(2\Delta)^{2d+1}\left(\frac{eM}{d}\right)^d\right)$ such points $(\alpha, \boldsymbol{w}, \boldsymbol{\lambda})$, corresponding to at most $\mathcal{O}\left(M(2\Delta)^{2d+1}\left(\frac{eM}{d}\right)^d\right)$ $\alpha$ values.

Let $\mathcal{A}_3$ be the set that contains all the points in $\mathcal{A}_2$ and the $\alpha$ points above. Then for any interval $I_t = (\alpha_t, \alpha_{t+1})$, the set $\mathbf{S}_t^3$ is fixed. Here, the set $\mathbf{S}_t^3 \in 2^{\mathbf{H}_{\boldsymbol{x},i}}$ consists of the set of all subsets $\mathcal{S} = \{h_{\mathcal{S},1}, \ldots, h_{\mathcal{S},S}\}$ of $\mathbf{H}_{\boldsymbol{x},i}$ such that for any *fixed* $\alpha \in (\alpha_t, \alpha_{t+1})$, there exist $\boldsymbol{w}_\alpha$ and $\boldsymbol{\lambda}$ such that $(\alpha, \boldsymbol{w}_\alpha, \boldsymbol{\lambda}_\alpha)$ satisfy

$$\begin{cases} \boldsymbol{h}(\alpha, \boldsymbol{w}_\alpha) = \mathbf{0}, j = 1, \ldots, S, \\ \nabla_{\boldsymbol{w}} L(\alpha, \boldsymbol{w}_\alpha, \boldsymbol{\lambda}_\alpha) = \mathbf{0}, \\ (\alpha, \boldsymbol{w}_\alpha) \in \overline{R}_{\boldsymbol{x},i}, \\ J_{\overline{\boldsymbol{k}}}(\boldsymbol{w}, \boldsymbol{\lambda})|_{(\alpha,\boldsymbol{w},\boldsymbol{\lambda})=(\alpha,\boldsymbol{w}_\alpha,\boldsymbol{\lambda}_\alpha)} \text{ has full row rank.} \end{cases}$$

Again, the condition $J_{\overline{\boldsymbol{k}}}(\boldsymbol{w}, \boldsymbol{\lambda})|_{(\alpha,\boldsymbol{w},\boldsymbol{\lambda})=(\alpha,\boldsymbol{w}_\alpha,\boldsymbol{\lambda}_\alpha)}$ has full row rank implies that $\overline{\boldsymbol{k}}(\alpha, \boldsymbol{w}, \boldsymbol{\lambda})$ defines a smooth one-dimensional manifold that consists of monotonic curves.

In summary, there are a set of $\alpha$ points $\mathcal{A}_3$ of at most $\mathcal{O}\left(\Delta^{4d}\left(\frac{eM}{d}\right)^d + M(2\Delta)^{2d+1}\left(\frac{eM}{d}\right)^d\right)$ elements such that for any interval $I_t = (\alpha_t, \alpha_{t+1})$ of consecutive points $(\alpha_t, \alpha_{t+1})$ in $\mathcal{A}_4$, there exists a set $\mathcal{C}_t$ of monotonic curves such that for any $\alpha \in (\alpha_t, \alpha_{t+1})$, we have

$$u_{\boldsymbol{x},i}^*(\alpha) = \max_{C \in \mathcal{C}_t}\{f_{\boldsymbol{x},i}(\alpha, \boldsymbol{w}) \mid \exists \boldsymbol{\lambda}, (\alpha, \boldsymbol{w}, \boldsymbol{\lambda}) \in C\}.$$

In other words, the value of $u_{\boldsymbol{x},i}^*(\alpha)$ for $\alpha \in I_t$ is the pointwise maximum of value of functions $f_{\boldsymbol{x},i}$ along the set of monotonic curves $\mathcal{C}$. From Proposition F.21, we have $u_{\boldsymbol{x},i}^*(\alpha)$ is continuous over $I_t$. Therefore, we conclude that the number of discontinuities of $u_{\boldsymbol{x},i}^*(\alpha)$ is at most $\mathcal{O}\left(\Delta^{4d}\left(\frac{eM}{d}\right)^d + M(2\Delta)^{2d+1}\left(\frac{eM}{d}\right)^d\right)$.

Finally, recall that

$$u_{\boldsymbol{x}}^*(\alpha) = \max_{i \in \{1,\ldots,N\}} u_{\boldsymbol{x},i}(\alpha),$$

and combining with Lemma F.1, we conclude that the number of discontinuity points of $u_{\boldsymbol{x}}^*(\alpha)$ is at most $\mathcal{O}\left(N\Delta^{4d}\left(\frac{eM}{d}\right)^d + MN(2\Delta)^{2d+1}\left(\frac{eM}{d}\right)^d\right)$.

(b) We now proceed to bound the number of local maxima of $u_{\boldsymbol{x}}^*(\alpha)$. To do that, we proceed with the following steps.

**Recalling useful properties from (a).** Recall that we can rewrite $u_{\boldsymbol{x}}^*(\alpha)$ as follows

$$u_{\boldsymbol{x}}^*(\alpha) = \max_{i \in \{1,\dots,N\}} u_{\boldsymbol{x},i}^*(\alpha),$$

where

$$u_{\boldsymbol{x},i}^*(\alpha) = \max_{\boldsymbol{w}:(\alpha,\boldsymbol{w}) \in \overline{R}_{\boldsymbol{x},i}} f_{\boldsymbol{x},i}(\alpha,\boldsymbol{w}).$$

In part (a), we show that: there exist $\alpha_1 < \dots < \alpha_T$, where $T = \mathcal{O}\left(N\Delta^{4d}\left(\frac{eM}{d}\right)^d + MN(2\Delta)^{2d+1}\left(\frac{eM}{d}\right)^d\right)$, such that for any interval $I_t = (\alpha_t, \alpha_{t+1})$, there exists a set of monotonic curves $\mathcal{C}_t$ such that

$$u_{\boldsymbol{x},i}^*(\alpha) = \max_{C \in \mathcal{C}_t}\{f_{\boldsymbol{x},i}(\alpha,\boldsymbol{w}) : (\alpha,\boldsymbol{w},\boldsymbol{\lambda}) \in C\}.$$

Note that, from the property of monotonic curves (Lemma F.19), for each monotonic curve $C$ and each $\alpha$, there exists at most one $(\boldsymbol{w},\boldsymbol{\lambda})$ such that $(\alpha,\boldsymbol{w},\boldsymbol{\lambda}) \in C$, hence the definition above is well-defined.

Now, for each interval $I_t$, and monotonic curve $C \in \mathcal{C}_t$, there is connected component $\overline{R}_{\boldsymbol{x},i}$ with the piece function $f_{\boldsymbol{x},i}$ and a set of boundaries $\mathcal{S} = \{h_{\mathcal{S},1},\dots,h_{\mathcal{S},S}\} \subset \mathbf{H}_{\boldsymbol{x},i}$ such that $C$ is on the smooth one-manifold $\mathcal{M}_{\mathcal{S}}$ in $\mathbb{R}^{d+S+1}$ defined by

$$\begin{cases} \boldsymbol{h}(\alpha,\boldsymbol{w}) = \boldsymbol{0}, \\ \nabla_{\boldsymbol{w}} L(\alpha,\boldsymbol{w},\boldsymbol{\lambda}) = \boldsymbol{0}, \end{cases}$$

where $L(\alpha,\boldsymbol{w},\boldsymbol{\lambda}) = f_{\boldsymbol{x},i}(\alpha,\boldsymbol{w}) + \boldsymbol{\lambda}^\top \boldsymbol{h}(\alpha,\boldsymbol{w})$. Note that for each $\alpha \in (\alpha_t, \alpha_{t+1})$ and for each monotonic curve $C$, there is unique $\boldsymbol{w},\boldsymbol{\lambda}$ such that $(\alpha,\boldsymbol{w},\boldsymbol{\lambda}) \in C$, and therefore $u_{\boldsymbol{x},i}^*$ is just pointwise maximum of $f_{\boldsymbol{x},i}$ along the curves $C \in \mathcal{C}_t$. From Proposition F.21, to bound the number of local maxima of $u_{\boldsymbol{x},i}^*$, it suffices to bound the number of local extrema of $f_{\boldsymbol{x},i}(\alpha,\boldsymbol{w})$ along each monotonic curves.

**Analyze the number of local maxima of $f_{\boldsymbol{x},i}$ along monotonic curves.** First, note that the number of local maxima of $f_{\boldsymbol{x},i}$ along a monotonic curve $C$ is upper-bounded by the number of local extrema along $C$. Moreover, any local extrema of $f_{\boldsymbol{x},i}$ along $C$ is a local extrema of $f_{\boldsymbol{x},i}$ on the smooth 1-manifold $\mathcal{M}_{\mathcal{S}}$, i.e., satisfying the following constraints:

$$\begin{cases} \boldsymbol{h}(\alpha,\boldsymbol{w}) = \boldsymbol{0}, \\ \nabla_{\boldsymbol{w}} L(\alpha,\boldsymbol{w},\boldsymbol{\lambda}) = \boldsymbol{0}, \end{cases}$$

To see this, WLOG, assume that $(\alpha',\boldsymbol{w}',\boldsymbol{\lambda}')$ is a local maximum of $f_{\boldsymbol{x},i}$ along $C$. By definition, there exists a neighborhood $V$ of $(\alpha',\boldsymbol{w}',\boldsymbol{\lambda}') \in C$ such that for any $(\alpha,\boldsymbol{w},\boldsymbol{\lambda}) \in V$, we have $f_{\boldsymbol{x},i}(\alpha',\boldsymbol{w}') \geq f_{\boldsymbol{x},i}(\alpha,\boldsymbol{w})$. Note that by definition (Definition 20), $C$ is an open set in $\mathcal{M}_{\mathcal{S}}$. Combining with Proposition F.20, we know that $V$ is also an open neighbor of $(\alpha',\boldsymbol{w}',\boldsymbol{\lambda}')$ in $\mathcal{M}_{\mathcal{S}}$. Therefore, $(\alpha',\boldsymbol{w}',\boldsymbol{\lambda}')$ is also a local maxima of $f_{\boldsymbol{x},i}$ along $\mathcal{M}_{\mathcal{S}}$.

Therefore, it suffices to give an upper-bound for the number of local extrema of $f_{\boldsymbol{x},i}$ restricted to $\mathcal{M}_{\mathcal{S}}$. Consider the Lagrangian function

$$\mathcal{L}(\alpha,\boldsymbol{w},\boldsymbol{\gamma},\boldsymbol{\lambda},\boldsymbol{\theta}) = f_{\boldsymbol{x},i}(\alpha,\boldsymbol{w}) + \sum_{j=1}^{S} \theta_j h_{\boldsymbol{x},i,j}^{\mathcal{S}}(\alpha,\boldsymbol{w}) + \sum_{t=1}^{d} \gamma_t \left[\frac{\partial f_{\boldsymbol{x},i}(\alpha,\boldsymbol{w})}{\partial w_t} + \sum_{j=1}^{S} \lambda_j \frac{\partial h_{\boldsymbol{x},i,j}^{\mathcal{S}}(\alpha,\boldsymbol{w})}{\partial w_t}\right]$$

$$= L(\alpha,\boldsymbol{w},\boldsymbol{\theta}) + \boldsymbol{\gamma}^\top \nabla_{\boldsymbol{w}} L(\alpha,\boldsymbol{w},\boldsymbol{\lambda}).$$

From Theorem F.3, for any local extrema $(\alpha, \boldsymbol{w}, \boldsymbol{\lambda})$ of $f_{\boldsymbol{x},i}(\alpha, \boldsymbol{w})$ in $\mathcal{M}_S$, there exists $\boldsymbol{\theta} \in \mathbb{R}^S$, $\boldsymbol{\gamma} \in \mathbb{R}^d$ such that

$$\begin{cases} \boldsymbol{h}(\alpha, \boldsymbol{w}) = \boldsymbol{0}, \\ \nabla_{\boldsymbol{w}} L(\alpha, \boldsymbol{w}, \boldsymbol{\lambda}) = \boldsymbol{0}, \\ \boldsymbol{\gamma}^\top \nabla_{\boldsymbol{w}} \boldsymbol{h}(\alpha, \boldsymbol{w}) = \boldsymbol{0}, \\ \nabla_{\boldsymbol{w}}[L(\alpha, \boldsymbol{w}, \boldsymbol{\theta}) + \boldsymbol{\gamma}^\top \nabla_{\boldsymbol{w}} L(\alpha, \boldsymbol{w}, \boldsymbol{\lambda})] = \boldsymbol{0}, \\ \partial_\alpha[L(\alpha, \boldsymbol{w}, \boldsymbol{\theta}) + \boldsymbol{\gamma}^\top \nabla_{\boldsymbol{w}} L(\alpha, \boldsymbol{w}, \boldsymbol{\lambda})] = \boldsymbol{0}. \end{cases}$$

From Assumption 3.3 and Bezout's theorem, the number of points $(\alpha, \boldsymbol{w}, \boldsymbol{\gamma}, \boldsymbol{\lambda}, \boldsymbol{\theta})$ that satisfy the equations above is at most $\Delta^{2S+2d+1}$. Hence, we conclude that there is at most $\Delta^{2S+2d+1}$ local extrema of $f_{\boldsymbol{x},i}$ along any monotonic curve $C$ of $\mathcal{M}_S$ such that $(\alpha, \boldsymbol{w}) \in \overline{R}_{\boldsymbol{x},i}$.

**Analyzing the number of local extrema of $u_{\boldsymbol{x}}^*$.** In the previous step, for any set of boundaries $\mathcal{S} \subset \mathbf{H}_{\boldsymbol{x},i}$ and $|\mathcal{S}| = S \leq d + 1$, we show that between all $I_t$, there are at most $\Delta^{2S+2d+1}$ local extrema for $f_{\boldsymbol{x},i}^*$ along any monotonic curve of $\mathcal{M}_S$. We now take the sum over any $\mathcal{S} \subset \mathbf{H}_{\boldsymbol{x},i}$, $|\mathcal{S}| = S \leq d$ and any region $R_i$ for $i = 1, \ldots, N$, we then conclude that the number local extrema of $u_{\boldsymbol{x}}^*(\alpha)$ across all intervals $I_t$ is $\mathcal{O}(\mathcal{N})$, where

$$\begin{aligned} \mathcal{N} &= N \sum_{S=0}^{d} \binom{M}{S} \Delta^{2S+2d+1} \\ &= N \Delta^{4d+2} \sum_{S=0}^{d+1} \binom{M}{S} && \text{(because } S \leq d) \\ &= N \Delta^{4d+2} \left( \frac{eM}{d} \right)^d && \text{(Lemma F.7).} \end{aligned}$$

$\square$

Combining Theorem F.25 and Lemma 2.1, we have the following result.

**Theorem F.26.** *Let $\mathcal{U} = \{u_\alpha : \mathcal{X} \to [0,1] \mid \alpha \in \mathcal{A}\}$, where $\mathcal{A} = [\alpha_{\min}, \alpha_{\max}] \subset \mathbb{R}$. Assume that any dual utility function $u_{\boldsymbol{x}}^*$ admits piecewise polynomial structure that satisfies Assumption 3. Then we have $\mathrm{Pdim}(\mathcal{U}) = \mathcal{O}(\log N + d \log(\Delta M))$. Here, $M$ and $N$ are the number of boundaries and functions, and $\Delta$ is the maximum degree of boundaries and piece functions.*

**Second step: Relaxing Assumption 3 to Assumption 1**

In this section, we show how we can relax Assumption 3 to our main Assumption 1. In particular, we show that for any dual utility function $u_{\boldsymbol{x}}^*$ that satisfies Assumption 1, we can construct a function $v_{\boldsymbol{x}}^*$ such that: (1) The piecewise structure of $v_{\boldsymbol{x}}^*$ satisfies Assumption 3, and (2) $\|u_{\boldsymbol{x}}^* - v_{\boldsymbol{x}}^*\|_\infty$ can be *arbitrarily* small. This means that, for a utility function class $\mathcal{U}$, we can construct a new function class $\mathcal{V}$ of which each dual function $v_{\boldsymbol{x}}^*$ satisfies Assumption 3. We then can establish a pseudo-dimension upper-bound for $\mathcal{V}$ using Theorem F.25, and then recover the learning guarantee for $\mathcal{U}$ using Lemma C.4.

We will now proceed via a sequence of claims towards establishing the reduction. First, we claim that under Assumption 1, we have the following regularity condition.

**Proposition F.27.** *Under Assumption 1, the Jacobian $J_{\boldsymbol{\mu}}(\boldsymbol{w}, \boldsymbol{\gamma})$ of the mapping $\boldsymbol{\mu} = (\mu_1, \ldots, \mu_{2S})$ for $S \leq d$, where*

$$\begin{cases} \mu_j(\alpha, \boldsymbol{w}, \boldsymbol{\gamma}) = h_j(\alpha, \boldsymbol{w}), & \text{for } j = 1, \ldots, S, \\ \mu_{S+j}(\alpha, \boldsymbol{w}, \boldsymbol{\gamma}) = \boldsymbol{\gamma}^\top \nabla_{\boldsymbol{w}} h_j(\alpha, \boldsymbol{w}), & \text{for } j = 1, \ldots, S, \end{cases}$$

*has full row rank when evaluated at $(\alpha, \boldsymbol{w}, \boldsymbol{\gamma})$ such that $\boldsymbol{\mu}(\alpha, \boldsymbol{w}, \boldsymbol{\gamma}) = \boldsymbol{0}$.*

*Proof.* Note that the Jacobian $J_{\boldsymbol{\mu}}(\boldsymbol{w}, \boldsymbol{\gamma})$ has the form

$$J_{\boldsymbol{\mu}}(\boldsymbol{w}, \boldsymbol{\gamma}) = \begin{bmatrix} J_{1\boldsymbol{w}} & J_{1\boldsymbol{\gamma}} \\ J_{2\boldsymbol{w}} & J_{2\boldsymbol{\gamma}} \end{bmatrix}.$$

Essentially, in order to show $J_{\boldsymbol{\mu}}(\boldsymbol{w}, \boldsymbol{\gamma})$ is full row rank, we will use that $J_{1\boldsymbol{w}}$ and $J_{2\boldsymbol{\gamma}}$ are both full row rank, which follows from Assumption 1.1.

In more detail, here $\boldsymbol{h} = (h_1, \ldots, h_S)$ and

$$J_{1\boldsymbol{w}} = J_{\boldsymbol{h}}(\boldsymbol{w})_{S \times d}, J_{1\boldsymbol{\gamma}} = \boldsymbol{0}_{S \times d},$$

$$J_{2\boldsymbol{w}} = \begin{bmatrix} \boldsymbol{\gamma}^\top H_{h_1}(\boldsymbol{w}) \\ \vdots \\ \boldsymbol{\gamma}^\top H_{h_S}(\boldsymbol{w}) \end{bmatrix}_{S \times d}, J_{2\boldsymbol{\gamma}} = J_{\boldsymbol{h}}(\boldsymbol{w})_{S \times d},$$

where $H_{h_j}(\boldsymbol{w})$ is the Hessian of $h_j$ w.r.t. $\boldsymbol{w}$. Now, in order to prove that $J_{\boldsymbol{\mu}}(\boldsymbol{w}, \boldsymbol{\gamma})$ has full row rank, suppose that there are real coefficients $\delta_1, \ldots, \delta_{2S}$ such that

$$\sum_{j=1}^{2S} \delta_j J_j = \boldsymbol{0},$$

where $J_j$ is the $j^{th}$ row of $J_{\boldsymbol{\mu}}(\boldsymbol{w}, \boldsymbol{\gamma})$. Restricting on the columns corresponding to $\boldsymbol{\gamma}$, we have

$$\sum_{j=S+1}^{2S} \delta_j J_{\boldsymbol{h}}(\boldsymbol{w})_j = \boldsymbol{0}.$$

From Assumption 1.1, we have $J_{\boldsymbol{h}}(\boldsymbol{w})$ has full row rank for any $(\alpha, \boldsymbol{w})$ that satisfies $\boldsymbol{h}(\alpha, \boldsymbol{w}) = \boldsymbol{0}$. It means $\delta_j = 0$ for $j = S+1, \ldots, 2S$. Combining with $\sum_{j=1}^{2S} \delta_j J_j = 0$, we have

$$\sum_{j=1}^{S} \delta_j J_j = 0.$$

Again, using the fact that $J_{\boldsymbol{h}}(\boldsymbol{w})$ has full row rank, we also claim that $\delta_j = 0$ for $j = 1, \ldots, S$. Therefore, $\delta_j = 0$ for $j = 1, \ldots, 2S$, meaning that $J_{\boldsymbol{\mu}}(\boldsymbol{w}, \boldsymbol{\gamma})$ has full row rank. $\square$

We now present the notion of algebraic dimension from the literature.

**Definition 21** (Stratification, real algebraic dimension, 86). *Let $S$ be a semi-algebraic set in $\mathbb{R}^n$. Then $S$ can be decomposed to finite disjoint union of smooth, connected manifolds $S_i$, called strata, i.e.,*

$$S = \cup_{i=1}^{p} S_i,$$

*where $S_i$ is a smooth manifold. The algebraic dimension of $S$ is the maximum dimension of the smooth manifolds in the stratification of S, i.e., $\mathrm{Adim}(S) = \max_{i=1}^{p} \dim(S_i)$.*

We establish the following straightforward connection between the regular values of a function defined on a semi-algebraic set and the algebraic dimension of its domain.

**Proposition F.28.** *Let $\boldsymbol{f} : S \to \mathbb{R}^n$ be a polynomial map, where $S$ is a semi-algebraic set. Then the set of non-regular values of $\boldsymbol{f}$ has Lebesgue measure 0 in $\mathbb{R}^n$. Assume that $\mathrm{Adim}(S) = n$, then given a regular value $\boldsymbol{v}$ of $\boldsymbol{f}$, $\boldsymbol{f}^{-1}(\boldsymbol{v})$ only contains isolated points in $S$.*

*Proof.* Consider a stratification $S = \cup_{i=1}^{p} S_i$. For any strata $S_i$, consider the restricted mapping $\boldsymbol{f}|_{S_i} : S_i \to \mathbb{R}^n$. From Sard's theorem, the set $C_i \subset \mathbb{R}^n$ of non-regular values of $\boldsymbol{f}|_{S_i}$ has Lebesgue measure 0 in $\mathbb{R}^n$. Therefore, the set of non-regular value $C = \cup_{i=1}^{p} C_i$ of $f$, which is a finite union of sets with Lebesgue measure 0, also has Lebesgue measure 0 in $\mathbb{R}^n$.

Now, given a regular value $\boldsymbol{v} \in \mathbb{R}^n$, and consider the restriction $\boldsymbol{f}|_{S_i} : S_i \to \mathbb{R}^n$, for any strata $S_i$. If $S_i$ is a smooth $k$-dimensional manifold for $k < n$, then $f^{-1}(\boldsymbol{v})$ must be empty. To see that, consider the differential map $d\boldsymbol{f}_x : T_x S_i \to \mathbb{R}^n$ (since $T_{f(x)}\mathbb{R}^n = \mathbb{R}^n$), which can never be a surjective (since $\dim(S_i) < n$). Therefore, $\boldsymbol{f}^{-1}(\boldsymbol{v})$ has to be an empty so that the surjective condition holds vacuously. If $S_i$ is a smooth $n$-dimensional manifold, then $\boldsymbol{f}^{-1}(\boldsymbol{v})$ is smooth 0-dimensional manifold in $S_i$, which means that it contains only isolated points. $\square$

Finally, the following supporting result uses Bezout's theorem to bound the number of non-singular points corresponding to the set $\Xi$ in Assumption 3.

**Proposition F.29.** *Consider a piece function $f_{\boldsymbol{x},i}$ and a set of $S$ boundary functions $\mathcal{S} = \{h_1, \ldots, h_S\}$. Let $L(\alpha, \boldsymbol{w}, \boldsymbol{\lambda}) = f_{\boldsymbol{x},i}(\alpha, \boldsymbol{w}) + \boldsymbol{\lambda}^\top \boldsymbol{h}(\alpha, \boldsymbol{w})$, where $\boldsymbol{h} = (h_1, \ldots, h_S)$, and consider the mapping $\boldsymbol{\mu} = (\mu_1, \ldots, \mu_{S+d})$*

$$\begin{cases} \mu_j(\alpha, \boldsymbol{w}, \boldsymbol{\lambda}) = h_j(\alpha, \boldsymbol{w}), \text{ for } j = 1, \ldots, S, \\ \mu_{S+z}(\alpha, \boldsymbol{w}, \boldsymbol{\lambda}) = \partial_{w_z} L(\alpha, \boldsymbol{w}, \boldsymbol{\lambda}), \text{ for } z = 1, \ldots, d. \end{cases}$$

*Under Assumption 1, we can choose $\boldsymbol{\tau} \in \mathbb{R}^d$ such that the set $\Xi$ of parameters $\alpha$ for which the system*

$$\begin{cases} \boldsymbol{h}(\alpha, \boldsymbol{w}) = \boldsymbol{0}, \\ \nabla_{\boldsymbol{w}} L(\alpha, \boldsymbol{w}, \boldsymbol{\lambda}) - \boldsymbol{\tau} = 0 \end{cases}$$

*has a solution $(\boldsymbol{w}, \boldsymbol{\lambda})$ where the Jacobian $J_{\boldsymbol{\mu}}(\boldsymbol{w}, \boldsymbol{\lambda})$ is singular, is finite. Moreover*

1. *Given such $\boldsymbol{\tau}$, the set $\Xi$ has at most $\Delta^{2S+2d}$ elements. Here, $\Delta$ is the maximum degree of the piece functions $f_{\boldsymbol{x},i}$ and boundary functions $h_j$.*

2. *The set of $T_{i,\mathcal{S}}$ of $\boldsymbol{\tau}$, which does not satisfy the above, has Lebesgue measure 0 in $\mathbb{R}^d$.*

*Proof.* Let $Z_{\boldsymbol{h}} = \{(\alpha, \boldsymbol{w}, \boldsymbol{\lambda}) \in \mathbb{R}^{S+d+1} \mid \boldsymbol{h}(\alpha, \boldsymbol{w}, \boldsymbol{\lambda}) = \boldsymbol{0}\}$, which is an algebraic set. From Assumption 1.1, since $J_{\boldsymbol{h}}(\boldsymbol{w}, \boldsymbol{\lambda})$ has full rank for any $(\alpha, \boldsymbol{w}, \boldsymbol{\lambda}) \in Z_{\boldsymbol{h}}$, $Z_{\boldsymbol{h}}$ has algebraic dimension $(d + S + 1) - S = d + 1$ (or more concretely, $Z_{\boldsymbol{h}}$ defines a smooth $(d + 1)$-dimensional manifold in $\mathbb{R}^{d+S+1}$).

Next, consider $Z_{\det(J_{\boldsymbol{\mu}}(\boldsymbol{w},\boldsymbol{\lambda}))} = \{(\alpha, \boldsymbol{w}, \boldsymbol{\lambda}) \in \mathbb{R}^{S+d+1} \mid \det(J_{\boldsymbol{\mu}}(\boldsymbol{w}, \boldsymbol{\lambda})) = 0\}$. Since each entry of $J_{\boldsymbol{\mu}}(\boldsymbol{w}, \boldsymbol{\lambda})$ is a polynomial, therefore $\det(J_{\boldsymbol{\mu}}(\boldsymbol{w}, \boldsymbol{\lambda}))$ is also a polynomial, and $Z_{\det(J_{\boldsymbol{\mu}}(\boldsymbol{w},\boldsymbol{\lambda}))}$ is an algebraic set. Let $W = Z_{\boldsymbol{h}} \cap Z_{\det(J_{\boldsymbol{\mu}}(\boldsymbol{w},\boldsymbol{\lambda}))}$, which is also an algebraic set. From Assumption 1.2, the set $W$ has (algebraic) dimension at most $d$.

Now, consider the mapping $P : W \to \mathbb{R}^d$, where $P(\alpha, \boldsymbol{w}, \boldsymbol{\lambda}) = \nabla_{\boldsymbol{w}} L(\alpha, \boldsymbol{w}, \boldsymbol{\lambda})$, which maps a point $(\alpha, \boldsymbol{w}, \boldsymbol{\lambda})$ in the algebraic set $W$ to a point in $\mathbb{R}^d$. From Proposition F.28, the set of non-regular values of $P$ has Lebesgue measure 0 in $\mathbb{R}^d$. Let $\boldsymbol{\tau} \in \mathbb{R}^d$ be a regular value of $P$. From Proposition F.28, $P^{-1}(\boldsymbol{\tau})$ only contains isolated points in $W$. Now, note that $W = Z_{\boldsymbol{h}} \cap Z_{\det(J_{\boldsymbol{\mu}}(\boldsymbol{w},\boldsymbol{\lambda}))}$, and $\det(J_{\boldsymbol{\mu}}(\boldsymbol{w}, \boldsymbol{\lambda}))$ is a polynomial of degree at most $\Delta^{S+d}$. From Bezout's theorem (Corollary F.10), we have $\left| P^{-1}(\boldsymbol{\tau}) \right| \leq \Delta^{2S+2d}$.

Therefore, any other solution $(\alpha, \boldsymbol{w}, \boldsymbol{\lambda})$ that satisfies $\boldsymbol{h}(\alpha, \boldsymbol{w}) = \boldsymbol{0}$ and $\nabla_{\boldsymbol{w}} L(\alpha, \boldsymbol{w}, \boldsymbol{\lambda}) - \boldsymbol{\tau} = \boldsymbol{0}$ has to have $\det(J_{\boldsymbol{\mu}}(\alpha, \boldsymbol{w}, \boldsymbol{\lambda})) \neq 0$, or $J_{\boldsymbol{\mu}}(\alpha, \boldsymbol{w}, \boldsymbol{\lambda})$ is non-singular, which concludes the proof. $\qquad\square$

We now present the main claim in this section, which says that for any function $u_{\boldsymbol{x}}^*(\alpha)$ that satisfies Assumption 1, we can construct a function $v_{\boldsymbol{x}}^*(\alpha)$ that satisfies Assumption 3 and that $\|u_{\boldsymbol{x}}^* - v_{\boldsymbol{x}}^*\|_\infty$ can be arbitrarily small.

**Lemma F.30.** *Let $u_{\boldsymbol{x}}^*$ be a dual utility function of a utility function class $\mathcal{U}$. Assume that the piecewise polynomial structure of $u_{\boldsymbol{x}}^*$ satisfies Assumption 1, then we can construct the function $v_{\boldsymbol{x}}^*$ such that $v_{\boldsymbol{x}}^*$ has piecewise polynomial structures that satisfies Assumption 3, and $\|u_{\boldsymbol{x}}^* - v_{\boldsymbol{x}}^*\|_\infty$ can be arbitrarily small.*

*Proof.* **Proof overview.** First, notice that Assumption 1 immediately implies Assumption 3.1. Therefore, given a dual utility function $u_{\boldsymbol{x}}^*$ of which the piecewise polynomial structure satisfies Assumption 1, the task now is to construct $v_{\boldsymbol{x}}^*$ such that $v_{\boldsymbol{x}}^*$ has a piecewise polynomial structure that satisfies Assumptions 3.2 and 3.3.

Given a set of boundary functions $\mathcal{S} = \{h_1, \ldots, h_S\}$ and a piece functions $f_{\boldsymbol{x},i}$ that satisfies Assumption 1, we will show that there is a set $T_{\mathcal{S},i} \subset \mathbb{R}^d$ that has Lebesgue measure 0 in $\mathbb{R}^d$ such that for any $\boldsymbol{\tau} \in \mathbb{R}^d - T_{\mathcal{S},i}$, if we subtract $\boldsymbol{\tau}^\top \boldsymbol{w}$ from the piece function $f_{\boldsymbol{x},i}(\alpha, \boldsymbol{w})$, i.e., $f'_{\boldsymbol{x},i}(\alpha, \boldsymbol{w}) = f_{\boldsymbol{x},i}(\alpha, \boldsymbol{w}) - \boldsymbol{\tau}^\top \boldsymbol{w}$, then the collection of boundaries functions $\mathcal{S} = \{h_1, \ldots, h_S\}$ as well as the new piece function $f'_{\boldsymbol{x},i}(\alpha, \boldsymbol{w})$ satisfy Assumption 3.2.

Then, given a set of boundary $\mathcal{S} = \{h_1, \ldots, h_S\}$, a new piece function $f'_{\boldsymbol{x},i}$, and the set $T_{\mathcal{S},i}$, we further show that there exists a set $A_{T_{\mathcal{S},i}} \subset \mathbb{R}$ that has Lebesgue measure 0 in $\mathbb{R}$ such that for

any $a \in \mathbb{R} - A_{T_{\mathcal{S},i}}$, if we consider another new piece function $f''_{\boldsymbol{x},i}(\alpha, \boldsymbol{w}) = f'_{\boldsymbol{x},i}(\alpha, \boldsymbol{w}) - a\alpha$ (perturbing the original piece function $f_{\boldsymbol{x},i}$ by $-\boldsymbol{\tau}^\top \boldsymbol{w} - a\alpha$), then the set of boundaries functions $\mathcal{S} = \{h_1, \ldots, h_S\}$ and the new piece function $f''_{\boldsymbol{x},i}(\alpha, \boldsymbol{w})$ satisfies Assumption 3.2 and 3.3.

Finally, for any piece $i$ and any set of boundary functions $\mathcal{S}$, if we choose $(a, \boldsymbol{\tau}) \in (\mathbb{R} - A_{T_{\mathcal{S},i}}) \times \mathbb{R}^d - T_{\mathcal{S},i}$, we will have the new piecewise structure that satisfies Assumption 3. Furthermore, since $(\mathbb{R} - A_{T_{\mathcal{S},i}}) \times \mathbb{R}^d - T_{\mathcal{S},i}$ has full measure in $\mathbb{R} \times \mathbb{R}^d$, we can choose $(a, \boldsymbol{\tau})$ arbitrarily small.

**Technical details.**

We first calculate the Jacobian matrix $J_{\boldsymbol{\mu}}$. For convenience, let $L(\alpha, \boldsymbol{w}, \boldsymbol{\theta}) = f_{\boldsymbol{x},i}(\alpha, \boldsymbol{w}) + \sum_{j=1}^S \theta_j h_{\mathcal{S},j}(\alpha, \boldsymbol{w}) = f_{\boldsymbol{x},i} + \boldsymbol{\theta}^\top \boldsymbol{h}$. Here, $\boldsymbol{h}(\alpha, \boldsymbol{w}) = (h_{\mathcal{S},1}(\alpha, \boldsymbol{w}), \ldots, h_{\mathcal{S},S}(\alpha, \boldsymbol{w}))$ is the mapping of considered boundaries.

We first calculate the Jacobian matrix $J_{\boldsymbol{\mu}}(\alpha, \boldsymbol{w}, \boldsymbol{\gamma}, \boldsymbol{\lambda}, \boldsymbol{\theta})$. For convenience, we decompose $J_{\boldsymbol{\mu}}(\alpha, \boldsymbol{w}, \boldsymbol{\gamma}, \boldsymbol{\lambda}, \boldsymbol{\theta})$ into block matrices

$$
J_{\boldsymbol{\mu}}(\alpha, \boldsymbol{w}, \boldsymbol{\gamma}, \boldsymbol{\lambda}, \boldsymbol{\theta}) = \begin{bmatrix} J_{1\alpha} & J_{1\boldsymbol{w}} & J_{1\boldsymbol{\gamma}} & J_{1\boldsymbol{\lambda}} & J_{1\boldsymbol{\theta}} \\ J_{2\alpha} & J_{2\boldsymbol{w}} & J_{2\boldsymbol{\gamma}} & J_{2\boldsymbol{\lambda}} & J_{2\boldsymbol{\theta}} \\ J_{3\alpha} & J_{3\boldsymbol{w}} & J_{3\boldsymbol{\gamma}} & J_{3\boldsymbol{\lambda}} & J_{3\boldsymbol{\theta}} \\ J_{4\alpha} & J_{4\boldsymbol{w}} & J_{4\boldsymbol{\gamma}} & J_{4\boldsymbol{\lambda}} & J_{4\boldsymbol{\theta}} \\ J_{5\alpha} & J_{5\boldsymbol{w}} & J_{5\boldsymbol{\gamma}} & J_{5\boldsymbol{\lambda}} & J_{5\boldsymbol{\theta}} \end{bmatrix},
$$

where the rows $J_1, J_2, J_3, J_4, J_5$ corresponds to $(\mu_1, \ldots, \mu_S)$, $(\mu_{S+1}, \ldots, \mu_{2S})$, $(\mu_{2S+1}, \ldots, \mu_{2S+d})$, $(\mu_{2S+d+1}, \ldots, \mu_{2S+2d})$, $(\mu_{2d+2S+1})$ respectively.

For the first column (corresponding to $\alpha$), we have

$$
J_{1\alpha} = \begin{bmatrix} \partial_\alpha h_1 \\ \vdots \\ \partial_\alpha h_S \end{bmatrix}_{S\times 1}, J_{2\alpha} = \begin{bmatrix} \sum_{t=1}^d \gamma_t \partial_{\alpha w_t}^2 h_1 \\ \vdots \\ \sum_{t=1}^d \gamma_t \partial_{\alpha w_t}^2 h_S \end{bmatrix}_{S\times 1}, J_{3\alpha} = \begin{bmatrix} \partial_{\alpha, w_1}^2 L(\alpha, \boldsymbol{w}, \boldsymbol{\lambda}) \\ \vdots \\ \partial_{\alpha, w_d}^2 L(\alpha, \boldsymbol{w}, \boldsymbol{\lambda}) \end{bmatrix}_{d\times 1}
$$

$$
J_{4\alpha} = \begin{bmatrix} \partial_{\alpha w_1}^2 (L(\alpha, \boldsymbol{w}, \boldsymbol{\theta}) + \boldsymbol{\gamma}^\top \nabla_{\boldsymbol{w}} L(\alpha, \boldsymbol{w}, \boldsymbol{\lambda})) \\ \vdots \\ \partial_{\alpha w_d}^2 (L(\alpha, \boldsymbol{w}, \boldsymbol{\theta}) + \boldsymbol{\gamma}^\top \nabla_{\boldsymbol{w}} L(\alpha, \boldsymbol{w}, \boldsymbol{\lambda})) \end{bmatrix}_{d\times 1}, J_{5\alpha} = [\partial_{\alpha\alpha}^2 (L(\alpha, \boldsymbol{w}, \boldsymbol{\theta}) + \boldsymbol{\gamma}^\top \nabla_{\boldsymbol{w}} L(\alpha, \boldsymbol{w}, \boldsymbol{\lambda}))]_{1\times 1}
$$

For the second column (corresponding to $\boldsymbol{w}$), we have

$$
J_{1\boldsymbol{w}} = J_{\boldsymbol{h}}(\boldsymbol{w})_{S\times d}, J_{2\boldsymbol{w}} = \begin{bmatrix} \boldsymbol{\gamma}^\top H_{h_1}(\boldsymbol{w}) \\ \vdots \\ \boldsymbol{\gamma}^\top H_{h_S}(\boldsymbol{w}) \end{bmatrix}_{S\times d}, J_{3\boldsymbol{w}} = H_{L(\alpha, \boldsymbol{w}, \boldsymbol{\lambda})}(\boldsymbol{w})_{d\times d},
$$

$$
J_{4\boldsymbol{w}} = H_{L(\alpha, \boldsymbol{w}, \boldsymbol{\theta}) + \boldsymbol{\gamma}^\top \nabla_{\boldsymbol{w}} L(\alpha, \boldsymbol{w}, \boldsymbol{\lambda})}(\boldsymbol{w})_{d\times d}, J_{5\boldsymbol{w}} = \nabla_{\boldsymbol{w}} [\partial_\alpha (L(\alpha, \boldsymbol{w}, \boldsymbol{\theta}) + \boldsymbol{\gamma}^\top \nabla_{\boldsymbol{w}} L(\alpha, \boldsymbol{w}, \boldsymbol{\lambda}))]_{1\times d}.
$$

Here, $J_{\boldsymbol{h}}(\boldsymbol{w})$ is the Jacobian of $\boldsymbol{h}$ w.r.t. $\boldsymbol{w}$, and $H_h(\boldsymbol{w})$ is the Hessian of $h$ w.r.t. $\boldsymbol{w}$.

For the third column (corresponding to $\boldsymbol{\gamma}$), we have

$$
J_{1\boldsymbol{\gamma}} = \mathbf{0}_{S\times d}, J_{2\boldsymbol{\gamma}} = J_{\boldsymbol{h}}(\boldsymbol{w})_{d\times S}, J_{3\boldsymbol{\gamma}} = \mathbf{0}_{d\times d},
$$

$$
J_{4\boldsymbol{\gamma}} = H_{L(\boldsymbol{w}, \boldsymbol{\lambda})}(\boldsymbol{w})_{d\times d}, J_{5\boldsymbol{\gamma}} = \nabla_{\boldsymbol{w}}(\partial_\alpha L(\alpha, \boldsymbol{w}, \boldsymbol{\lambda}))_{1\times d}.
$$

For the fourth column (corresponding to $\boldsymbol{\lambda}$), we have

$$
J_{1\boldsymbol{\lambda}} = \mathbf{0}_{S\times S}, J_{2\boldsymbol{\lambda}} = \mathbf{0}_{S\times S}, J_{3\boldsymbol{\lambda}} = J_{\boldsymbol{h}}(\boldsymbol{w})_{d\times S}^\top,
$$

$$
J_{4\boldsymbol{\lambda}} = [H_{h_1}(\boldsymbol{w})\boldsymbol{\gamma}, \ldots, H_{h_S}(\boldsymbol{w})\boldsymbol{\gamma}]_{d\times S}, J_{5\boldsymbol{\lambda}} = [\partial_\alpha \boldsymbol{\gamma}^\top \nabla_{\boldsymbol{w}} h_1, \ldots, \partial_\alpha \boldsymbol{\gamma}^\top \nabla_{\boldsymbol{w}} h_S]_{1\times S}.
$$

For the fifth column (corresponding to $\boldsymbol{\theta}$), we have

$$
J_{1\boldsymbol{\theta}} = \mathbf{0}_{S\times S}, J_{2\boldsymbol{\theta}} = \mathbf{0}_{S\times S}, J_{3\boldsymbol{\theta}} = \mathbf{0}_{d\times S}, J_{4\boldsymbol{\theta}} = J_{\boldsymbol{h}}(\boldsymbol{w})_{d\times S}^\top, J_{5\boldsymbol{\theta}} = [\partial_\alpha h_1, \ldots, \partial_\alpha h_S]_{1\times S}.
$$

First, consider the mapping $\boldsymbol{\mu_1}(\alpha, \boldsymbol{w}, \boldsymbol{\lambda}) = (\boldsymbol{h}(\alpha, \boldsymbol{w}), \nabla_{\boldsymbol{w}} L(\alpha, \boldsymbol{w}, \boldsymbol{\lambda}))$, where $\boldsymbol{h} = (h_1, \ldots, h_S)$ is a polynomial mapping formed by the set of considered boundaries, and $L(\alpha, \boldsymbol{w}, \boldsymbol{\lambda}) = f_{\boldsymbol{x}, i}(\alpha, \boldsymbol{w}) + \boldsymbol{\lambda}^\top \boldsymbol{h}(\alpha, \boldsymbol{w})$ is the corresponding Lagrangian. From Proposition F.29, there exists the set $T_{\mathcal{S}, i} \subset \mathbb{R}^d$ of Lebesgue measure 0 such that for any $\boldsymbol{\tau} \in \mathbb{R}^d - T_{\mathcal{S}, i}$, there is a set of values $\Xi_{i, \mathcal{S}, \boldsymbol{\tau}}$ of size at most $\Delta^{2S+2d}$ such that any solution $(\alpha, \boldsymbol{w}, \boldsymbol{\lambda})$ that satisfies the system

$$\begin{cases} \boldsymbol{h}(\alpha, \boldsymbol{w}) = \boldsymbol{0}, \\ \nabla_{\boldsymbol{w}}[f_{\boldsymbol{x}, i}(\alpha, \boldsymbol{w}) + \boldsymbol{\lambda}^\top \boldsymbol{h}(\alpha, \boldsymbol{w})] - \boldsymbol{\tau} = \boldsymbol{0} \end{cases}$$

will have $J_{\boldsymbol{\mu_1}}(\boldsymbol{w}, \boldsymbol{\lambda})$ non-singular, except for solutions that have $\alpha \in \Xi_{i, \mathcal{S}, \boldsymbol{\tau}}$. Therefore, choosing $f'_{\boldsymbol{x}, i}(\alpha, \boldsymbol{w}) = f_{\boldsymbol{x}, i}(\alpha, \boldsymbol{w}) - \boldsymbol{\tau}^\top \boldsymbol{w}$, and note that $J_{\boldsymbol{\mu_1}}(\boldsymbol{w}, \boldsymbol{\lambda}) \equiv J_{\boldsymbol{\mu'_1}}(\boldsymbol{w}, \boldsymbol{\lambda})$ for $\boldsymbol{\mu'_1} = (\boldsymbol{h}, \nabla_{\boldsymbol{w}} L(\alpha, \boldsymbol{w}, \boldsymbol{\lambda}) - \boldsymbol{\tau})$, we have constructed a new piece function $f'_{\boldsymbol{x}, i}$ such that $f'_{\boldsymbol{x}, i}$ and $h_1, \ldots, h_S$ that satisfy Assumption 3.

Now, we will show that for any $\boldsymbol{\tau}$ chosen as above, we will have the Jacobian $J_{\boldsymbol{\mu'_2}}(\boldsymbol{w}, \boldsymbol{\gamma}, \boldsymbol{\lambda}, \boldsymbol{\theta})$

$$\begin{cases} (\mu'_2)_j(\alpha, \boldsymbol{w}, \boldsymbol{\gamma}, \boldsymbol{\lambda}, \boldsymbol{\theta}) & = h_j(\alpha, \boldsymbol{w}), j = 1, \ldots, S \\ (\mu'_2)_{S+j}(\alpha, \boldsymbol{w}, \boldsymbol{\gamma}, \boldsymbol{\lambda}, \boldsymbol{\theta}) & = \boldsymbol{\gamma}^\top \nabla_{\boldsymbol{w}} h_j, j = 1, \ldots, S \\ (\mu'_2)_{2S+i}(\alpha, \boldsymbol{w}, \boldsymbol{\gamma}, \boldsymbol{\lambda}, \boldsymbol{\theta}) & = \partial_{w_i} L'(\alpha, \boldsymbol{w}, \boldsymbol{\lambda}), i = 1 \ldots, d \\ (\mu'_2)_{2S+d+i}(\alpha, \boldsymbol{w}, \boldsymbol{\gamma}, \boldsymbol{\lambda}, \boldsymbol{\theta}) & = \partial_{w_i}[L'(\alpha, \boldsymbol{w}, \boldsymbol{\theta}) + \boldsymbol{\gamma}^\top \nabla_{\boldsymbol{w}} L'(\alpha, \boldsymbol{w}, \boldsymbol{\lambda})], i = 1, \ldots, d \end{cases}$$

where $L'(\alpha, \boldsymbol{w}, \boldsymbol{\lambda}) = f'_{\boldsymbol{x}, i}(\alpha, \boldsymbol{w}) + \boldsymbol{\lambda}^\top \boldsymbol{h}(\alpha, \boldsymbol{w})$ is non-singular when evaluated at any solution $(\alpha, \boldsymbol{w}, \boldsymbol{\gamma}, \boldsymbol{\lambda}, \boldsymbol{\theta})$ of $\boldsymbol{\mu'}(\alpha, \boldsymbol{w}, \boldsymbol{\gamma}, \boldsymbol{\lambda}, \boldsymbol{\theta}) = \boldsymbol{0}$ such that $\alpha \notin \Xi_{i, \mathcal{S}, \boldsymbol{\tau}}$. First, recall that the form of $J_{\boldsymbol{\mu'_2}}(\boldsymbol{w}, \boldsymbol{\gamma}, \boldsymbol{\lambda}, \boldsymbol{\theta})$ is

$$J_{\boldsymbol{\mu'_2}}(\boldsymbol{w}, \boldsymbol{\gamma}, \boldsymbol{\lambda}, \boldsymbol{\theta}) = \begin{bmatrix} (\boldsymbol{w}) & (\boldsymbol{\gamma}) & (\boldsymbol{\lambda}) & (\boldsymbol{\theta}) \\ J_{\boldsymbol{h}}(\boldsymbol{w})_{S \times d} & \boldsymbol{0}_{S \times d} & \boldsymbol{0}_{S \times S} & \boldsymbol{0}_{S \times S} \\ J_{2\boldsymbol{w}} & J_{\boldsymbol{h}}(\boldsymbol{w})_{S \times d} & \boldsymbol{0}_{S \times S} & \boldsymbol{0}_{S \times S} \\ H_{L(\alpha, \boldsymbol{w}, \boldsymbol{\lambda})}(\boldsymbol{w})_{d \times d} & \boldsymbol{0}_{d \times d} & J_{\boldsymbol{h}}(\boldsymbol{w})_{d \times S}^\top & \boldsymbol{0}_{d \times S} \\ J_{4\boldsymbol{w}} & H_{L(\alpha, \boldsymbol{w}, \boldsymbol{\lambda})}(\boldsymbol{w})_{d \times d} & J_{4\boldsymbol{\lambda}} & J_{\boldsymbol{h}}(\boldsymbol{w})_{d \times S}^\top \end{bmatrix}$$

The important point is that: $(\alpha, \boldsymbol{w}, \boldsymbol{\gamma}, \boldsymbol{\lambda}, \boldsymbol{\theta})$ that satisfies $\boldsymbol{\mu'}(\alpha, \boldsymbol{w}, \boldsymbol{\gamma}, \boldsymbol{\lambda}, \boldsymbol{\theta}) = \boldsymbol{0}$ and $\alpha \notin \Xi_{\mathcal{S}, i, \boldsymbol{\tau}}$ also satisfies $\boldsymbol{\mu'_1}(\alpha, \boldsymbol{w}, \boldsymbol{\lambda}) = 0$ and

$$J_{\boldsymbol{\mu'_1}}(\boldsymbol{w}, \boldsymbol{\lambda}) = \begin{bmatrix} J_{\boldsymbol{h}}(\boldsymbol{w})_{S \times d} & \boldsymbol{0}_{S \times S} \\ H_{L(\alpha, \boldsymbol{w}, \boldsymbol{\lambda})}(\boldsymbol{w})_{d \times d} & J_{\boldsymbol{h}}(\boldsymbol{w})_{d \times S}^\top \end{bmatrix}$$

has full row rank. We will leverage this observation to show that $J_{\boldsymbol{\mu'_2}}(\boldsymbol{w}, \boldsymbol{\gamma}, \boldsymbol{\lambda}, \boldsymbol{\theta})$ also has full row rank. Now, let $\delta_1, \ldots, \delta_{2S+2d}$ be real coefficients such that

$$\sum_{t=1}^{2S+2d} \delta_t \cdot J_t = 0,$$

where $J_t$ is the $t^{th}$ row of the Jacobian $J_{\boldsymbol{\mu'_2}}(\boldsymbol{w}, \boldsymbol{\gamma}, \boldsymbol{\lambda}, \boldsymbol{\theta})$. First, consider the column corresponding to $\boldsymbol{\gamma}$ and $\boldsymbol{\theta}$ of $J_{\boldsymbol{\mu'_2}}$, we have

$$\sum_{t=S+1}^{2S} \delta_t \cdot ((J_{\boldsymbol{h}}(\boldsymbol{w}))_t, \boldsymbol{0}) + \sum_{t=2S+d+1}^{2S+2d} \delta_t \cdot ((H_{L(\alpha, \boldsymbol{w}, \boldsymbol{\lambda})}(\boldsymbol{w}))_t, (J_{\boldsymbol{h}}(\boldsymbol{w})^\top)_t) = 0.$$

Notice that the above is exactly the rows of $J_{\boldsymbol{\mu'_1}}(\boldsymbol{w}, \boldsymbol{\lambda})$, which has full row rank. Therefore $\delta_t = 0$ for $t = S+1, \ldots, 2S, 2S+d+1, \ldots, 2S+2d$. Consider this fact and the column corresponding to $\boldsymbol{w}, \boldsymbol{\lambda}$ of $J_{\boldsymbol{\mu'_2}}$, we have

$$\sum_{t=1}^{S} \delta_t \cdot ((J_{\boldsymbol{h}}(\boldsymbol{w}))_t, \boldsymbol{0}) + \sum_{t=2S+1}^{2S+d} \delta_t \cdot ((H_{L(\alpha, \boldsymbol{w}, \boldsymbol{\lambda})}(\boldsymbol{w}))_t, (J_{\boldsymbol{h}}(\boldsymbol{w})^\top)_t) = 0.$$

Again, due to $J_{\boldsymbol{\mu'_1}}(\boldsymbol{w}, \boldsymbol{\lambda})$ having full rank, we have $\delta_t = 0$ for $t = 1, \ldots, S, 2S+1, \ldots, 2S+d$. In summary, we have $\delta_t = 0$ for $t = 1, \ldots, 2S+2d$, which implies $J_{\boldsymbol{\mu'_2}}$ having full row rank.

Now, we will show that there exists a set $A_{T_{\mathcal{S},i}} \subset \mathbb{R}$ with Lebesgue measure $0$ such that for any $a \in \mathbb{R} - A_{T_{\mathcal{S},i}}$, we have that the Jacobian $J_{\boldsymbol{\mu}''}(\alpha, \boldsymbol{w}, \boldsymbol{\gamma}, \boldsymbol{\lambda}, \boldsymbol{\theta})$ with $\mu''$ given by

$$
\begin{cases}
\mu_j''(\alpha, \boldsymbol{w}, \boldsymbol{\gamma}, \boldsymbol{\lambda}, \boldsymbol{\theta}) & = h_j(\alpha, \boldsymbol{w}), j = 1, \ldots, S \\
\mu_{S+j}''(\alpha, \boldsymbol{w}, \boldsymbol{\gamma}, \boldsymbol{\lambda}, \boldsymbol{\theta}) & = \boldsymbol{\gamma}^\top \nabla_{\boldsymbol{w}} h_j, j = 1, \ldots, S \\
\mu_{2S+i}''(\alpha, \boldsymbol{w}, \boldsymbol{\gamma}, \boldsymbol{\lambda}, \boldsymbol{\theta}) & = \partial_{w_i} L''(\alpha, \boldsymbol{w}, \boldsymbol{\lambda}), i = 1 \ldots, d \\
\mu_{2S+d+i}''(\alpha, \boldsymbol{w}, \boldsymbol{\gamma}, \boldsymbol{\lambda}, \boldsymbol{\theta}) & = \partial_{w_i} [L''(\alpha, \boldsymbol{w}, \boldsymbol{\theta}) + \boldsymbol{\gamma}^\top \nabla_{\boldsymbol{w}} L''(\alpha, \boldsymbol{w}, \boldsymbol{\lambda})], i = 1, \ldots, d \\
\mu_{2S+2d+1}''(\alpha, \boldsymbol{w}, \boldsymbol{\gamma}, \boldsymbol{\lambda}, \boldsymbol{\theta}) & = \partial_\alpha [L''(\alpha, \boldsymbol{w}, \boldsymbol{\theta}) + \boldsymbol{\gamma}^\top \nabla_{\boldsymbol{w}} L''(\alpha, \boldsymbol{w}, \boldsymbol{\lambda})],
\end{cases}
$$

has full row rank when evaluated at the solution $(\alpha, \boldsymbol{w}, \boldsymbol{\gamma}, \boldsymbol{\lambda}, \boldsymbol{\theta}) = \boldsymbol{0}$ of $\boldsymbol{\mu}''(\alpha, \boldsymbol{w}, \boldsymbol{\gamma}, \boldsymbol{\lambda}, \boldsymbol{\theta}) = \boldsymbol{0}$ and $\alpha \notin \Xi_{\mathcal{S},i}$. Here, $L''(\alpha, \boldsymbol{w}, \boldsymbol{\lambda}) = f_{\boldsymbol{x},i}''(\alpha, \boldsymbol{w}) + \boldsymbol{\lambda}^\top h(\alpha, \boldsymbol{w})$, and $f_{\boldsymbol{x},i}''(\alpha, \boldsymbol{w}) = f_{\boldsymbol{x},i}'(\alpha, \boldsymbol{w}) - a\alpha$ (e.g., perturbing $f_{\boldsymbol{x},i}(\alpha, \boldsymbol{w}) - \boldsymbol{\tau}^\top \boldsymbol{w} - a\alpha$ to have $f_{\boldsymbol{x},i}''(\alpha, \boldsymbol{w})$). The important point in this step is that: any solution $(\alpha, \boldsymbol{w}, \boldsymbol{\gamma}, \boldsymbol{\lambda}, \boldsymbol{\theta})$ of $\boldsymbol{\mu}''(\alpha, \boldsymbol{w}, \boldsymbol{\gamma}, \boldsymbol{\lambda}, \boldsymbol{\theta}) = \boldsymbol{0}$ also satisfies $\boldsymbol{\mu}_2'(\alpha, \boldsymbol{w}, \boldsymbol{\gamma}, \boldsymbol{\lambda}, \boldsymbol{\theta}) = \boldsymbol{0}$, and $J_{\boldsymbol{\mu}_{1:2S+2d}''}(\boldsymbol{w}, \boldsymbol{\gamma}, \boldsymbol{\lambda}, \boldsymbol{\theta})$ is exactly $J_{\boldsymbol{\mu}_2'}(\boldsymbol{w}, \boldsymbol{\gamma}, \boldsymbol{\lambda}, \boldsymbol{\theta})$. Therefore, consider any interval $I_t = (\alpha_t, \alpha_{t+1})$ where $\alpha_t, \alpha_{t+1}$ are two consecutive points in $\Xi_{\mathcal{S},i}$, we have $J_{\boldsymbol{\mu}_{1:2S+2d}''}(\alpha, \boldsymbol{w}, \boldsymbol{\gamma}, \boldsymbol{\lambda}, \boldsymbol{\theta})$ has full row rank, since $J_{\boldsymbol{\mu}_{1:2S+2d}''}(\boldsymbol{w}, \boldsymbol{\gamma}, \boldsymbol{\lambda}, \boldsymbol{\theta})$ has full row rank. Now, $\boldsymbol{\mu}_{1:2S+2d}'' = \boldsymbol{0}$ defines a smooth manifold $\mathcal{M}$. Consider the mapping $P : \mathcal{M} \to \mathbb{R}$, where $P = \mu_{2S+2d+1}''$. From Sard's theorem F.15, there exists the set $A_{T_{\mathcal{S},i,t}} \subset \mathbb{R}$ with Lebesgue measure $0$ such that any $\alpha \in \mathbb{R} - A_{T_{\mathcal{S},i,t}}$ is a regular value of $P$. Let $A_{T_{\mathcal{S},i}} = \cap_t A_{T_{\mathcal{S},i,t}}$, which has Lebesgue measure $0$ in $\mathbb{R}$, we have for any $a \in \mathbb{R} - A_{T_{\mathcal{S},i}}$, we have $J_{\boldsymbol{\mu}''}(\alpha, \boldsymbol{w}, \boldsymbol{\gamma}, \boldsymbol{\lambda}, \boldsymbol{\theta})$ has full row rank when evaluated at the solution $(\alpha, \boldsymbol{w}, \boldsymbol{\gamma}, \boldsymbol{\lambda}, \boldsymbol{\theta})$ of $\boldsymbol{\mu}''(\alpha, \boldsymbol{w}, \boldsymbol{\gamma}, \boldsymbol{\lambda}, \boldsymbol{\theta}) = \boldsymbol{0}$ and $\alpha \notin \Xi_{\mathcal{S},i}$. This is exactly Assumption 3.3.

Now, to choose $(a, \boldsymbol{\tau})$ that works for any $f_{\boldsymbol{x},i}$ and any set of boundary functions $\mathcal{S} = \{h_1, \ldots, h_S\}$, we simply choose $(a, \boldsymbol{\tau}) \in B = \cap_{i,\mathcal{S}}[(\mathbb{R} - T_{\mathcal{S},i}) \times (\mathbb{R}^d - A_{T_{\mathcal{S},i}})]$, which is a full measure set in $\mathbb{R} \times \mathbb{R}^d$ since it is a finite intersection of full measure sets.

Finally, we will construct the $v_{\boldsymbol{x}}^*$ that: (1) has the piecewise structure that satisfies Assumption 1, and (2) $\|u_{\boldsymbol{x}}^* - v_{\boldsymbol{x}}^*\|_\infty$ can be arbitrarily small. The construction is as follows:

- The set of boundary functions is the same as $u_{\boldsymbol{x}}^*$ : $\{h_{\boldsymbol{x},1}, \ldots, h_{\boldsymbol{x},M}\}$.
- In any connected components $R_{\boldsymbol{x},i}$, the piece functions $f_{\boldsymbol{x},i}(\alpha, \boldsymbol{w})$ is perturbed by an amount $-a\alpha - \boldsymbol{\tau}^\top \boldsymbol{w}$, i.e.,

$$
f_{\boldsymbol{x},i}(\alpha, \boldsymbol{w}) \leftarrow f_{\boldsymbol{x},i}(\alpha, \boldsymbol{w}) - a\alpha - \boldsymbol{\tau}^\top \boldsymbol{w},
$$

where, $(a, \boldsymbol{\tau})$ is chosen random from $B_\epsilon \cap B$. Here, $B_\epsilon = \{(a, \boldsymbol{\tau}) \mid \max\{a, \tau_1, \ldots, \tau_d\} \leq \epsilon\}$ is the $\epsilon$-$\ell_\infty$ ball in $T \times A$. We note that $B_\epsilon \cap B$ is non-empty, since $B$ has full measure in $T \times A$, meaning that we can always choose $(\alpha, \boldsymbol{w})$, no matter how small $\epsilon$ is.

By constructing $v_{\boldsymbol{x}}^*$ as above, we have:

- The structure of $v_{\boldsymbol{x}}^*$ satisfies Assumption 1, and
- In any region $\overline{R}_{\boldsymbol{x},i}$, we have

$$
\left| f_{\boldsymbol{x},i}(\alpha, \boldsymbol{w}) - f_{\boldsymbol{x},i}^{v^*}(\alpha \boldsymbol{w}) \right| = \left| a\alpha + \boldsymbol{\tau}^\top \boldsymbol{w} \right| \leq \epsilon C,
$$

where $C = \max\{|\alpha_{\min}|, |\alpha_{\max}|, |w_{\min}|, |w_{\max}|\}$.

This implies

$$
\max_{\boldsymbol{w}:(\alpha, \boldsymbol{w}) \in \overline{R}_{\boldsymbol{x},i}} f_{\boldsymbol{x},i}(\alpha, \boldsymbol{w}) - 2\epsilon C \leq \max_{\boldsymbol{w}:(\alpha, \boldsymbol{w}) \in \overline{R}_{\boldsymbol{x},i}} f_{\boldsymbol{x},i}^{v^*}(\alpha, \boldsymbol{w}) \leq \max_{\boldsymbol{w}:(\alpha, \boldsymbol{w}) \in \overline{R}_{\boldsymbol{x},i}} f_{\boldsymbol{x},i}(\alpha, \boldsymbol{w}) + 2\epsilon C,
$$

or

$$
u_{\boldsymbol{x},i}^*(\alpha) - 2\epsilon C \leq v_{\boldsymbol{x},i}^*(\alpha) \leq u_{\boldsymbol{x},i}^*(\alpha) + 2\epsilon C \Rightarrow \|u_{\boldsymbol{x},i}^* - v_{\boldsymbol{x},i}^*\|_\infty \leq 2\epsilon C.
$$

Thus $\|u_{\boldsymbol{x}}^* - v_{\boldsymbol{x}}^*\|_\infty \leq 2\epsilon C$, and since $\epsilon$ can be arbitrarily small, we have the desired conclusion.

$\square$

**Final step: recovering the guarantee under Assumption 1**

We now complete the formal proof for Theorem 4.1. Let $\mathcal{U} = \{u_\alpha : \mathcal{X} \to [0, H] \mid \alpha \in \mathcal{A}\}$ be a function class of which each dual utility $u_{\boldsymbol{x}}^*$ satisfies Assumption 1. From Lemma F.30, there exists a function class $\mathcal{V} = \{v_\alpha : \mathcal{X} \to [0, H] \mid \alpha \in \mathcal{A}\}$ such that for any problem instance $\boldsymbol{x}$, we have $\|u_{\boldsymbol{x}}^* - v_{\boldsymbol{x}}^*\|_\infty$ can be arbitrarily small, and any $v_{\boldsymbol{x}}^*$ satisfies Assumption 3. From Theorem F.25, we have $\text{Pdim}(\mathcal{V}) = \mathcal{O}(\log N + d \log(\Delta M))$. From Lemma C.4, we have $\mathscr{R}_m(\mathcal{V}) = \mathcal{O}\left(\frac{\text{Pdim}(\mathcal{V})}{m}\right)$. From Lemma C.3, we have $\hat{\mathscr{R}}_S(\mathcal{U}) = \mathcal{O}\left(\sqrt{\frac{\log N + d \log(\Delta M)}{m}}\right)$, where $S \in \mathcal{X}^m$. Finally, a standard result from learning theory gives us the final claim.

## G   Additional details for Section 4.1

In this section, we provide the formal proof for Theorem 4.2.

**Theorem 4.2 (restated).** *Consider the utility function class* $\mathcal{U} = \{u_\alpha : \mathcal{X} \to [0, H] \mid \alpha \in [\alpha_{\min}, \alpha_{\max}]\}$. *Assume that the parameter-dependent dual function* $f_{\boldsymbol{x}}(\alpha, \boldsymbol{w})$ *admits piecewise polynomial structure with $N$ piece functions $f_{\boldsymbol{x},i}$ (for $i = 1, \ldots, N$) and $M$ boundaries functions $h_{\boldsymbol{x},j}$ (for $j = 1, \ldots, M$) satisfying Assumption 1. Let* $\mathcal{V} = \{v_\alpha : \mathcal{X} \to [0, H] \mid \alpha \in \mathcal{A}\}$ *be the induced function class by $\xi$-a-ERM oracle of $\mathcal{U}$. Then for any distribution $\mathcal{D}$ over $\mathcal{X}$, for any $\delta \in (0, 1)$, with probability at least $1 - \delta$ over the draw of $S = \{\boldsymbol{x}_1, \ldots, \boldsymbol{x}_m\} \sim \mathcal{D}^m$, we have*

$$|\mathbb{E}_{\boldsymbol{x} \sim \mathcal{D}}[u_{\hat{\alpha}_{ERM}}(\boldsymbol{x})] - \mathbb{E}_{\boldsymbol{x} \sim \mathcal{D}}[u_{\alpha^*}(\boldsymbol{x})]| = \mathcal{O}\left(\sqrt{\frac{\log N + d \log(\Delta M) + \log(1/\delta)}{m}} + \xi\right).$$

*Here, $\hat{\alpha}_S \in \arg\min_{\alpha \in \mathcal{A}} \frac{1}{m} \sum_{i=1}^m v_\alpha(\boldsymbol{x}_i)$, $\alpha^* \in \max_{\alpha \in \mathcal{A}} \mathbb{E}_{\boldsymbol{x} \sim \mathcal{D}}[v_\alpha(\boldsymbol{x})]$, and $\Delta = \max\{\Delta_p, \Delta_d\}$ is the maximum degree of piece functions $f_{\boldsymbol{x},i}$ and boundaries $h_{\boldsymbol{x},j}$.*

*Proof.* Given the assumption, from the final step of the proof of Theorem 4.1 (Paragraph F.5), we have $\hat{\mathscr{R}}_S(\mathcal{U}) = \mathcal{O}\left(\sqrt{\frac{\log N + d \log(\Delta M)}{m}}\right)$, where $S \in \mathcal{X}^m$. Moreover, from the $\xi$-a-ERM oracle assumption, given $u_{\boldsymbol{x}}^*(\alpha)$ corresponding to a problem instance $\boldsymbol{x}$ and a hyperparameter $\alpha$, we have access to $v_{\boldsymbol{x}}^*(\alpha)$ such that $|v_{\boldsymbol{x}}^*(\alpha) - u_{\boldsymbol{x}}^*(\alpha)| < \xi$. Therefore, from Lemma C.3, we have

$$\hat{\mathscr{R}}_{\boldsymbol{S}}(\mathcal{V}) \leq \hat{\mathscr{R}}_{\boldsymbol{S}}(\mathcal{U}) + \xi = \mathcal{O}\left(\sqrt{\frac{\log N + d \log(\Delta M)}{m}} + \xi\right).$$

Finally, a standard result from learning theory (Theorem C.2) gives us the final claim. $\square$

## H   Additional details for Section 5

### H.1   Tuning the interpolation parameter for activation functions

#### H.1.1   Omitted proofs for parametric regression setting

**Theorem 5.1 (restated).** *Let $\mathcal{L}^{AF}$ denote loss function class defined above, with activation functions $o_1, o_2$ having maximum degree $\Delta$ and maximum breakpoints $p$. Given a problem instance $\boldsymbol{x} = (X, Y)$, the dual loss function is defined as $\ell_{\boldsymbol{x}}^*(\alpha) := \min_{\boldsymbol{w} \in \mathcal{W}} f(\boldsymbol{x}, \alpha, \boldsymbol{w}) = \min_{w \in \mathcal{W}} f_{\boldsymbol{x}}(\alpha, \boldsymbol{w})$. Then, $f_{\boldsymbol{x}}(\alpha, \boldsymbol{w})$ admits piecewise polynomial structure with bounded pieces and boundaries. Further, if the piecewise structure of $f_{\boldsymbol{x}}(\alpha, \boldsymbol{w})$ satisfies Assumption 1, then for any $\delta \in (0, 1)$, w.p. at least $1 - \delta$ over the draw of problem instances $\boldsymbol{x} \sim \mathcal{D}^m$, where $\mathcal{D}$ is some distribution over $\mathcal{X}$, we have*

$$|\mathbb{E}_{\boldsymbol{x} \sim \mathcal{D}}[\ell_{\hat{\alpha}_{ERM}}(\boldsymbol{x})] - \mathbb{E}_{\boldsymbol{x} \sim \mathcal{D}}[\ell_{\alpha^*}(\boldsymbol{x})]| = \mathcal{O}\left(\sqrt{\frac{L^2 W \log \Delta + LW \log(Tpk) + \log(1/\delta)}{m}}\right).$$

*Proof.* Let $x_1, \ldots, x_T$ denote the fixed examples from the *fixed* dataset $(X, Y)$. We will show a bound $N$ on a partition of the combined parameter-hyperparameter space $\mathcal{W} \times \mathbb{R}$, such that within

each piece the function $f_{(X,Y)}(\alpha, \boldsymbol{w})$ is given by a fixed bounded-degree polynomial function in $\alpha, \boldsymbol{w}$ on the given fixed dataset $(X, Y)$, where the boundaries of the partition are induced by at most $M$ distinct polynomial threshold functions. This structure allows us to use our result Theorem 4.1 to establish a learning guarantee for the function class $\mathcal{L}^{\text{AF}}$.

The proof proceeds by an induction on the number of network layers $L$. For a single layer $L = 1$, the neural network prediction at node $j \in [k_1]$ is given by

$$\hat{y}_{ij} = \alpha o_1(\boldsymbol{w}_j x_i) + (1 - \alpha) o_2(\boldsymbol{w}_j x_i),$$

for $i \in [T]$. $\mathcal{W} \times \mathbb{R}$ can be partitioned by $2Tk_1p$ affine boundary functions of the form $\boldsymbol{w}_j x_i - t_k$, where $t_k$ is a breakpoint of $o_1$ or $o_2$, such that $\hat{y}_{ij}$ is a fixed polynomial of degree at most $\Delta + 1$ in $\alpha, \boldsymbol{w}$ in any piece of the partition $\mathcal{P}_1$ induced by the boundary functions. By Warren's theorem, we have $|\mathcal{P}_1| \leq 2 \left( \frac{4eTk_1p(\Delta+1)}{W_1} \right)^{W_1}$.

Now suppose that the neural network function computed at any node in layer $r \leq L$ for some $r \geq 1$ is given by a piecewise polynomial function of $\alpha, \boldsymbol{w}$ with at most $|\mathcal{P}_r| \leq \prod_{q=1}^{r} 2 \left( \frac{4eTk_qp(\Delta+1)^q}{W_q} \right)^{W_q}$ pieces, and at most $2Tp \sum_{q=1}^{r} k_q$ polynomial boundary functions with degree at most $(\Delta + 1)^r$. Let $j' \in [k_{r+1}]$ be a node in layer $r + 1$. The node prediction is given by $\hat{y}_{ij'} = \alpha o_1(\boldsymbol{w}_{j'} \hat{y}_i) + (1 - \alpha) o_2(\boldsymbol{w}_{j'} \hat{y}_i)$, where $\hat{y}_i$ denotes the incoming prediction to node $j'$ for input $x_i$. By inductive hypothesis, there are at most $2Tk_{r+1}p$ polynomials of degree at most $(\Delta + 1)^r + 1$ such that in each piece of the refinement of $\mathcal{P}_r$ induced by these polynomial boundaries, $\hat{y}_{ij'}$ is a fixed polynomial with degree at most $(\Delta + 1)^{r+1}$. By Warren's theorem, the number of pieces in this refinement is at most $|\mathcal{P}_{r+1}| \leq \prod_{q=1}^{r+1} 2 \left( \frac{4eTk_qp(\Delta+1)^q}{W_q} \right)^{W_q}$.

Thus $f_{(X,Y)}(\alpha, \boldsymbol{w})$ is piecewise polynomial with at most $2Tp \sum_{q=1}^{L} k_q = 2mpk$ polynomial boundary functions with degree at most $(\Delta + 1)^{2L}$, and number of pieces at most $|\mathcal{P}_L| \leq \Pi_{q=1}^{L} 2 \left( \frac{4eTk_qp(\Delta+1)^q}{W_q} \right)^{W_q}$. Assume that the piecewise polynomial structure of $f_{(X,Y)}(\alpha, \boldsymbol{w})$ satisfies Assumption 1, then applying Theorem 4.1 and a standard learning theory result gives us the final claim. $\square$

### H.1.2  Binary classification setting

In this section, we consider an alternative setting of tuning the interpolation parameter for activation functions in the classification setting. In the binary classification setting, the output of the final layer corresponds to the prediction $g(\alpha, \boldsymbol{w}, x) = \hat{y} \in \mathbb{R}$, where $\boldsymbol{w} \in \mathcal{W} \subset \mathbb{R}^W$ is the vector of parameters (network weights), and $\alpha$ is the architecture hyperparameter. The 0-1 loss on a single example $\boldsymbol{x} = (X, Y)$ is given by $\mathbb{I}_{\{g(\alpha, \boldsymbol{w}, x) \neq y\}}$, and on a set of $T$ examples as

$$\ell_\alpha^c(\boldsymbol{x}) = \min_{\boldsymbol{w} \in \mathcal{W}} \frac{1}{T} \sum_{(x,y) \in (X,Y)} \mathbb{I}_{\{g(\alpha, \boldsymbol{w}, x) \neq y\}} = \min_{\boldsymbol{w} \in \mathcal{W}} f(\boldsymbol{x}, \boldsymbol{w}, \alpha).$$

For a fixed dataset $\boldsymbol{x} = (X, Y)$, the dual class loss function is given by $\mathcal{L}_c^{\text{AF}} = \{\ell_\alpha^c : \mathcal{X} \to [0, 1] \mid \alpha \in \mathcal{A}\}$.

**Theorem H.1.** *Let $\mathcal{L}_c^{AF}$ denote loss function class defined above, with activation functions $o_1, o_2$ having maximum degree $\Delta$ and maximum breakpoints $p$. Given a problem instance $\boldsymbol{x} = (X, Y)$, the dual loss function is defined as $\ell_{\boldsymbol{x}}^*(\alpha) := \min_{\boldsymbol{w} \in \mathcal{W}} f(\boldsymbol{x}, \alpha, \boldsymbol{w}) = \min_{w \in \mathcal{W}} f_{\boldsymbol{x}}(\alpha, \boldsymbol{w})$. Then, $f_{\boldsymbol{x}}(\alpha, \boldsymbol{w})$ admits piecewise constant structure. For any $\delta \in (0, 1)$, w.p. at least $1 - \delta$ over the draw of problem instances $S \sim \mathcal{D}^m$, where $\mathcal{D}$ is some distribution over $\mathcal{X}$, we have*

$$\left| \mathbb{E}_{(X,Y)\sim\mathcal{D}}[\ell_{\hat{\alpha}}((X, Y))] - \mathbb{E}_{(X,Y)\sim\mathcal{D}}[\ell_{\alpha^*}((X, Y))] \right| = \mathcal{O}\left( \sqrt{\frac{L^2 W \log \Delta + LW \log Tpk + \log(1/\delta)}{m}} \right).$$

*Proof.* As in the proof of Theorem 5.1, the loss function $\mathcal{L}_c$ can be shown to be piecewise constant as a function of $\alpha, \boldsymbol{w}$, with at most $|\mathcal{P}_L| \leq \Pi_{q=1}^{L} 2 \left( \frac{4eTk_qp(\Delta+1)^q}{W_q} \right)^{W_q}$ pieces. We can apply Theorem 3.2 to obtain the desired learning guarantee for $\mathcal{L}_c^{\text{AF}}$. $\square$

## H.2  Data-driven hyperparameter tuning for graph polynomial kernels

### H.2.1  Classification setting

In this section, we demonstrate the applicability of our proposed results in a simple scenario: tuning the hyperparameter of a graph kernel. Here, we consider the classification case and defer the regression case to the Appendix.

**Partially labeled graph instance.** Consider a graph $\mathcal{G} = (\mathcal{V}, \mathcal{E})$, where $\mathcal{V}$ and $\mathcal{E}$ are sets of vertices and edges, respectively. Let $n = |\mathcal{V}|$ be the number of vertices. Each vertex in the graph is associated with a $d$-dimensional feature vector, and let $X \in \mathbb{R}^{n \times d}$ denote the matrix that contains all the vertices (as feature vectors) in the graph. We also have a set of indices $\mathcal{Y}_L \subset [n]$ of labeled vertices, where each vertex belongs to one of $C$ categories and $L = |\mathcal{Y}_L|$ is the number of labeled vertices. Let $y \in [F]^L$ be the vector representing the true labels of labeled vertices, where the coordinate $y_l$ of $y$ corresponds to the label of vertex $l \in \mathcal{Y}_L$.

We want to build a model for classifying the remaining (unlabeled) vertices, which correspond to $\mathcal{Y}_U = [n] \setminus \mathcal{Y}_L$, where $[n] = \{1, \ldots, n\}$. A popular and effective approach for this is to train a graph convolutional network (GCN) [22]. Along with the vertex matrix $X$, we are also given the distance matrix $\boldsymbol{\delta} = [\delta_{i,j}]_{(i,j) \in [n]^2}$ encoding the correlation between vertices in the graph. The adjacency matrix $A$ is given by a polynomial kernel of degree $\Delta$ and hyperparameter $\alpha > 0$, i.e., $A_{i,j} = (\delta(i,j) + \alpha)^\Delta$. Let $\tilde{A} = A + I_n$, where $I_n$ is the identity matrix. Also, let $\tilde{D} = [\tilde{D}_{i,j}]_{[n]^2}$ where $\tilde{D}_{i,j} = 0$ if $i \neq j$, and $\tilde{D}_{i,i} = \sum_{j=1}^{n} \tilde{A}_{i,j}$ for $i \in [n]$. We then denote a problem instance $\boldsymbol{x} = (X, y, \boldsymbol{\delta}, \mathcal{Y}_L)$ and call $\mathcal{X}$ the set of all problem instances.

**Network architecture.** We consider a simple two-layer GCN $f$ [22], which takes the adjacency matrix $A$ and vertex matrix $X$ as inputs and outputs $Z = f(X, A)$ of the form

$$Z = \hat{A} \operatorname{ReLU}(\hat{A} X W^{(0)}) W^{(1)},$$

where $\hat{A} = \tilde{D}^{-1} \tilde{A}$ is the row-normalized adjacency matrix, $W^{(0)} \in \mathbb{R}^{d \times d_0}$ is the weight matrix of the first layer, and $W^{(1)} \in \mathbb{R}^{d_0 \times F}$ is the hidden-to-output weight matrix. Here, $z_i$ is the $i^{th}$-row of $Z$ representing the score prediction of the model. The prediction $\hat{y}_i$ for vertex $i \in \mathcal{Y}_U$ is then computed from $Z$ as $\hat{y}_i = \max z_i$ which is the maximum coordinate of vector $z_i$.

**Objective function and the loss function class.** We consider the 0-1 loss function corresponding to hyperparameter $\alpha$ and network parameters $\boldsymbol{w} = (\boldsymbol{w}^{(0)}, \boldsymbol{w}^{(1)})$ for given problem instance $\boldsymbol{x}$, $f_{\boldsymbol{x}}(\alpha, \boldsymbol{w}) = \frac{1}{|\mathcal{Y}_L|} \sum_{i \in \mathcal{Y}_L} \mathbb{I}_{\{\hat{y}_i \neq y_i\}}$. The dual loss function corresponding to hyperparameter $\alpha$ for instance $\boldsymbol{x}$ is given as $\ell_\alpha(\boldsymbol{x}) = \min_{\boldsymbol{w}} f_{\boldsymbol{x}}(\alpha, \boldsymbol{w})$, and the corresponding loss function class is $\mathcal{L}^{\text{GCN}} = \{l_\alpha : \mathcal{X} \to [0, 1] \mid \alpha \in \mathcal{A}\}$.

To analyze the learning guarantee of $\mathcal{L}^{\text{GCN}}$, we first show that any dual loss function $\ell_{\boldsymbol{x}}^*(\alpha) := \ell_\alpha(\boldsymbol{x}) = \min_{\boldsymbol{w}} f_{\boldsymbol{x}}(\alpha, \boldsymbol{w})$, $f_{\boldsymbol{x}}(\alpha, \boldsymbol{w})$ has a piecewise constant structure, where: The pieces are bounded by rational functions of $\alpha$ and $\boldsymbol{w}$ with bounded degree and positive denominators. We bound the number of connected components created by these functions and apply Theorem 3.2 to derive our result.

**Theorem H.2.** *Let $\mathcal{L}^{GCN}$ denote the loss function class defined above. Given a problem instance $\boldsymbol{x}$, the dual loss function is defined as $\ell_{\boldsymbol{x}}^*(\alpha) := \min_{\boldsymbol{w} \in \mathcal{W}} f(\boldsymbol{x}, \alpha, \boldsymbol{w}) = \min_{\boldsymbol{w} \in \mathcal{W}} f_{\boldsymbol{x}}(\alpha, \boldsymbol{w})$. Then $f_{\boldsymbol{x}}(\alpha, \boldsymbol{w})$ admits piecewise constant structure. Furthermore, for any $\delta \in (0, 1)$, w.p. at least $1 - \delta$ over the draw of problem instances $S = (\boldsymbol{x}_1, \ldots, \boldsymbol{x}_m) \sim \mathcal{D}^m$, where $\mathcal{D}$ is some problem distribution over $\mathcal{X}$, we have*

$$|\mathbb{E}_{\boldsymbol{x} \sim \mathcal{D}}[\ell_{\hat{\alpha}_S}(\boldsymbol{x})] - \mathbb{E}_{\boldsymbol{x} \sim \mathcal{D}}[\ell_{\alpha^*}(\boldsymbol{x})]| = \mathcal{O}\left( \sqrt{\frac{d_0(d + F) \log nF\Delta + \log(1/\delta)}{m}} \right).$$

To prove Theorem H.2, we first show that given any problem instance $\boldsymbol{x}$, the function $f(\boldsymbol{x}, \boldsymbol{w}; \alpha) = f_{\boldsymbol{x}}(\alpha, \boldsymbol{w})$ is a piecewise constant function, where the boundaries are rational threshold functions of $\alpha$ and $\boldsymbol{w}$. We then proceed to bound the number of rational functions and their maximum degrees, which can be used to give an upper-bound for the number of connected components, using Lemma F.9. After giving an upper-bound for the number of connected components, we then use Theorem 3.2 to recover the learning guarantee for $\mathcal{U}$.

**Lemma H.3.** *Given a problem instance $\boldsymbol{x} = (X, y, \boldsymbol{\delta}, \mathcal{Y}_L)$ that contains the vertices representation $X$, the label of labeled vertices, the indices of labeled vertices $\mathcal{Y}_L$, and the distance matrix $\boldsymbol{\delta}$, consider the function*

$$f_{\boldsymbol{x}}(\alpha, \boldsymbol{w}) := f(\boldsymbol{x}, \alpha, \boldsymbol{w}) = \frac{1}{|\mathcal{Y}_L|} \sum_{i \in \mathcal{Y}_L} \mathbb{I}_{\{\hat{y}_i \neq y_i\}}$$

*which measures the 0-1 loss corresponding to the GCN parameter $\boldsymbol{w}$, polynomial kernel parameter $\alpha$, and labeled vertices on problem instance $\boldsymbol{x}$. Then we can partition the space of $\boldsymbol{w}$ and $\alpha$ into*

$$\mathcal{O}\left( \left( \frac{(nF^2)(2\Delta + 6)}{1 + dd_0 + d_0 F} \right)^{1 + dd_0 + d_0 F} (\Delta + 1)^{nd_0} \right)$$

*connected components, in each of which the function $f_{\boldsymbol{x}}(\alpha, \boldsymbol{w})$ is a constant function.*

*Proof.* First, recall that $Z = \text{GCN}(X, A) = \hat{A}\text{ReLU}(\hat{A}XW^{(0)})W^{(1)}$, where $\hat{A} = \tilde{D}^{-1}\tilde{A}$ is the row-normalized adjacency matrix, and the matrices $\tilde{A} = [\tilde{A}_{i,j}] = A + I_n$ and $\tilde{D} = [\tilde{D}_{i,j}]$ are calculated as

$$A_{i,j} = (\delta_{i,j} + \alpha)^\Delta,$$

$$\tilde{D}_{i,j} = 0 \text{ if } i \neq j, \text{ and } \tilde{D}_{i,i} = \sum_{j=1}^{n} \tilde{A}_{i,j} \text{ for } i \in [n].$$

Here, recall that $\boldsymbol{\delta} = [\delta_{i,j}]$ is the distance matrix. We first proceed to analyze the output $Z$ step by step as follow:

- Consider the matrix $T^{(1)} = XW^{(0)}$ of size $n \times d_0$. It is clear that each element of $T^{(1)}$ is a polynomial of $W^{(0)}$ of degree at most 1.
- Consider the matrix $T^{(2)} = \hat{A}T^{(1)}$ of size $n \times d_0$. We can see that each element of matrix $\hat{A}$ is a rational function of $\alpha$ of degree at most $\Delta$. Moreover, by definition, the denominator of each rational function is strictly positive. Therefore, each element of matrix $T^{(2)}$ is a rational function of $W^{(0)}$ and $\alpha$ of degree at most $\Delta + 1$.
- Consider the matrix $T^{(3)} = \text{ReLU}(T^{(2)})$ of size $n \times d_0$. By definition, we have

$$T_{i,j}^{(3)} = \begin{cases} T_{i,j}^{(2)}, & \text{if } T_{i,j}^{(2)} \geq 0 \\ 0, & \text{otherwise.} \end{cases}$$

This implies that there are $n \times d_0$ boundary functions of the form $\mathbb{I}_{T_{i,j}^{(2)} \geq 0}$ where $T_{i,j}^{(2)}$ is a rational function of $W^{(0)}$ and $\alpha$ of degree at most $\Delta + 1$ with strictly positive denominators. From Theorem F.9, the number of connected components given by those $n \times d_0$ boundaries are $\mathcal{O}\left((\Delta + 1)^{nd_0}\right)$. In each connected component, the form of $T^{(3)}$ is fixed, in the sense that each element of $T^{(3)}$ is a rational function in $W^{(0)}$ and $\alpha$ of degree at most $\Delta + 1$.
- Consider the matrix $T^{(4)} = T^{(3)}W^{(1)}$. In connected components defined above, it is clear that each element of $T^{(4)}$ is either 0 or a rational function in $W^{(0)}, W^{(1)}$, and $\alpha$ of degree at most $\Delta + 2$.
- Finally, consider $Z = \hat{A}T^{(4)}$. In each connected component defined above, we can see that each element of $Z$ is either 0 or a rational function in $W^{(0)}, W^{(1)}$, and $\alpha$ of degree at most $\Delta + 3$.

In summary, we proved above that the space of $\boldsymbol{w}, \alpha$ can be partitioned into $\mathcal{O}((\Delta + 1)^{nd_0})$ connected components, over each of which the output $Z = \text{GCN}(X, A)$ is a matrix with each element is rational function in $W^{(0}, W^{(1)}$, and $\alpha$ of degree at most $\Delta + 3$. Now in each connected component $C$, each corresponding to a fixed form of $Z$, we will analyze the behavior of $f_{\boldsymbol{x}}(\alpha, \boldsymbol{w})$, where

$$f_{\boldsymbol{x}}(\alpha, \boldsymbol{w}) = \frac{1}{|\mathcal{Y}_L|} \sum_{i \in \mathcal{Y}_L} \mathbb{I}_{\hat{y}_i \neq y_i}.$$

Here $\hat{y}_i = \arg\max_{j \in 1, \ldots, F} Z_{i,j}$, assuming that we break tie arbitrarily but consistently. For any $F \geq j > k \geq 1$, consider the boundary function $\mathbb{I}_{Z_{i,j} \geq Z_{i,k}}$, where $Z_{i,j}$ and $Z_{i,k}$ are rational functions in $\alpha$ and $\boldsymbol{w}$ of degree at most $\Delta + 3$, and have strictly positive denominators. This means that the

boundary function $\mathbb{I}_{Z_{i,j} \geq Z_{i,k}}$ can also equivalently rewritten as $\mathbb{I}_{\tilde{Z}_{i,j} \geq 0}$, where $\tilde{Z}_{i,j}$ is a polynomial in $\alpha$ and $\boldsymbol{w}$ of degree at most $2\Delta + 6$. There are $\mathcal{O}(nF^2)$ such boundary functions, partitioning the connected component $C$ into at most $\mathcal{O}\left(\left(\frac{(nF^2)(2\Delta+6)}{1+dd_0+d_0F}\right)^{1+dd_0+d_0F}\right)$ connected components. In each connected components, $\hat{y}_i$ is fixed for all $i \in \{1, \ldots, n\}$, meaning that $f_{\boldsymbol{x}}(\alpha, \boldsymbol{w})$ is a constant function.

In conclusion, we can partition the space of $\boldsymbol{w}$ and $\alpha$ into $\mathcal{O}\left(\left(\frac{(nF^2)(2\Delta+6)}{1+dd_0+d_0F}\right)^{1+dd_0+d_0F} \times (\Delta+1)^{nd_0}\right)$ connected components, in each of which the function $f_{\boldsymbol{x}}(\alpha, \boldsymbol{w})$ is a constant function. $\qquad\square$

**Theorem H.2 (restated).** *Let $\mathcal{L}^{GCN}$ denote the loss function class defined above. Given a problem instance $\boldsymbol{x}$, the dual loss function is defined as $\ell_{\boldsymbol{x}}^*(\alpha) := \min_{\boldsymbol{w} \in \mathcal{W}} f(\boldsymbol{x}, \alpha, \boldsymbol{w})) = \min_{\boldsymbol{w} \in \mathcal{W}} f_{\boldsymbol{x}}(\alpha, \boldsymbol{w})$. Then $f_{\boldsymbol{x}}(\alpha, \boldsymbol{w})$ admits piecewise constant structure. Furthermore, for any $\delta \in (0, 1)$, w.p. at least $1 - \delta$ over the draw of problem instances $\boldsymbol{x} = (\boldsymbol{x}_1, \ldots, \boldsymbol{x}_m) \sim \mathcal{D}^m$, where $\mathcal{D}$ is some problem distribution over $\mathcal{X}$, we have*

$$|\mathbb{E}_{\boldsymbol{x} \sim \mathcal{D}}[\ell_{\hat{\alpha}_{ERM}}(\boldsymbol{x})] - \mathbb{E}_{\boldsymbol{x} \sim \mathcal{D}}[\ell_{\alpha^*}(\boldsymbol{x})]| = \mathcal{O}\left(\sqrt{\frac{d_0(d+F)\log nF\Delta + \log(1/\delta)}{m}}\right).$$

*Proof.* Given a problem instance $\boldsymbol{x}$, from Lemma H.3, we can partition the space of $\boldsymbol{w}$ and $\alpha$ into $\mathcal{O}\left(\left(\frac{(nF^2)(2\Delta+6)}{1+dd_0+d_0F}\right)^{1+dd_0+d_0F}(\Delta+1)^{nd_0}\right)$ connected components, each of which the function $f_{\boldsymbol{x}}(\alpha, \boldsymbol{w})$ remains constant. Combining with Theorem 3.2, we have the final claim.

$\qquad\square$

### H.2.2 The regression setting

The case is a bit more tricky, since our piece function now is not a polynomial, but instead a rational function of $\alpha$ and $\boldsymbol{w}$. Therefore, we need a stronger assumption (Assumption 2) to have Theorem H.5.

**Graph instance and associated representations.** Consider a graph $\mathcal{G} = (\mathcal{V}, \mathcal{E})$, where $\mathcal{V}$ and $\mathcal{E}$ are sets of vertices and edges, respectively. Let $n = |\mathcal{V}|$ be the number of vertices. Each vertex in the graph is associated with a feature vector of $d$-dimension, and let $X \in \mathbb{R}^{n \times d}$ be the matrix that contains the feature vectors for all the vertices in the graph. We also have a set of indices $\mathcal{Y}_L \subset [n]$ of labeled vertices, where each vertex belongs to one of $C$ categories and $L = |\mathcal{Y}_L|$ is the number of labeled vertices. Let $y \in [-R, R]^L$ be the vector representing the true labels of labeled vertices, where the coordinate $y_l$ of $Y$ corresponds to the label vector of vertex $l \in \mathcal{Y}_L$.

**Label prediction.** We want to build a model for classifying the other unlabeled vertices, which belongs to the index set $\mathcal{Y}_U = [n] \setminus \mathcal{Y}_L$. To do that, we train a graph convolutional network (GCN) [22] using semi-supervised learning. Along with the vertices representation matrix $X$, we are also given the distance matrix $\boldsymbol{\delta} = [\delta_{i,j}]_{(i,j) \in [n]^2}$ encoding the correlation between vertices in the graph. Using the distance matrix $D$, we then calculate the following matrices $A, \tilde{A}, \tilde{D}$ which serve as the inputs for the GCN. The matrix $A = [A_{i,j}]_{(i,j) \in [n]^2}$ is the adjacency matrix which is calculated using distance matrix $\boldsymbol{\delta}$ and the polynomial kernel of degree $\Delta$ and hyperparameter $\alpha > 0$

$$A_{i,j} = (\delta(i,j) + \alpha)^{\Delta}.$$

We then let $\tilde{A} = A + I_n$, where $I_n$ is the identity matrix, and $\tilde{D} = [\tilde{D}_{i,j}]_{[n]^2}$ of which each element is calculated as

$$\tilde{D}_{i,j} = 0 \text{ if } i \neq j, \text{ and } \tilde{D}_{i,i} = \sum_{j=1}^{n} \tilde{A}_{i,j} \text{ for } i \in [n].$$

**Network architecture.** We consider a simple two-layer graph convolutional network (GCN) $f$ [22], which takes the adjacency matrix $A$ and vertices representation matrix $X$ as inputs and output $Z = f(X, A)$ of the form

$$Z = \text{GCN}(X, A) = \hat{A}\,\text{ReLU}(\hat{A}XW^{(0)})W^{(1)},$$

where $\hat{A} = \tilde{D}^{-1}\tilde{A}$, $W^{(0)} \in \mathbb{R}^{d \times d_0}$ is the weight matrix of the first layer, and $W^{(1)} \in \mathbb{R}^{d_0 \times 1}$ is the hidden-to-output weight matrix. Here, $z_i$ is the $i^{th}$ element of $Z$ representing the prediction of the model for vertice $i$.

**Objective function and the loss function class.** We consider mean squared loss function corresponding to hyperparameter $\alpha$ and networks parameter $\boldsymbol{w} = (\boldsymbol{w}^{(0)}, \boldsymbol{w}^{(1)})$ when operating on the problem instance $\boldsymbol{x}$ as follows

$$f_{\boldsymbol{x}}(\alpha, \boldsymbol{w}) = \frac{1}{|\mathcal{Y}_L|} \sum_{i \in \mathcal{Y}_L} (z_i - y_i)^2.$$

We then define the loss function corresponding to hyperparameter $\alpha$ when operating on the problem instance $\boldsymbol{x}$ as

$$\ell_\alpha(\boldsymbol{x}) = \min_{\boldsymbol{w}} f_{\boldsymbol{x}}(\alpha, \boldsymbol{w}).$$

We then define the loss function class for this problem as follow

$$\mathcal{L}_r^{\text{GCN}} = \{\ell_\alpha : \mathcal{X} \to [0, R^2] \mid \alpha \in \mathcal{A}\},$$

and our goal is to analyze the pseudo-dimension of the function class $\mathcal{L}_r^{\text{GCN}}$.

**Lemma H.4.** *Given a problem instance $\boldsymbol{x} = (X, y, \boldsymbol{\delta}, \mathcal{Y}_L)$ that contains the graph $\mathcal{G}$, its vertices representation $X$, the indices of labeled vertices $\mathcal{Y}_L$, and the distance matrix $\boldsymbol{\delta}$, consider the function*

$$f_{\boldsymbol{x}}(\alpha, \boldsymbol{w}) := f(\boldsymbol{x}, \alpha, \boldsymbol{w}) = \frac{1}{|\mathcal{Y}_L|} \sum_{i \in \mathcal{Y}_L} (z_i - y_i)^2.$$

*which measures the mean squared loss corresponding to the GCN parameter $\boldsymbol{w}$, polynomial kernel parameter $\alpha$, and labeled vertices on problem instance $\boldsymbol{x}$. Then we can partition the space of $\boldsymbol{w}$ and $\alpha$ into $\mathcal{O}((\Delta + 1)^{nd_0})$ connected components, in each of which the function $f_{\boldsymbol{x}}(\alpha, \boldsymbol{w})$ is a rational function in $\alpha$ and $\boldsymbol{w}$ of degree at most $2(\Delta + 3)$.*

*Proof.* First, recall that $Z = \text{GCN}(X, A) = \hat{A}\text{ReLU}(\hat{A}XW^{(0)})W^{(1)}$, where $\hat{A} = \tilde{D}^{-1/2}\tilde{A}\tilde{D}^{-1/2}$ is the row-normalized adjacency matrix, and the matrices $\tilde{A} = [\tilde{A}_{i,j}] = A + I_n$ and $\tilde{D} = [\tilde{D}_{i,j}]$ are calculated as

$$A_{i,j} = (\delta_{i,j} + \alpha)^\Delta,$$

$$\tilde{D}_{i,j} = 0 \text{ if } i \neq j, \text{ and } \tilde{D}_{i,i} = \sum_{j=1}^n \tilde{A}_{i,j} \text{ for } i \in [n].$$

Here, recall that $\boldsymbol{\delta} = [\delta_{i,j}]$ is the distance matrix. We first proceed to analyze the output $Z$ step by step as follows:

- Consider the matrix $T^{(1)} = XW^{(0)}$ of size $n \times d_0$. It is clear that each element of $T^{(1)}$ is a polynomial of $W^{(0)}$ of degree at most 1.
- Consider the matrix $T^{(2)} = \hat{A}T^{(1)}$ of size $n \times d_0$. We can see that each element of matrix $\hat{A}$ is a rational function of $\alpha$ of degree at most $\Delta$. Moreover, by definition, the denominator of each rational function is strictly positive. Therefore, each element of the matrix $T^{(2)}$ is a rational function of $W^{(0)}$ and $\alpha$ of degree at most $\Delta + 1$.
- Consider the matrix $T^{(3)} = \text{ReLU}(T^{(2)})$ of size $n \times d_0$. By definition, we have

$$T_{i,j}^{(3)} = \begin{cases} T_{i,j}^{(2)}, & \text{if } T_{i,j}^{(2)} \geq 0 \\ 0, & \text{otherwise.} \end{cases}$$

This implies that there are $n \times d_0$ boundary functions of the form $\mathbb{I}_{T_{i,j}^{(2)} \geq 0}$ where $T_{i,j}^{(2)}$ is a rational function of $W^{(0)}$ and $\alpha$ of degree at most $\Delta + 1$ with strictly positive denominators. From Theorem F.9, the number of connected components given by those $n \times d_0$ boundaries are $\mathcal{O}\left((\Delta + 1)^{nd_0}\right)$. In each connected component, the form of $T^{(3)}$ is fixed, in the sense that each element of $T^{(3)}$ is a rational function in $W^{(0)}$ and $\alpha$ of degree at most $\Delta + 1$.

- Consider the matrix $T^{(4)} = T^{(3)}W^{(1)}$. In connected components defined above, it is clear that each element of $T^{(4)}$ is either 0 or a rational function in $W^{(0)}, W^{(1)}$, and $\alpha$ of degree at most $\Delta + 2$.
- Finally, consider $Z = \hat{A}T^{(4)}$. In each connected component defined above, we can see that each element of $Z$ is either 0 or a rational function in $W^{(0)}, W^{(1)}$, and $\alpha$ of degree at most $\Delta + 3$.

In summary, we proved that the space of $\boldsymbol{w}, \alpha$ can be partitioned into $\mathcal{O}((\Delta + 1)^{nd_0})$ connected components, over each of which the output $Z = \text{GCN}(X, A)$ is a matrix with each element is a rational function in $W^{(0)}, W^{(1)}$, and $\alpha$ of degree at most $\Delta + 3$. It means that in each piece, the loss function would be a rational function of degree at most $2(\Delta + 3)$, as claimed. $\qquad\square$

**Theorem H.5.** *Consider the loss function class $\mathcal{L}_r^{GCN}$ defined above. For a problem instance $\boldsymbol{x}$, the dual loss function $\ell_{\boldsymbol{x}}^*(\alpha) := \min_{\boldsymbol{w} \in \mathcal{W}} f_{\boldsymbol{x}}(\alpha, \boldsymbol{w})$, where $f_{\boldsymbol{x}}(\alpha, \boldsymbol{w})$ admits piecewise polynomial structure (Lemma H.4). If we assume the piecewise polynomial structure satisfies Assumption 3, then for any $\delta \in (0, 1)$, w.p. at least $1 - \delta$ over the draw of $m$ problem instances $S \sim \mathcal{D}^m$, where $\mathcal{D}$ is some problem distribution over $\mathcal{X}$, we have*

$$|\mathbb{E}_{S \sim \mathcal{D}}[\ell_{\hat{\alpha}_{ERM}}(S)] - \mathbb{E}_{S \sim \mathcal{D}}[\ell_{\alpha^*}(S)]| = \mathcal{O}\left(\sqrt{\frac{nd_0 \log \Delta + d \log(\Delta F) + \log(1/\delta)}{m}}\right).$$

# I  A discussion on how to capture flatness of optima in our framework

Our definition of dual utility function $u_{\boldsymbol{x}}^*(\alpha) = \max_{\boldsymbol{w} \in \mathcal{W}} f_{\boldsymbol{x}}(\alpha, \boldsymbol{w})$ implicitly assumes an ERM oracle. As discussed in Section B, this ERM oracle assumption makes the function $u_{\boldsymbol{x}}^*(\alpha)$ well-defined and simplifies the analysis. However, one may argue that assuming the ERM oracle will make the behavior of tuned hyperparameters significantly different compared to when using common optimization algorithms in deep learning. The difference potentially stems from the fact that the global optimum found by an ERM oracle might have a sharp curvature, compared to the local optima found by other optimization algorithms, which tend to have flatter local curvature due to their implicit bias.

In this section, we consider the following simplified scenario where the ERM oracle also finds the near-optimum that is locally flat, and explain how our framework could potentially be useful in this case. Instead of defining $u_{\boldsymbol{x}}^*(\alpha) = \max_{\boldsymbol{w} \in \mathcal{W}} f_{\boldsymbol{x}}(\alpha, \boldsymbol{w})$, we define $u_{\boldsymbol{x}}^*(\alpha) = \max_{\boldsymbol{w} \in \mathcal{W}} f_{\boldsymbol{x}}'(\alpha, \boldsymbol{w})$, where the surrogate function $f_{\boldsymbol{x}}'(\alpha, \boldsymbol{w})$ is defined as follows.

**Definition 22** (Surrogate function construction). *Assume that $f_{\boldsymbol{x}}(\alpha, \boldsymbol{w})$ admits piecewise polynomial structure, meaning that:*

1. *The domain $\mathcal{A} \times \mathcal{W}$ of $f_{\boldsymbol{x}}$ is divided into $N$ connected components by $M$ polynomials $h_{\boldsymbol{x},1}, \ldots, h_{\boldsymbol{x},M}$ in $\alpha, \boldsymbol{w}$, each of degree at most $\Delta_b$. The resulting partition $\mathcal{P}_{\boldsymbol{x}} = \{R_{\boldsymbol{x},1}, \ldots, R_{\boldsymbol{x},N}\}$ consists of connected sets $R_{\boldsymbol{x},i}$, each formed by a connected component $C_{\boldsymbol{x},i}$ and its adjacent boundaries.*
2. *Within each $R_{\boldsymbol{x},i}$, $f_{\boldsymbol{x}}$ takes the form of a polynomial $f_{\boldsymbol{x},i}$ in $\alpha$ and $\boldsymbol{w}$ of degree at most $\Delta_p$.*

*Defining the function surrogate $f_{\boldsymbol{x}}'(\alpha, \boldsymbol{w})$ as follows:*

1. *The domain $\mathcal{A} \times \mathcal{W}$ of $f_{\boldsymbol{x}}'(\alpha, \boldsymbol{w})$ is partitioned into $N$ connected components by $M$ polynomials $h_{\boldsymbol{x},1}, \ldots, h_{\boldsymbol{x},M}$ in $\alpha, \boldsymbol{w}$ similar to $f_{\boldsymbol{x}}$. This results in a similar partition $\mathcal{P}_{\boldsymbol{x}} = \{R_{\boldsymbol{x},1}, \ldots, R_{\boldsymbol{x},N}\}$.*
2. *In each region $R_{\boldsymbol{x},i}$, $f_{\boldsymbol{x}}'$ is defined as*

$$f_{\boldsymbol{x}}'(\alpha, \boldsymbol{w}) = f_{\boldsymbol{x},i}'(\alpha, \boldsymbol{w}) = f_{\boldsymbol{x},i}(\alpha, \boldsymbol{w}) - \eta \|\nabla_{\boldsymbol{w},\boldsymbol{w}}^2 f_{\boldsymbol{x}}(\alpha, \boldsymbol{w})\|_F^2,$$

*for some fixed $\eta > 0$. Here, $\|\cdot\|_F$ denotes the Frobenius norm. We can see that $\|\nabla_{\boldsymbol{w},\boldsymbol{w}}^2 f_{\boldsymbol{x}}(\alpha, \boldsymbol{w})\|_F^2$ is a polynomial of $\alpha, \boldsymbol{w}$ of degree at most $2\Delta_p$. Therefore, $f_{\boldsymbol{x}}'(\alpha, \boldsymbol{w})$ is also a polynomial of degree at most $2\Delta_p$ in the region $R_{\boldsymbol{x},i}$.*

From the above construction, we can see that $f_{\boldsymbol{x}}'(\alpha, \boldsymbol{w})$ also admits piecewise polynomial structure, where the input domain partition $\mathcal{P}_{\boldsymbol{x}}$ is the same as $f_{\boldsymbol{x}}(\alpha, \boldsymbol{w})$. In each region $R_{\boldsymbol{x},i}$, the function $f_{\boldsymbol{x}}'(\alpha, \boldsymbol{w})$ is also a polynomial in $\alpha, \boldsymbol{w}$ of degree at most $2\Delta_p$. Therefore, our framework is still applicable in this case. Moreover, the construction above naturally introduces an extra hyperparameter $\eta$, which is the magnitude of curvature regularization. This makes the analysis more challenging, but

for simplicity, we here assume that $\eta$ is fixed and good enough for balancing the effect of the flatness regularization.

We can see that by defining $u_{\boldsymbol{x}}^*(\alpha) = \max_{\boldsymbol{w} \in \mathcal{W}} f_{\boldsymbol{x}}'(\alpha, \boldsymbol{w})$, we can capture the generalization behavior of tuned hyperparameter $\alpha$, when the solution $w^*$ of $\max_{\boldsymbol{w} \in \mathcal{W}} f_{\boldsymbol{x}}'(\alpha, \boldsymbol{w})$ is: (1) near optimal w.r.t. $\max_{\boldsymbol{w} \in \mathcal{W}} f_{\boldsymbol{x}}(\alpha, \boldsymbol{w})$, and (2) locally flat.

**Remark 5.** *We note that the example above is an oversimplified scenario. To truly understand the behavior of data-driven hyperparameter tuning without an ERM oracle, we need a better analysis to capture the behavior of $u_{\boldsymbol{x}}^*(\alpha)$ in such a scenario. This analysis should consider the joint interaction between the model, data, and the optimization algorithm, and remains an interesting direction for future work.*

# J An application to Pfaffian functions

In this section, we demonstrate how to apply our results beyond polynomial settings. In particular, we will apply our results to the case where the boundary functions involve *Pfaffian functions*, which includes commonly appearing functions like the exponential function and the logarithm function. We refer the reader to [70] for a formal background on Pfaffian functions.

We first consider the case where the boundary functions are Pfaffian functions instead of polynomials as discussed in Section 3. Concretely, given a problem instance $\boldsymbol{x} \in \mathcal{X}$, there are $M$ boundary functions $h_{\boldsymbol{x},1}, \ldots, h_{\boldsymbol{x},N}$ that partition the space $\mathcal{A} \times \mathcal{W}$ of hyperparameter $\alpha$ and parameters $\boldsymbol{w}$ into connected components, in each of which the dual utility function $u_{\boldsymbol{x}}^*(\alpha)$ remains constant. In this case, following Theorem 3.2, the main idea is again to bound the number of connected components in the space $\mathcal{A} \times \mathcal{W}$ partitioned by $M$ Pfaffian boundary functions. We now recall a standard result that allows us to handle this case.

**Lemma J.1** ([70, 87]). *Let $h_1, \ldots, h_M$ be $M$ Pfaffian functions of degree at most $\Delta$ that come from a Pfaffian chain $\mathcal{C}(x_1, \ldots, x_d, f_1, \ldots, f_q)$ of $d$ variables, length $q$, and Pfaffian degree $D$. Then the number of connected components of the complement of solution sets $\mathbb{R}^d - \cup_{i=1}^M \{\boldsymbol{x} \in \mathbb{R}^d \mid h_i(\boldsymbol{x}) = 0\}$ is upper-bounded by $2^{qd(qd-1)/2} \Delta^d [d^2(\Delta + D)]^{dq} \left(\frac{2eM}{d}\right)^d$.*

Combining Lemma J.1 and Theorem 3.2, we have the following result, which provides a generalization guarantee for function classes for which each dual utility function admits a piecewise constant structure partitioned by Pfaffian boundary functions.

**Corollary J.2.** *Consider the utility function class $\mathcal{U} = \{u_\alpha : \mathcal{X} \to [0, H] \mid \alpha \in \mathcal{A}\}$. Assuming that for any $\boldsymbol{x}$, there are $M$ boundary functions $h_{\boldsymbol{x},1}, \ldots, h_{\boldsymbol{x},M}$ partitioning the domain $\mathcal{A} \times \mathcal{W}$ into $\mathcal{P}_{\boldsymbol{x}} = \{R_{\boldsymbol{x},1}, \ldots, R_{\boldsymbol{x},N}\}$, over each of which $f_{\boldsymbol{x},i}$ remains constant. Assuming that the boundary functions $h_{\boldsymbol{x},i}$, where $i = 1, \ldots, M$, are Pfaffian functions of degree at most $\Delta$ from a Pfaffian chain of length $q$ and Pfaffian degree $D$. Then for any distribution $\mathcal{D}$ over $\mathcal{X}$, and any $\delta \in (0, 1)$, with probability at least $1 - \delta$ over the draw of $S = \{\boldsymbol{x}_1, \ldots, \boldsymbol{x}_m\} \sim \mathcal{D}^m$, we have that*

$$|\mathbb{E}_{\boldsymbol{x} \sim \mathcal{D}}[u_{\hat{\alpha}_S}(\boldsymbol{x})] - \mathbb{E}_{\boldsymbol{x} \sim \mathcal{D}}[u_{\alpha^*}(\boldsymbol{x})]| = \mathcal{O}\left(\sqrt{\frac{q^2 d^2 + qd \log(\Delta + D) + d \log M + \log(1/\delta)}{m}}\right).$$

*Proof.* From Lemma J.1, we have $N \leq \mathcal{O}\left(2^{qd(qd-1)/2} \Delta^d [d^2(\Delta + D)]^{dq} \left(\frac{2eM}{d}\right)^d\right)$. Substituting into Theorem 3.2, we have the conclusion. $\square$

We leave the more general scenario of piecewise-Pfaffian setting (where the piece functions are Pfaffian in addition to the boundaries) as an open question for future work and have the following conjectured bound.

**Conjecture 1.** *Consider the utility function class $\mathcal{U} = \{u_\alpha : \mathcal{X} \to [0, H] \mid \alpha \in [\alpha_{\min}, \alpha_{\max}]\}$. Assume that the parameter-dependent dual function $f_{\boldsymbol{x}}(\alpha, \boldsymbol{w})$ admits piecewise Pfaffian structure with $N$ piece functions $f_{\boldsymbol{x},i}$ (for $i = 1, \ldots, N$) and $M$ boundaries functions $h_{\boldsymbol{x},j}$ (for $j = 1, \ldots, M$) satisfying Assumption 3. Furthermore, assuming that the boundary functions $h_{\boldsymbol{x},j}$ and piece functions $f_{\boldsymbol{x},i}$ are Pfaffian functions of degree at most $\Delta$ from a Pfaffian chain of length $q$ and Pfaffian degree*

*D. Then for any distribution $\mathcal{D}$ over $\mathcal{X}$, for any $\delta \in (0, 1)$, with probability at least $1 - \delta$ over the draw of $S = \{\boldsymbol{x}_1, \ldots, \boldsymbol{x}_m\} \sim \mathcal{D}^m$, we have*

$$\left| \mathbb{E}_{\boldsymbol{x} \sim \mathcal{D}}[u_{\hat{\alpha}_{ERM}}(\boldsymbol{x})] - \mathbb{E}_{\boldsymbol{x} \sim \mathcal{D}}[u_{\alpha^*}(\boldsymbol{x})] \right| =$$

$$\mathcal{O}\left( \sqrt{\frac{q^2 d^2 + qd \log(\Delta + D) + \log N + d \log M + T + \log(1/\delta)}{m}} \right).$$

*Here, $\hat{\alpha}_S \in \arg\min_{\alpha \in \mathcal{A}} \frac{1}{m} \sum_{i=1}^{m} u_\alpha(\boldsymbol{x}_i)$ and $\alpha^* \in \max_{\alpha \in \mathcal{A}} \mathbb{E}_{\boldsymbol{x} \sim \mathcal{D}}[u_\alpha(\boldsymbol{x})]$.*

