# OpenReview forum: "Sample complexity of data-driven tuning of model hyperparameters in neural networks with structured parameter-dependent dual function"
_NeurIPS.cc/2025/Conference — NeurIPS 2025 poster_

### Official Review · Reviewer_x4WD · 2025-06-08

**Clarity:** 2
**Significance:** 2
**Originality:** 3
**Rating:** 3
**Confidence:** 1

**Summary:**

This paper seeks to address a special case for hyperparameter setting wherein one wishes to tune hyperparameters for training in a manner suitable for retaining the same set of hyperparameters throughout a variety of learning tasks/objectives applied to a comparable data corpus. Their method involves tuning (one hyperparameter at a time?) by way of aggregating the tuning into a dual utility function in which they show they can bound the number of discontinuities in a manner that bounds oscillations and a resulting pseudo-randomness of such dual objective function. This helps them to reduce sample complexity. They leave extension of the methods towards a setting with multiple hyperparameters to future work.

**Questions:**

I found the claim in Related Work section represents this as "the first work to focus on tuning hyperparameters for deep networks using a data driven lens" kind of suspect. Can you please clarify what is meant by "a data driven lens" in this context and how it significantly differs from prior training conventions?

**Ethical Concerns:**

["NO or VERY MINOR ethics concerns only"]

**Final Justification:**

Based on consideration of the author's rebuttal (particular noting that relevant work has been accepted to other conferences) I have reduced my confidence score from 2 to 1 (meaning my review can be considered an educated guess). Apologies that I don't feel qualified to consider the full theoretical scope of your work, but the next time I find an application in industry requiring tuning of a single hyper-parameter I will consider taking another look.

**Limitations:**

The limitations to practical applications leveraging multiple hyperparameters are fairly apparent. I am not sure if the claimed contributions, even taken at face value, would be sufficient to compensate for that.

**Paper Formatting Concerns:**

-

**Quality:**

3

**Strengths And Weaknesses:**

Strengths:
Questions surrounding how we could treat functions containing discontinuities as polynomial class are very interesting and of potential relevance to all kind of domains.

Weaknesses:
In the context of an AI research conference the idea that a hyperparameter training objective targeting a single variable at a time is possibly kind of laughable, but then I have seen interviews with Turing award winners where they describe spending the majority of their research career trying to iterate on how they train models on MNIST data set, so perhaps it is not that far fetched.

---

> ### Author Rebuttal · Authors · 2025-07-31
>
> We thank the reviewer for the feedback. We appreciate that the reviewer found our work interesting and potentially relevant to all kinds of domains. We address the reviewer's concerns below.
>
> ## Questions
> 1. __Question__: “The idea that a hyperparameter training objective targeting a single variable at a time is possibly kind of laughable”.
>
>    __Answer__:
>
>      - We note that in machine learning research, __theoretical research is far lagging behind empirical research__. In the context of hyperparameter tuning, it is true that one might tune multiple hyperparameters using arbitrarily fancy rules, provided that the tuned hyperparameters work better in some sense. However, __theoretical analysis for hyperparameter tuning remains an under-explored question__, and __our work is one of the first works attacking this question theoretically in the context of deep learning__, with strong guarantees. __Even for a single hyperparameter case, our results are non-trivial and require lots of advanced tools__ from algebraic/differential geometry.
>
>      - Another example is paper [1], which __provides guarantees for learning just a single multi-head attention layer__ (with very strong data assumptions as well), while nowadays, LLMs typically have dozens (e.g., 80 for Llama 3) of multi-head attention layers (along with other layers) stacked together. Yet, __this paper is still presented at STOC’25, one of the best theory conferences__. That is just __one among many examples that theory in machine learning needs to start with simplified settings__ compared to empirical research, but is still valuable. Understanding simple scenarios is the first and necessary step to advance the understanding of the field, and the questions for learnability and computational efficiency are active areas of research even for a single-layer neural network, or in some case, even a single neuron (our work applies to any number of layers and neurons). __Therefore, we feel that this criticism is a bit harsh and unfair.__
>
> 2. __Question__: "What is meant by “data-driven lens” and how it significantly differs from prior training conventions."
>
>    __Answer__: The data-driven setting assumes that we have to __tune the hyperparameter for multiple related tasks comes from some application-specific task distribution__. Our goal is to tune the hyperparameter so that it performs well for future/unseen tasks that come from such a task distribution. Unlike prior work with theoretical guarantees (e.g. Bayesian Optimization), this learning-theoretic framework enables much stronger guarantees that are distribution-independent (previous work assumes strong priors like Gaussian data and the guarantees depend on the Gaussian variance parameters) and do not need unrealistic smoothness or convexity assumptions on the function that captures the performance of the algorithm as a function of the hyperparameter. __We have discussed the data-driven line of work in Appendix A, as well as describing this setting in detail in the main body (see Section 2, from Line 171).__ We are happy to reiterate this point again in the final version of the paper.
>
> 3. __Question__: "The limitations to practical applications leveraging multiple hyperparameters are fairly apparent. I am not sure if the claimed contributions, even taken at face value, would be sufficient to compensate for that."
>
>    __Answer__:
>         - Again, we understand that practically, one might want to tune multiple hyperparameters. However, we want to emphasize again that the __theoretical analysis for the hyperparameter tuning is far lagging behind__. __As agreed by all other reviewers (p5Ux, uK76, and sAc2), our theoretical analysis, even for one single parameter, is a novel contribution and non-trivial__, requiring many advanced tools in algebraic and differential geometry.
>
>       - Moreover, __we strongly believe that our techniques can be helpful for future work to attack the high-dimensional hyperparameter case__. Many of __our key techniques can potentially be generalized and serve as critical components for attacking this case__. As observed by reviewers uK76 and p5uX, there are two key components in our analysis: bounded oscillations and monotonic curves. To attack the high-dimensional case, one may want to find the analog of those components in such a setting.
>
>      - For “monotonic curves”, a natural analog would be “monotonic hypersurfaces”, which should have this property: for any $\alpha’ \in \mathbb{R}^N$, there is at most one point $s$ in the monotonic hypersurface $M$ in the space $\mathcal{A} \times \mathcal{W}$ such that its $\alpha$-coordinate is equal to $\alpha’$, e.g., $s[\alpha] = \alpha’$. For example, the hyperplane $\alpha_1 = w, \alpha_2 = 2w$ is an example of monotonic hypersurface in the space $\mathbb{R}^2 \times \mathbb{R}$. We expect the condition of “monotonic hypersurface” would also be $J_{f, w}(\alpha, w)|_{(\alpha, w) = (a, b)} \neq 0$ when evaluating at any point $(a, b)$ in monotonic hypersurface $M$ (potentially with some additional conditions), and the extension idea will also base on Implicit Function Theorem, similar to “monotonic curves”.
>
>     - For “bounded oscillations”, we also expect a high dimensional extension: given a set of problem instances $x_1, \dots, x_N$ and real-valued thresholds $r_1, \dots, r_N$, can we control the number of connected components of a partition in the space $\mathcal{A}$ such that in each connected component $C$, $(\text{sign}(u^*_\alpha(x_1) - r_1)$,...,$\text{sign}(u_\alpha(x_N) - r_N))$ is fixed for any $\alpha \in C$. To do that, one might want to use some of our tools (e.g. Lemma F.9) as well as some more advanced tools from algebraic and differential geometry (e.g., algebraic projection, stratification, or surgery theory might potentially help), to first analyze the number of connected components formed by the projection of monotonic hypersurfaces on the space $\mathcal{A} \times \mathcal{W}$ onto the space $\mathcal{A}$, and then analyzing the behavior of $u_x(\alpha)$ on each component.
>
>    - Some of the other tools that we used are already applicable to high-dimensional cases (see e.g., Lemma F.9), and some additional tools have to be developed for the high-dimensional hyperparameter case, but we note that our approach offers concrete ideas and notions to attack that problem. We are happy to incorporate this discussion in the final version of the paper.
>
> ## Summary
> We hope that our answers sufficiently address the reviewer’s concerns, and we are happy to answer any further questions. __We respectfully request that the reviewer re-evaluate our work in light of our rebuttal. Many thanks!__
>
> ## References
>
> [1] Sitan Chen and Yuanzhi Li, Provably learning a multi-head attention layer, STOC’25

---

> ### Author Response · Authors · 2025-08-05
> **Follow up on Reviewer x4WD**
>
> We are reaching out to follow up on our response and to check if you have any further questions. In our rebuttal, we addressed your concerns about:
> 1. We show multiple other scenarios, where __theoretical papers presented at highly respected conferences in the field also initiate the theoretical understanding by analyzing very simple scenarios__ corresponding to important deep learning problems (e.g., studying a single-layer network or even a single neuron). We can provide many examples if requested, and we want to emphasize that, __though lagging behind empirical research, theoretical research can also bring insightful views and be helpful for future research.__
>
> 2. The scope of our results by showing that it is __novel and non-trivial, and foremostly, our overall approach can potentially be extended to the high-dimensional case__. Furthermore, there are already several interesting single hyperparameters, e.g., in activation functions and graph kernels.
>
> We hope that this resolves the weaknesses outlined in your review and would __really appreciate a prompt response for confirmation, or any additional questions or clarifications__. Thank you again for your thoughtful feedback.

---

### Official Review · Reviewer_p5UX · 2025-06-30

**Clarity:** 3
**Significance:** 3
**Originality:** 4
**Rating:** 5
**Confidence:** 3

**Summary:**

This work tackles the challenge of hyperparameter tuning in deep learning by analyzing its sample complexity in a multi-task setting, where tasks are drawn from an unknown distribution. The authors develop a theoretical framework to study the learning complexity of tuning hyperparameters, focusing on utility functions that depend on both hyperparameters and model parameters. By analyzing the structure of the dual utility function—which measures performance under optimal model parameters—they derive bounds on its discontinuities and oscillations using tools from differential geometry, algebraic geometry, and constrained optimization.

Their key results include sample complexity guarantees for tuning hyperparameters when the underlying parameter-dependent dual function is piecewise constant or piecewise polynomial. They apply these insights to practical settings, such as interpolating between activation functions in neural networks and tuning graph kernel parameters in graph neural networks. Additionally, they relax the assumption of exact optimization, showing their guarantees hold even with approximate empirical risk minimization. The work provides a rigorous foundation for understanding hyperparameter selection in multi-task learning scenarios.

**Questions:**

1. The piecewise-polynomial assumption for the dual function $f_x(\alpha, w)$ is central to the theory. Is there any possible empirical studies (even on simple synthetic tasks) that the theoretical results holds for some toy models?

2. (related to weaknesses 2) Could the analysis be extended to multiple hyperparameters? What are the major challenges in such multi-dimensional scenarios.

3. The theoretical guatantee is build on the exact minimizer of the optimization problem (u^*_{x}(\alpha) = \sup_{w}f_{x}(w,\alpha)). However, in practice, optimized with solvers that converge to approximate solutions. I am wondering whether the structural properties (e.g., discontinuities, oscillations of $u_x^*$) are preserved under approximation.

**Ethical Concerns:**

["NO or VERY MINOR ethics concerns only"]

**Limitations:**

Yes

**Paper Formatting Concerns:**

I did not notice any major formatting issues in this paper.

**Quality:**

4

**Strengths And Weaknesses:**

Strengths:

1. Theoretical results are technically solid and fine-grained, proofs are well-structured. Although I didn't check every detail of the proofs, the results seem to present no apparent error.
2. Their results captures good characteristics of the parameter-dependent utility function and its dual functions, and then establishes concrete bounds for learnability of the utility function class. New techniques are developed to support these results. The formalization of this general framework in analysis of hyperparameter tuning is a significant contribution.
3. This paper is well-written and easy to under stand, presenting complex theoretical concepts in a clear and accessible manner.

Weaknesses:

1. The theoretical result is developed for one-dimensional hyperparameter, whilst real-world tuning often involves many hyperparameters of very different nature (such as learning rates, dropout rates, batch sizes...). Extending the analysis to hyperparameter spaces with larger dimensions remains an open challenge.
2. I am concerning the applicability of the result in practice. Since the theory is build over the draw of problem instances $S=\left\{x_1, \ldots, x_m\right\} \sim \mathcal{D}^m$, as most real-world problems occurs to one dataset, without access to multiple independent problem instances for hyperparameter tuning.

---

> ### Author Rebuttal · Authors · 2025-07-31
>
> We thank the reviewer for carefully reviewing our paper. We appreciate that the reviewer found our paper clear and easy to understand and values our technical contributions. We will address the reviewer’s concerns as follows.
>
> ## Questions
> 1. __Question__: "Theoretical is developed for one-dimensional … extending the analysis to high-dimensional hyperparameter spaces remains an open challenge."
>
>    __Answer__:  It is true that our results hold for a single scalar hyperparameter. However, we note that this contribution is already novel and requires novel connection between algebraic/differential geometries and learning theory. Moreover, we strongly believe that our techniques can be helpful for future work to attack the high-dimensional hyperparameter case. See below for the concrete example.
>
> 2. __Question__: "The theory is built over the draw of problem instances, while most real-world problems occur in one dataset."
>
>     __Answer__: We note that our data-driven framework is designed for a different, and increasingly common, class of problems where the goal is to find hyperparameters that generalize across a distribution of related tasks. In this framework, a “problem instance” can also be interpreted as a task, not necessarily a massive, monolithic dataset, and we may want to learn how to set the best hyperparameter for fine-tuning for a new unseen task. This setting relates to several modern machine learning paradigms, for example:
>
>       - Multi-task learning: this is the most related application. Imagine a scenario where a company wants to build predictive models for hundreds of different clients, each has its own dataset/problem instance $x_i$. Our theory provides guarantees for learning a hyperparameter $\alpha$ (i.e., weight decay magnitude of NNs), using the datasets of previous clients, for a new client in the future.
>
>      -  Meta-learning: In meta-learning, the training data itself consists of a large number of small learning tasks.
>
>      - Scientific applications (i.e., GNNs for drug discovery): in this case, a single problem instance is a single graph already.
>   Therefore, though not covering every scenario of machine learning, we strongly believe that our contribution is still valuable for future work.
>
> 3. __Question__: The piecewise-polynomial assumption for the dual function fx(α,w) is central to the theory. Is there any possible empirical studies (even on simple synthetic tasks) that the theoretical results holds for some toy models?
>
>    __Answer__: Indeed, the piecewise-polynomial structure of NN outputs is a __very well-known and classical fact in the machine learning literature__ (see e.g., [1,2,3] for examples). This is also the foremost motivation of our analysis. Moreover, __in terms of our results (e.g., Theorem 6.1, H.1, H.2, H.3), we give concrete proofs for the piecewise polynomial structure__, not just assuming that property. For example, we showed the piecewise structure in the following scenarios:
>
>       - Polynomial kernels: E.g., tuning the scaling parameter $\gamma$ in $k_\textup{poly}(x_1, x_2) = (\gamma(x_1 \cdot x_2) + c_0)^d$.
>
>      -  Interpolating two distance metrics using a polynomial map $k_\textup{interp}(x_1, x_2) = (\alpha d_1(x_1, x_2) + (1 - \alpha)d_2(x_1, x_2) + c_0)^d$.
>
>      - Another example includes tuning the hyperparameter in the parametric ReLU (PReLU) activation function given by $f(x​)= x , \text{ if } x>0   \text{ and }  f(x​) = \alpha x, \text{ for } x\le 0$, where $\alpha$ is a learnable hyperparameter.
>
>    __There are further examples, for which we have concrete proofs (see e.g., Theorem 6.1 (from line 1534), Theorem H.1 (from line 1561), Theorem H.3 (from line 1606)).__
>
> 4. __Question__: "(related to weaknesses 2) Could the analysis be extended to multiple hyperparameters? What are the major challenges in such multi-dimensional scenarios."
>
>    __Answer__: We thank the reviewer for asking this question. Though providing analysis on a single hyperparameter, we strongly believe that our technique can serve as a starting point for future works attacking the high-dimensional hyperparameter scenario. As observed by Reviewer uK76, there are two key components in our analysis: bounded oscillations and monotonic curves. To attack the high-dimensional case, one may want to find the analog of those components in such a setting.
>
>       - For “monotonic curves”, a natural analog would be “monotonic hypersurfaces”, which should have this property: for any $\alpha’ \in \mathbb{R}^N$, there is at most one point $s$ in the monotonic hypersurface $M$ in the space $\mathcal{A} \times \mathcal{W}$ such that its $\alpha$-coordinate is equal to $\alpha’$, e.g., $s[\alpha] = \alpha’$. For example, the hyperplane $\alpha_1 = w, \alpha_2 = 2w$ is an example of monotonic hypersurface in the space $\mathbb{R}^2 \times \mathbb{R}$. We expect the condition of “monotonic hypersurface” would also be $J_{f, w}(\alpha, w)|_{(\alpha, w) = (a, b)} \neq 0$ when evaluating at any point $(a, b)$ in monotonic hypersurface $M$ (potentially with some additional conditions), and the extension idea will also base on Implicit Function Theorem, similar to “monotonic curves”.
>
>     - For “bounded oscillations”, we also expect a high dimensional extension: given a set of problem instances $x_1, \dots, x_N$ and real-valued thresholds $r_1, \dots, r_N$, can we control the number of connected components of a partition in the space $\mathcal{A}$ such that in each connected component $C$, $(\text{sign}(u^*_\alpha(x_1) - r_1)$,...,$\text{sign}(u_\alpha(x_N) - r_N))$ is fixed for any $\alpha \in C$. To do that, one might want to use some of our tools (e.g. Lemma F.9) as well as some more advanced tools from algebraic and differential geometry (e.g., algebraic projection, stratification, or surgery theory might potentially help), to first analyze the number of connected components formed by the projection of monotonic hypersurfaces on the space $\mathcal{A} \times \mathcal{W}$ onto the space $\mathcal{A}$, and then analyzing the behavior of $u_x(\alpha)$ on each component.
>
>    Some of the other tools that we used are already applicable to high-dimensional cases (see e.g., Lemma F.9), and some additional tools have to be developed for the high-dimensional hyperparameter case, but we note that our approach offers concrete ideas and notions to attack that problem. We are happy to incorporate this discussion in the final version of the paper.
>
> 5. __Question__: "The theoretical guarantee is built on the exact minimizer of the optimization problem (u^{x}(\alpha) = \sup{w}f_{x}(w,\alpha)). However, in practice, optimized with solvers that converge to approximate solutions. I am wondering whether the structural properties (e.g., discontinuities, oscillations of $ u_x^*$) are preserved under approximation."
>
>    __Answer__: We thank the reviewer for this insightful question. __Actually, this is exactly what we cover in Section 5__, where we assume that we only have access to an approximate solution $v_\alpha(x)$ that is close enough to $u^{x}(\alpha) = \sup{w}f_{x}(w,\alpha)$. We note that the underlying structure remains unchanged because the structure of $f_{x}(w,\alpha)$ is unchanged. What actually changes is the point $w’$ that we use to evaluate the performance of the network — this is $w^*$ that maximizes $f_{x}(w,\alpha)$ for the exact case $u_x(\alpha)$, or $\tilde{w}_\alpha$ such that $f(\tilde{w}, \alpha)$ is close to $u_x(\alpha)$ for the approximation case (e.g., $w'$ is learned using SGD with momentum).
>
> ## Summary
> Again, we thank the reviewer for the very fruitful discussion. We hope that we have sufficiently addressed the reviewer’s concern, and we are happy to address further questions from the reviewer.
>
> ## References
>
> [1] Peter Bartlett, Vitaly Maiorov, and Ron Meir. Almost linear VC dimension bounds for piecewise polynomial networks. NeurIPS’98
>
> [2] G Montúfar, R Pascanu, K Cho, Y Bengio, On the number of linear regions of deep neural networks, NeurIPS’14
>
> [3] PL Bartlett, N Harvey, C Liaw, A Mehrabian, Nearly-tight VC-dimension and pseudodimension bounds for piecewise linear neural networks, JMLR’19

---

> > ### Comment · Reviewer_p5UX · 2025-08-06
> >
> > Thanks for addressing my concerns. Given the authors' rebuttal, I see the potential of extending the method to multiple hyperparameters scenario. It is also interesting (for future work) to see an case study of this protocol on real data. I will remain my rating of this paper.

---

### Official Review · Reviewer_uK76 · 2025-07-03

**Clarity:** 3
**Significance:** 3
**Originality:** 3
**Rating:** 5
**Confidence:** 2

**Summary:**

This paper introduces a novel theoretical framework to analyze the sample complexity of tuning a single, continuous hyperparameter for deep neural networks in a data-driven, multi-task setting. The central challenge addressed is that the network's performance for a given hyperparameter, $\alpha$, is defined implicitly through an optimization over the high-dimensional weight parameter, $w$. This makes the utility function, $u_x^*(\alpha) = \sup_w f_x(\alpha, w)$, difficult to analyze directly.

The authors' main contribution is a new technique that leverages tools from differential and algebraic geometry to bound the learning-theoretic complexity of the utility function class. They show that even if $u_x^*(\alpha)$ is not a well-behaved function (e.g., not piecewise polynomial), its structural properties can be bounded. Specifically, by bounding the number of discontinuities and local extrema of $u_x^*(\alpha)$, they can bound its pseudo-dimension. This is achieved by analyzing the geometry of the underlying parameter-dependent dual function, $f_x(\alpha, w)$. The analysis is presented for two cases: when $f_x(\alpha, w)$ is piecewise constant and, more generally, when it is piecewise polynomial. These theoretical results are then instantiated to provide sample complexity guarantees for tuning an activation function interpolation parameter and a graph kernel parameter in GNNs.

**Questions:**

1.  he restriction to a single hyperparameter ($\alpha \in \mathbb{R}$) is the main limitation. The analysis relies on properties of 1D curves and partitioning the real line. Could you elaborate on the fundamental obstacles to extending this to $\alpha \in \mathbb{R}^n$? What would be the high-dimensional analog of "bounded oscillations" or the "monotonic curves" that are central to your proof? A more detailed discussion on a potential (even if speculative) path forward would greatly strengthen the paper's perceived long-term impact.

2.  The sample complexity bounds (e.g., in Theorem 6.1) have dependencies on network size ($W, L$) and polynomial degrees. Could you comment on whether you believe these bounds are tight or if they are primarily intended to show polynomial learnability? Furthermore, regarding Assumption 1 (ELICQ/ND), could you provide some intuition on why one might expect these to hold (or fail) for a standard architecture like a ResNet with common loss functions? For instance, do symmetries in the network architecture pose a problem for the LICQ condition?

3.  Theorem 5.1 accounts for an approximate-ERM oracle via an additive error term $\xi$. In practice, the optimization error from GD-based methods might depend on the hyperparameter, i.e., $\xi = \xi(\alpha)$, as $\alpha$ can affect the difficulty of the optimization landscape. Does your analysis assume $\xi$ is a constant independent of $\alpha$? If so, how would the results change if $\xi$ is a function of $\alpha$, and what assumptions might be needed on the function $\xi(\alpha)$ to retain the guarantees?

**Ethical Concerns:**

["NO or VERY MINOR ethics concerns only"]

**Final Justification:**

My questions are well addressed and I will keep my score.

**Limitations:**

yes

**Quality:**

3

**Strengths And Weaknesses:**

Strength:

1. Addresses the critical and under-theorized problem of hyperparameter tuning in deep learning.
2. The technical approach is highly novel, applying advanced geometric tools in a creative way to a problem in learning theory.
3. The work appears technically rigorous and provides a foundational result for a difficult problem.

Weakness:

1. The analysis is restricted to a single hyperparameter, and the path to extending it to multiple hyperparameters is not obvious.
2. The theory relies on regularity assumptions that may be difficult to verify for large, practical deep learning models.
3. The paper is very technical and dense, which may limit the reach of its core ideas to a broad audience.

---

> ### Author Rebuttal · Authors · 2025-07-31
>
> We thank the reviewer for the very careful and constructive feedback. We appreciate that the reviewer found our technique highly novel and creative, and our results to be foundational for a difficult problem. We address the reviewer’s concerns and questions below.
>
> ## Questions
> 1. __Question__:"The fundamental obstacles to extending this to the high-dimensional case? What would be the analog for “bounded oscillations” and the “monotonic curves” that are central to your proof."
>
>     __Answer__: We appreciate your deep understanding of our technique and this brilliant question! It is true that “bounded oscillations” and “monotonic curves” are two important recipes of our proofs, and they are the key to generalizing this framework to the high-dimensional case.
>
>      - For “monotonic curves”: a natural analog would be “monotonic hypersurfaces”, which should have this property: for any $\alpha’ \in R^N$, there is at most one point $s$ in the monotonic hypersurface $M$ in the space $\mathcal{A} \times \mathcal{W}$ such that its $\alpha$-coordinate is equal to $\alpha’$, e.g., $s[\alpha] = \alpha’$. For example, the hyperplane $\alpha_1 = w, \alpha_2 = 2w$ is an example of monotonic hypersurface in the space $\mathbb{R}^2 \times \mathbb{R}$. We expect the condition of “monotonic hypersurface” would also be $|J_{f, w}(\alpha, w)|_{(\alpha, w) = (a, b)} \neq 0$ when evaluating at any point $(a, b)$ on the monotonic hypersurface $M$ (potentially with some additional conditions), and we also expect the use of the Implicit Function Theorem, similar to “monotonic curves”.
>
>       - For “bounded oscillations”, we also expect a high dimensional extension: given a set of problem instances $x_1, \dots, x_N$ and real-valued thresholds $r_1, \dots, r_N$, can we control the number of connected components of a partition in the space $\mathcal{A}$ such that in each connected component $C$, $(\text{sign}(u^*_\alpha(x_1) - r_1)$,...,$\text{sign}(u_\alpha(x_N) - r_N))$ is fixed for any $\alpha \in C$. To do that, one might want to use some of our tools (e.g. Lemma F.9) as well as some more advanced tools from algebraic and differential geometry (e.g., algebraic projection, stratification, or surgery theory might potentially help), to first analyze the number of connected components formed by the projection of monotonic hypersurfaces on the space $\mathcal{A} \times \mathcal{W}$ onto the space $\mathcal{A}$, and then analyzing the behavior of $u_x(\alpha)$ on each component.
>
>    Some of the other tools that we used are already applicable to high-dimensional cases (see e.g., Lemma F.9), and some additional tools have to be developed for the high-dimensional hyperparameter case, but we note that our approach offers concrete ideas and notions to attack that problem. We are happy to incorporate this discussion in the final version.
>
> 2. __Question__: "Could you comment on whether you believe these bounds are tight or if they are primarily intended to show polynomial learnability? Furthermore, regarding Assumption 1 (ELICQ/ND), could you provide some intuition on why one might expect these to hold (or fail) for a standard architecture like a ResNet with common loss functions? For instance, do symmetries in the network architecture pose a problem for the LICQ condition?".
>
>     __Answer__: This is a great question. Developing lower bounds to determine if our bounds are tight is an interesting question, and the main achievement of the current work is to establish a polynomial upper bound on the sample complexity (which was previously unknown for hyperparameter tuning in neural networks).
>
>      Regarding Assumption 1 (ELICQ/ND), one would expect these conditions to hold almost surely for a random problem instance $x$ (or hold for a very slight perturbation of the problem instance if not the original problem instance $x$, thanks to Sard’s theorem, see Theorem F.15, line 1041 in Appendix). For example, see our Figure 1 ($w \in \mathbb{R}^2$), page 6, where we use a random $x$ which dictates the coefficient of $f_x(\alpha, w)$, and the geometry behaves as expected. We expect that in the very high-dimensional regime, the degenerate case will be even more unlikely to occur, and even if it occurs, a small perturbation on the problem instance $x$ should do the trick (which is a key technique of our proof, relaxing Assumption 3 (very complicated, see line 1268) to Assumption 1 (see from line 1391), and we suspect that a more advanced perturbation analysis can even remove Assumption 1 completely. Also, intuitively, symmetries should actually reduce the complexity of the dual utility function $u_x(\alpha)$ (e.g. fewer discontinuities and local extrema), even though the proofs are technically more challenging and perturbation analysis seems more technically feasible.
>
> 3. __Question__: “In Theorem 5.1, what if the optimization error $\xi$ depends on $\alpha$, as $\alpha$ can affect the difficulty of the optimization? Does your analysis assume $\xi$ is a constant independent of $\alpha$? If so, how would the results change if $\xi$ is a function of $\alpha$, and what assumptions might be needed on the function $\xi(\alpha)$ to retain the guarantee?”
>
>    __Answer__: We assume that the error of the approximate optimizer of the network is upper bounded by $\xi$ for all $\alpha$, although it need not be the same for all $\alpha$. While it seems interesting to try to get a sharper bound in terms of $\xi(\alpha)$ instead of a uniform approximation error bound of $\xi$ for all $\alpha$, prior work ([1] Theorem 4.9) shows examples indicating the tightness of this analysis.
>
> ## Summary
> Again, we thank the reviewer for the very fruitful discussion. We hope that we have sufficiently addressed the reviewer’s concern, and we are happy to address further questions from the reviewer.
>
>
> ## References
> [1] Balcan et al. Refined Bounds for Algorithm Configuration: The Knife-edge of Dual Class Approximability, ICML’20

---

> > ### Comment · Reviewer_uK76 · 2025-08-06
> >
> > I want to thank the authors for the reply. I also read other reviews and discussions and decided to keep my rating.

---

### Official Review · Reviewer_sAc2 · 2025-07-19

**Clarity:** 3
**Significance:** 2
**Originality:** 3
**Rating:** 4
**Confidence:** 2

**Summary:**

- Hyperparameter tuning across tasks is still dominated by heuristic search – theory lags behind practice.
- The paper develops a learning-theoretic framework that links the number of discontinuities and local extrema of the task-wise dual utility to the pseudo-dimension of the whole class.
- The paper analyzes two structural cases: (i) piecewise-constant and (ii) piecewise-polynomial parameter-dependent dual functions $f_x(\omega,w)$, yielding explicit sample-complexity bounds.
- Applications illustrate how the theory certifies learnability when tuning an activation-function interpolation parameter and a GNN kernel bandwidth, although only single-hyperparameter settings are treated.

**Questions:**

1. Can the authors supply concrete evidence that common activations or graph kernels yield piecewise-polynomial $f_x$ in practice?
2. How do the bounds scale when extending to two continuous hyperparameters?
3. Compared to Bayesian optimization, do your bounds imply a smaller or larger sample requirement under comparable smoothness assumptions? A discussion would clarify practical relevance.

**Ethical Concerns:**

["NO or VERY MINOR ethics concerns only"]

**Final Justification:**

Authors addressed my concerns satisfactorily

**Quality:**

2

**Strengths And Weaknesses:**

**Strengths**

- *Originality* - Introduces a novel bridge between algebraic/differential-geometric regularity of deep nets and classical capacity measures.
- *Quality* - Proofs are mathematically detailed and cover both piecewise-constant and considerably harder piecewise-polynomial regimes.
- *Significance* - Provides the first non-trivial sample-complexity guarantees for continuous hyperparameter tuning in deep networks.

**Weaknesses**

- *Clarity* - Core definitions (e.g. monotonic curves, ELICQ/ND conditions) are deferred to long appendices; the main text gives few intuitive examples, making it hard to grasp practical meaning.
- *Significance* - Results hold for a single scalar hyperparameter and require piecewise-polynomial structure – assumptions whose prevalence in modern architectures is unverified.
- *Quality* - No experiments demonstrate whether the theoretical bounds are tight or how they translate to wall-clock savings vs. Bayesian optimization; even toy simulations would strengthen claims.

---

> ### Author Rebuttal · Authors · 2025-07-31
>
> We thank the reviewer for their time and their recognition of the importance and lack of theoretical results for hyperparameter tuning, given that the state-of-the-art is largely dominated by heuristic search. We also appreciate that the reviewer finds our results novel, non-trivial, and significant, and acknowledges the technical challenge as well as the connection between the advanced geometrical tools and learning theoretic results that we draw upon. We address the reviewer’s concerns and questions below.
>
> ## Questions
> 1. __Question__: "Core definition (e.g., monotonic curves, ELICQ/ND conditions) are deferred to appendices; the main text gives few intuitive examples, making it hard to grasp practical meaning."
>
>     __Answer__: It is true that we have to defer some of the core definitions (including “monotonic curves”) to the Appendix, mainly due to the page limit. However, __for ELICQ/ND conditions, we give formal definitions as well as a detailed discussion (see Page 6,  Assumption 1 and Remark 1)__, and we highlight the __key property of monotonic curves in the appropriate context within the proof sketch (lines 294-295, 308-310)__. We will elaborate further on the core notions (e.g. monotonic curves) in the main body for better readability in the final version of the paper.
>
> 2. __Question__: "Results hold for a single scalar hyperparameter and require piecewise-polynomial structure-assumptions whose prevalence in modern architectures is unverified./Can the authors supply concrete evidence that common activations or graph kernels yield piecewise-polynomial fix in practice?"
>
>    __Answser__:
>
>       - __On the piecewise-polynomial property__: Indeed, the piecewise-polynomial structure of NN output is a __very well-known and classical fact in the machine learning literature__ (see e.g., [1,2,3] for examples) for piecewise-polynomial activation functions, including the popular ReLU activation (for ReLU the activation itself is piecewise linear, but the network output is a piecewise polynomial function of the weights and hyperparameters). This is also the foremost motivation of our analysis. Moreover, __in terms of our results (e.g., Theorem 6.1, H.1, H.2, H.3), we give concrete proofs for the piecewise polynomial structure__ for actual examples of hyperparameters in neural network activation functions and graph kernels, not just assuming that property. For example, we have shown the piecewise-polynomial structure in the following scenarios:
>         - Polynomial kernels: E.g., tuning the scaling parameter $\gamma$ in $k_\textup{poly}(x_1, x_2) = (\gamma(x_1 \cdot x_2) + c_0)^d$
>        - Interpolating two distance metrics using a polynomial map $k_\textup{interp}(x_1, x_2) = (\alpha d_1(x_1, x_2) + (1 - \alpha)d_2(x_1, x_2) + c_0)^d$.
>        - Another example includes tuning the hyperparameter in the parametric ReLU (PReLU) activation function given by $f(x​)= x , \text{ if } x>0   \text{ and }  f(x​) = \alpha x, \text{ for } x\le 0$, where $\alpha$ is a learnable hyperparameter.
>        - There are further examples, for which we have concrete proofs (see e.g., Theorem 6.1 (from lime 1534), Theorem H.1 (from line 1561), Theorem H.3 (from line 1606)).
>    - __On the single-scalar hyperparameter scenario__: It is true that our results hold for a single scalar hyperparameter. However, as the reviewer agrees, __this is a novel and non-trivial technical contribution that requires a novel connection between algebraic/differential geometries and learning theory__. Solving the non-trivial single hyperparameter setting achieved in this work is crucial if one ever hopes to understand the structure of the neural network behaviour as a function of multiple hyperparameters. Moreover, __our analysis also serves as a basis for potential future work on tuning multiple hyperparameters__, as __many ideas of our techniques are concrete and helpful for such extension__ (as noticed by reviewers uK76 and p5UX). Concretely, the following key components of our analysis can be extended into a high-dimensional case:
>      - For “monotonic curves”: a natural analog would be “monotonic hypersurfaces”, which should have this property: for any $\alpha’ \in R^N$, there is at most one point $s$ in the monotonic hypersurface $M$ in the space $\mathcal{A} \times \mathcal{W}$ such that its $\alpha$-coordinate is equal to $\alpha’$, e.g., $s[\alpha] = \alpha’$. For example, the hyperplane $\alpha_1 = w, \alpha_2 = 2w$ is an example of monotonic hypersurface in the space $\mathbb{R}^2 \times \mathbb{R}$. We expect the condition of “monotonic hypersurface” would also be $|J_{f, w}(\alpha, w)|_{(\alpha, w) = (a, b)} \neq 0$ when evaluating at any point $(a, b)$ on the monotonic hypersurface $M$ (potentially with some additional conditions), and we also expect the use of the Implicit Function Theorem, similar to “monotonic curves”.
>       - For “bounded oscillations”, we also expect a high dimensional extension: given a set of problem instances $x_1, \dots, x_N$ and real-valued thresholds $r_1, \dots, r_N$, can we control the number of connected components of a partition in the space $\mathcal{A}$ such that in each connected component $C$, $(sign(u^*_\alpha(x_1) - r_1)$,...,$sign(u_\alpha(x_N) - r_N))$ is fixed for any $\alpha \in C$. To do that, one might want to use some of our tools (e.g. Lemma F.9) as well as some more advanced tools from algebraic and differential geometry (e.g., algebraic projection, stratification, or surgery theory might potentially help), to first analyze the number of connected components formed by the projection of monotonic hypersurfaces on the space $\mathcal{A} \times \mathcal{W}$ onto the space $\mathcal{A}$, and then analyzing the behavior of $u_x(\alpha)$ on each component.
>
>        Besides, some other additional extensions of our techniques and arguments have to be developed for the high-dimensional hyperparameter case, but we note that our approach offers concrete ideas and notions to attack that problem. We are happy to incorporate this discussion in the final version of the paper.
>
> 3. __Question__: "How do the bounds scale when extending to two continuous hyperparameters?"
>
>    __Answer__: We note that extension to two continuous hyperparameters is an interesting question for further investigation. As discussed above, we believe that our work can serve as a strong starting point to attack the high-dimensional scenario.
>
> 4. __Question__: "Compared to Bayesian optimization, do your bounds imply a smaller or larger sample requirement under comparable smoothness assumptions? A discussion would clarify practical relevance."
>
>    __Answer__: Great question. Strictly speaking, our results are incomparable with the sample requirement in Bayesian optimization for the following reasons:
>
>       - We give uniform convergence bounds, which imply small generalization error for *any* approach that achieves a small training set error. This includes a bound on the error of the hyperparameter learned by Bayesian optimization even if the assumptions of smoothness and bounded noise needed in the usual analysis of Bayesian optimization do not hold. Our results say if the training sample size is large enough (roughly Pdim/epsilon^2 using our bounds on the pseudo-dimension Pdim), then the test error will be within epsilon of the training error.
>
>     - Bayesian optimization assumes a Gaussian prior and the sample complexity depends on the prior parameters (e.g. bound on variance) as well as kernel hyperparameters, while our results are more worst-case and apply to arbitrary unknown distributions. Our assumptions are more realistic, at least in the case of neural networks with piecewise polynomial activation functions. Also, note that Bayesian optimization requires careful tuning of additional hyperparameters (e.g. kernel and acquisition function design).
>
>    To summarize, our theory complements the known bounds for Bayesian optimization and also applies in broader, more worst-case contexts. We are happy to include this discussion in our paper and thank the reviewer for prompting this comparison.
>
> ## Summary
> We hope that our answers address the reviewer’s concerns, and we are happy to answer any further questions from the reviewer. We respectfully request the reviewer to re-evaluate our work in light of our rebuttal. Many thanks!
>
> ## References
> [1] Peter Bartlett, Vitaly Maiorov, and Ron Meir. Almost linear VC dimension bounds for piecewise polynomial networks. NeurIPS’98
>
> [2] G Montúfar, R Pascanu, K Cho, Y Bengio, On the number of linear regions of deep neural networks, NeurIPS’14
>
> [3] PL Bartlett, N Harvey, C Liaw, A Mehrabian, Nearly-tight VC-dimension and pseudodimension bounds for piecewise linear neural networks, JMLR’19

---

> ### Author Response · Authors · 2025-08-05
> **Follow up on Reviewer sAc2**
>
> We are reaching out to follow up on our response and to check if you have any further questions. In our rebuttal, we addressed your concerns about:
> 1. core definitions deferred to Appendix, by pointing out that __some of the mentioned definitions actually presented formally as well as informally in the main body__ (ELICQ/ND conditions and their discussion, Page 6), and the __rest have been discussed intuitively__ (monotonic curves, lines 294-295, 308-310),
> 2. The scope of our results by showing that it is __novel and non-trivial__, and foremostly, the __general idea can potentially be extended to the high-dimensional case__.
> 3. Comparison to the Bayesian optimization (__which we already did, see lines 32-45__), showing that __our result is fundamentally different and is hard to compare with it.__
>
> We hope that this resolves the weaknesses outlined in your review and __would really appreciate a follow-up response for confirmation, or any additional questions or clarifications__. Thank you again for your thoughtful feedback.

---

### Note · Authors · 2025-08-11

Here is our final remark on the fruitful rebuttal:

1. On the scope and primary contributions of our work as a learning theory paper:
    - __Novelty and technical challenge__:  Theoretical understanding of tuning even a single hyperparameter (without assumptions on smoothness or distribution niceness) required highly non-trivial and novel techniques that were developed in this work. For example, we are not aware of any previous work giving concrete bounds on the pseudodimension for general function classes defined in terms of a potentially highly non-convex optimization problem.
     - __A foundation for future works__: as noticed by reviewer uK76 and p5UX, our efforts in solving the one-hyperparameter (and multiple parameters, that is $w \in \mathbb{R}^d$) can pave the way for future works handling multiple hyperparameters.

2. On other concerns:
     - Deferred core definitions (e.g., monotonic curves, ELICQ/ND conditions): we did present ELICQ/ND conditions in detail (page 6, Assumption 1 and Remark 1). For the monotonic curves, we discussed the key properties in the appropriate context within the proof sketch (lines 294-295, 208-310).
     - Comparison to Bayesian optimization: We gave a detailed comparison and distinction between our results and the Bayesian method in the rebuttals. We provided that discussion already in the main body (lines 32-45).
     - Clarification on the data-driven settings: We explained again the data-driven setting and how our result is novel, technically challenging in this setting. We also provided this discussion already in the main body (lines 171-185) and the Appendix (Appendices A, B).

Finally, we thank the reviewers and the AC for your time and efforts!

---

### Decision · Program_Chairs · 2025-09-17

**Decision:**

Accept (poster)

**Comment:**

This paper proposes a framework for theoretically analyzing the complexity of hyperparameter tuning in machine learning (e.g., deep learning) settings. They use the so-called dual utility function to derive bounds on the learning complexity using technical tools from differential/algebraic geometry in tandem with optimization. Under the assumption that the underlying parameter-dependent dual function belongs to a certain class (e.g., piecewise polynomial), they are able to give rigorous complexity guarantees.

The strengths of the paper are mostly in its rigorous, original approach to the problem. The setting is quite general as well, since the lower level problem is not required to have a unique solution, in contrast to many works on hyperparameter tuning that take a bilevel optimization perspective. Another strength would be that the paper makes a lot of progress towards bridging the gap between rigorous theory and practical empirics, by allowing for inexact optimization (which allows approximate empirical risk minimization).

The main weakness of the paper has to do with the dimensionality of the hyperparameters - it is assumed that they are one-dimensional. Reviewer x4WD pointed this out as a weakness and the authors responded in the rebuttal that it's very difficult to make progress in this very general setting (i.e., without a unique solution to the lower level problem) and that it might eventually be possible to extend this approach to high-dimensional problems. I agree with Reviewer x4WD that this is a weakness but I also agree with the authors that their work represents nontrivial progress even if it cannot yet handle the high-dimensional case yet.

Since, on the whole, the reviews of the paper were positive and the main criticism, brought by Reviewer x4WD, was sufficiently addressed during the rebuttal period, I recommend accepting the paper.